# MULTI-DRAFT SPECULATIVE SAMPLING: CANONICAL DECOMPOSITION AND THEORETICAL LIMITS

[*]**Ashish Khisti**[1 2]         [*]**M.Reza Ebrahimi**[1]         **Hassan Dbouk**[1]

**Arash Behboodi**[1]         [†]**Roland Memisevic**[1]         [†]**Christos Louizos**[1]

[1]Qualcomm AI Research[‡]  [2]University of Toronto

## ABSTRACT

We consider multi-draft speculative sampling, where the proposal sequences are sampled independently from different draft models. At each step, a token-level draft selection scheme takes a list of valid tokens as input and produces an output token whose distribution matches that of the target model. Previous works have demonstrated that the optimal scheme (which maximizes the probability of accepting one of the input tokens) can be cast as a solution to a linear program. In this work we show that the optimal scheme can be decomposed into a two-step solution: in the first step an importance sampling (IS) type scheme is used to select one intermediate token; in the second step (single-draft) speculative sampling is applied to generate the output token. For the case of two identical draft models we further 1) establish a necessary and sufficient condition on the distributions of the target and draft models for the acceptance probability to equal one and 2) provide an explicit expression for the optimal acceptance probability. Our theoretical analysis also motives a new class of token-level selection schemes based on weighted importance sampling. Our experimental results demonstrate consistent improvements in the achievable block efficiency and token rates over baseline schemes in a number of scenarios.

## 1 INTRODUCTION

The transformer architecture (Vaswani et al., 2017) has revolutionized the field of natural language processing and deep learning. One of the key factors contributing to the success story of transformers, as opposed to prior recurrent-based architectures (Hochreiter and Schmidhuber, 1997; Chung et al., 2014), is their inherent train-time parallelization due to the attention mechanism. This allows for massive scaling and lead to the development of state-of-the-art Large Language Models (LLMs) (Touvron et al., 2023; Achiam et al., 2023; Brown et al., 2020; Chowdhery et al., 2023) which have demonstrated remarkable performance across a wide range of tasks. Despite their parallelizable training, LLM inference is sequential, owing to their auto-regressive nature. This limits their text-generation to one token per one forward pass, which is known to be memory-bound (Shazeer, 2019).

To alleviate the memory-bound nature of auto-regressive decoding of LLMs, speculative decoding (Chen et al., 2023; Leviathan et al., 2023) leverages an arbitrary smaller language model (draft model) that generates multiple candidate tokens in an auto-regressive manner. The LLM (target model) is then used to score all the tokens in the draft *in parallel*, and the draft tokens are verified through a sequence of token-level rejection sampling which guarantees that the final sequence follows the same distribution as that of the target model. In order for speculative decoding to be beneficial, the combined cost of auto-regressively sampling from the draft model and parallel verification via the target model should be smaller than auto-regressively sampling from the target model. Intuitively, this requires that the draft model distribution resembles that of the target model, which can be measured via the acceptance rate of the speculative decoding process, i.e., the rate at which we accept/reject draft tokens.

[*] Joint first authors [†] Joint last authors [‡] Qualcomm AI Research is an initiative of Qualcomm Technologies, Inc. Correspondence to akhisti@ece.utoronto.ca and ebrahimi@qti.qualcomm.com.

A large number of works on speculative decoding (Sun et al., 2024b; Jeon et al., 2024; Miao et al., 2024; Sun et al., 2024a) have emerged recently in an effort to further improve decoding efficiency. The authors in (Sun et al., 2024b) propose SpecTr, a multi-draft extension where the draft model generates $K$ candidate token sequences (which could be sampled in a batch) for each time-step (as opposed to one). The authors consider a token-level selection scheme with the objective of maximizing the probability of accepting some token in the set of available tokens. They demonstrate that this problem can be cast into the framework of optimal transport and solved using a linear program. However due to complexity reasons, the authors instead propose a modified sequential rejection sampling scheme.

## 1.1 MAIN CONTRIBUTIONS

- We revisit the optimal transport framework introduced in (Sun et al., 2024b) and introduce a canonical decomposition — we demonstrate that the optimal acceptance probability can be achieved by a two-step scheme: the first step involves selecting a token from the input set using a type of importance sampling; the second step involves speculative sampling using the selected token and the target distribution.

- For the case of $K = 2$ identical draft models, we establish an analytical expression for the optimal acceptance probability. In this setting, we also establish a necessary and sufficient condition for the acceptance probability to equal one. We also discuss numerical evidence for the case of more than 2 drafts.

- We propose a new token-selection scheme based on weighted importance sampling. To enable a faster implementation, we present heuristic approaches that reduce computation while also penalizing the acceptance probability.

- We present experimental results involving the OPT model over a variety of tasks. We compare the performance of our proposed schemes with baselines and demonstrate improvements in the block efficiency and token rates.

## 2 BACKGROUND AND RELATED WORK

Auto-regressive sampling from LLMs is inherently sequential and memory-bound (Shazeer, 2019). Several approaches have been proposed in the literature to accelerate LLM inference (Shazeer, 2019; Jaszczur et al., 2021; Frantar et al., 2022; Frantar and Alistarh, 2023; Stern et al., 2018; Chen et al., 2023; Leviathan et al., 2023; Jeon et al., 2024; Sun et al., 2024b; Miao et al., 2024). Model compression techniques, such as quantization (Frantar et al., 2022; Bondarenko et al., 2024) and sparsification (Jaszczur et al., 2021; Frantar and Alistarh, 2023) have been shown to reduce the overall complexity of LLMs at the expense of some degradation in decoding quality.

For lossless LLM inference acceleration, speculative decoding (Chen et al., 2023; Leviathan et al., 2023; Stern et al., 2018) has emerged as a promising and orthogonal alternative. Earlier works on greedy decoding can draft and predict multiple tokens by augmenting the base LLM (Stern et al., 2018) or aggressive decoding (Ge et al., 2022). However, LLM text-generation often requires sampling with non-zero temperature from the generated logits. To that end, speculative decoding (Chen et al., 2023; Leviathan et al., 2023) was proposed. In speculative decoding, auto-regressive sampling is delegated to a smaller language model (draft model) that generates multiple candidate tokens. The LLM (target model) is then used to score all the tokens in the draft *in parallel*, and the draft tokens are verified through a sequence of token-level rejection sampling. Speculative decoding guarantees that the final sequence follows the same distribution as that of the target model. The performance of speculative methods highly depends on the choice of the draft model. Zhou et al. (2023) use knowledge distillation (Hinton et al., 2015) to better align the draft and target models which results in higher token acceptance rates.

More recently, the works of Sun et al. (2024b); Miao et al. (2024); Jeon et al. (2024) extend speculative decoding to the multi-draft setting where the draft model(s) generate multiple token sequences per time-step. Specifically, Sun et al. (2024b) formulate the token-level draft selection problem as a discrete optimal transport problem with membership cost and propose SpecTr: a new decoding algorithm that allows for multiple candidates for each token in the draft. A related setting is also studied in Miao et al. (2024); Jeon et al. (2024) where the authors consider a token tree based construction for improving the draft sequences as well as a token-level selection method

different form Sun et al. (2024b). Instead of using a dedicated draft model, Cai et al. (2024) propose augmenting the target model with extra decoding heads that can concurrently draft multiple tokens. The extra heads are fine-tuned using parameter-efficient methods, and can be added to any pre-trained target model. Orthogonally, Sun et al. (2024a) study block-level verification in the single-draft setting as a block-level optimal transport problem. They propose a computationally-efficient algorithm that optimally solves the block-level transport problem, and report speedups over prior token-level verification (Leviathan et al., 2023).

## 3 TOKEN-LEVEL OPTIMAL DRAFT SELECTION: THEORETICAL ANALYSIS

We focus on token-level optimal draft selection framework introduced in Sun et al. (2024b). For sake of completeness we review the speculative sampling schemes in Leviathan et al. (2023); Chen et al. (2023) in Appendix A. We assume that $\Omega = \{1, 2, \ldots, n\}$ denotes the vocabulary of tokens and at a given step, say $t$, $\mathcal{S} = \{X_1, \ldots, X_K\}$, denotes the $K$ valid tokens under consideration. Each of these tokens is generated in an i.i.d. fashion from a distribution $p(\cdot)$ determined by the underlying draft model and the context sequence $u^t \in \Omega^t$ i.e, for each $y \in \Omega$, we have $p(y) = \mathcal{M}_s(y|u^t)$, where $\mathcal{M}_s$ denotes the distribution generated by the small (draft) model. In a similar fashion we let $q(\cdot)$ be the distribution over $\Omega$ associated with the large model i.e., $q(y) = \mathcal{M}_b(y|u^t)$ where $\mathcal{M}_b$ denotes the distribution generated by the large model. Note that we do not explicitly indicate the sequence $u^t$ when discussing $p(\cdot)$ and $q(\cdot)$, as it is fixed and common to both models throughout our analysis.

Given an input $\mathcal{S} \sim \prod_{i=1}^{K} p(X_i)$ consisting of $K$ candidate tokens $(X_1, \ldots, X_K)$, a *token-level selection rule* (TLSR) is a conditional distribution $\mathcal{P}(\cdot|\mathcal{S})$ over $\Omega$. A *valid* TLSR must satisfy the constraint that for each $z \in \Omega$, $\sum_{\mathcal{S}} \mathcal{P}(z|\mathcal{S})p(\mathcal{S}) = q(z)$. A natural metric to optimize for TLSR is the probability that one of the tokens is accepted i.e., if $Z \sim \mathcal{P}(\cdot|\mathcal{S})$ denotes the output of the TLSR, then we wish to maximize $\Pr(Z \in \mathcal{S})$.

**Problem 1 (Optimal Token Level Selection Rule)** *Given distributions $p(\cdot)$ and $q(\cdot)$ find a valid TLSR that maximizes the probability of acceptance: $P(\text{acc}) = \Pr(Z \in \mathcal{S})$ and let $P^\star(\text{acc})$ be the optimal value.*

Problem 1 was studied in Sun et al. (2024b) and shown to be an instance of optimal transport, which can be cast as a linear program. The authors used this framework to establish the optimality of speculative sampling (Chen et al., 2023; Leviathan et al., 2023) in the case of a single draft i.e., $K = 1$. For $K > 1$ the authors established an information theoretic upper bond on $P^\star(\text{acc})$. In this work, we revisit Problem 1 and develop new insights into the structure of the optimal solution. In fact, we establish that the optimal solution in the case of multiple drafts has a natural connection to importance sampling (Tokdar and Kass, 2010). For the case of $K = 2$ drafts we exactly characterize $P^\star(\text{acc})$ and state necessary and sufficient conditions on $p(\cdot)$ and $q(\cdot)$ for $P^\star(\text{acc})$ to equal 1.

We begin by defining a family of schemes that we will refer to as *importance weighted* sampling.

**Definition 1 (Importance Weighted Sampling)** *An importance weighted sampling scheme takes as input the set of candidate tokens $\mathcal{S} = \{X_1, \ldots, X_K\}$ and outputs a token $Y_I \in \mathcal{S}$ defined by the conditional distribution:*

$$\Pr(Y_I = y | X_{1:K} = x_{1:K}) = \begin{cases} \beta_y(x_1, \ldots, x_K), & y \in \{x_1, \ldots, x_K\} \\ 0, & y \notin \{x_1, \ldots, x_K\} \end{cases} \tag{1}$$

*where $\sum_{y \in \Omega} \beta_y(x_1, \ldots, x_K) = 1$ for each $x_{1:K} \in \Omega^K$ and $0 \le \beta_y(x_1, \ldots, x_K) \le 1$.*

Note that instead of considering the probability over the value of the selected token in (1), one can instead consider the probability of selecting an index $i$ between $\{1, \ldots, K\}$ i.e., $\Pr(I = i|X_{1:K} = x_{1:K})$. Such a distribution maps to (1) by simply summing over all indices where $x_i = y$. We note that the form in (1) will be more convenient in the sequel. Also note that the classical importance sampling scheme (Tokdar and Kass, 2010) corresponds to the case where $\Pr(I = i|X_{1:K} = x_{1:K}) \propto q(x_i)/p(x_i)$. However the family of schemes in Definition 1 is not

---

We also consider the case when the tokens are generated from different distributions. See Remark 1 as well as the experimental results in Section 5.

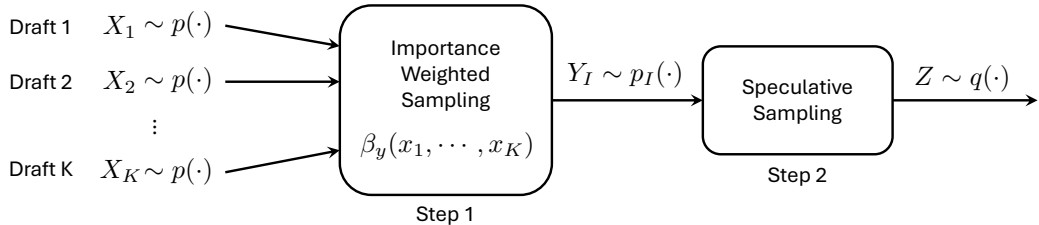

Figure 1: Optimal Approach for Multi-Draft Speculative Sampling

restricted to such a choice and we treat $\beta_y(x_1, \ldots, x_K)$ as free parameters that can be optimized. While an importance weighted sampling scheme may not lead be a valid TLSR, as explained next, it is a key building block in a canonical decomposition for token selection.

Our first result is a decomposition for the optimal token level selection rule that establishes a connection to the importance weighted sampling in Definition 1. The proof is in Appendix B.

**Theorem 1 (Optimal Acceptance Probability and Canonical Decomposition)** *Let $P^\star(\mathrm{acc})$ be the acceptance probability for the optimal token level selection rule in Problem 1. Then we can express:*

$$P^\star(\mathrm{acc}) = \max_{\{\beta_y(x_{1:K})\}} \left\{ \sum_{y \in \Omega} \min \left( q(y), \sum_{x_1, \ldots, x_K \in \Omega} \beta_y(x_{1:K}) \cdot \prod_{i=1}^{K} p(x_i) \right) \right\}, \qquad (2)$$

*where the maximum is over $\beta_y(x_{1:K})$ for each $\{x_1, \ldots, x_K, y\} \in \Omega$ such that $0 \leq \beta_y(x_{1:K}) \leq 1$, and*

$$\sum_{y \in \Omega} \beta_y(x_{1:K}) = 1, \quad \forall x_{1:K} \in \Omega^K, \qquad (3)$$

*and furthermore*

$$\beta_y(x_{1:K}) = 0, \quad y \notin \{x_1, \ldots, x_K\}. \qquad (4)$$

*In addition, if $\{\beta_y^\star(x_{1:K})\}$ denotes the parameters that achieve the maximum in (2), then $P^\star(\mathrm{acc})$ can be attained by a two step canonical decomposition: in the first step, given the list of input tokens $\{x_1, \ldots, x_K\}$, we apply Importance Weighted Sampling in Definition 1 with parameters $\beta_y^\star(x_1, \ldots, x_K)$ to output an intermediate token $y \in \{x_1, \ldots, x_K\}$; in the second step we apply a single-draft speculative sampling scheme (Chen et al., 2023; Leviathan et al., 2023) on the selected token $y$ to generate the final output token.*

Figure 1 illustrates the proposed system in Theorem 1, where the first step involves importance weighted sampling to output an intermediate token and the second step involves speculative sampling. This approach requires computing the optimal $\beta_y^\star(x_{1:K})$. In practice one can use sub-optimal choices that are faster to compute, as will be discussed in the sequel.

**Remark 1** *Although our theoretical development focuses on the case of identical drafts, our result in Theorem 1 naturally extends when the the $K$ tokens are instead sampled from a joint distribution i.e., $p(x_1, \ldots x_K)$. This is discussed in Section B.1 in the supplementary material. In particular Theorem 1 naturally extends to the setting when the $K$ tokens are sampled independently from different distributions $\mathcal{S} \sim \prod_{i=1}^{K} p_i(X_i)$ as in Miao et al. (2024) as well as to the case when the tokens are sampled without replacement (Jeon et al., 2024).*

We next build upon Theorem 1 to establish new analytical results for the optimal acceptance probability involving $K = 2$ drafts. Our first result is a characterization of the necessary and sufficient condition on the draft and target distributions $p(\cdot)$ and $q(\cdot)$ respectively that leads to $P^\star(\mathrm{accept}) = 1$.

**Theorem 2** *With $K = 2$ drafts, a necessary and sufficient condition for $P^\star(\mathrm{acc}) = 1$ in the Definition 1 is the following:*

$$\sum_{x \in \mathcal{S}} q(x) \geq \left( \sum_{x \in \mathcal{S}} p(x) \right)^2, \qquad \forall \mathcal{S} \subseteq \Omega. \qquad (5)$$

Note that the acceptance probability can equal 1 even when $p(\cdot)$ and $q(\cdot)$ are not identical. Thus when the distribution of the draft model is close to the target model but not equal the acceptance probability can equal 1. This is in contrast to the case of $K = 1$, where it is known that the acceptance probability can only equal 1 when $p(\cdot)$ and $q(\cdot)$ are identical distributions (Sun et al., 2024b). Furthermore to the best of our knowledge, previously proposed schemes for the multi-draft setting, such as SpecTr (Sun et al., 2024b) and SpecInfer (Miao et al., 2024) based on modified rejection sampling also require $p(\cdot) = q(\cdot)$ for the acceptance probability to be 1. Theorem 1 is interesting in the context of our two-step system in Fig. 1. In this case, the output of importance weighted sampling block $Y$ matches the target distribution $q(\cdot)$ and the second step involving speculative sampling is not needed.

**Example 1** *Consider $\Omega = \{1, 2\}$ and let the draft and target distributions be given by $\mathbf{p} = (p_1, p_2)$ and $\mathbf{q} = (q_1, q_2)$ respectively. We assume $K = 2$ drafts. In this case (5) reduces to $q_1 \geq p_1^2$ and $q_2 \geq p_2^2$. If $p_1 = p_2 = 0.5$ then it follows that $P^\star(\mathrm{acc}) = 1$ if and only if $0.25 \leq q_1 \leq 0.75$. In contrast for the optimal scheme for $K = 1$ draft we have $P^\star(\mathrm{acc}) = 1$ only when $q_1 = q_2 = 0.5$.*

The proof of Theorem 2 in Appendix C involves analyzing the output distribution $p_I(\cdot)$ of the Importance Weighted Sampling Scheme in Theorem 1 and demonstrating that a feasible choice of $\beta_y(x_1, x_2)$ exists and sets $p_I(\cdot) = q(\cdot)$ when the condition (5) is satisfied. The proof is based on the Fourier-Motzkin (FM) elimination technique (Ziegler, 2012). However a direct application of such a technique to satisfy the constraints $q(i) = p_I(i)$ for each $i \in \Omega$ becomes intractable. Our key idea is to demonstrate that instead considering a relaxation of the form $q(i) \geq p_I(i)$ leads to the same solution as the equality constraints and is amenable to analysis using Fourier-Motzkin elimination. We explain this further with an example involving $\Omega = \{1, 2, 3\}$ in Appendix C.

The core technical challenge in the proof of Theorem 2 is in determining whether a system of linear equations has a non-negative solution. Such problems have been studied previously in the literature, with Chernikova (1964); Dines (1926) providing an algorithm. Such considerations lead to a geometric viewpoint involving polyhedral cones which we discuss in Appendix D. We explain how the double-description method (Fukuda and Prodon, 1995) for finding dual representations of polyhedral cones can be used to numerically verify the necessary and sufficient condition for the acceptance probability to equal 1. In fact this approach was used to verify analogous conditions to Theorem 2 for up to $K = 6$ drafts and all alphabets of size $|\Omega| \leq 14$, although we only provide an analytical proof of the condition for $K = 2$ drafts in this paper. The reason for this is that our key step i.e., Lemma 2 in the Appendix that makes use of Fourier-Motzkin elimination, does not easily generalize to the case of $K > 2$ drafts.

Our final result is an explicit expression for the optimal acceptance probability for the case of $K = 2$ drafts.

**Theorem 3** *For $K = 2$ drafts and for a draft distribution $p(\cdot)$ and target distribution $q(\cdot)$ and arbitrary token alphabet $\Omega$, the acceptance probability $P^\star(\mathrm{acc})$ for the optimal token level selection rule is given by:*

$$P^\star(\mathrm{acc}) = \min_{\mathcal{S} \subseteq \Omega} \left\{ \sum_{s \in \mathcal{S}} q(s) + \left( \sum_{s \in \mathcal{S}^c} p(s) \right)^2 + 2 \left( \sum_{s \in \mathcal{S}} p(s) \right) \left( \sum_{s \in \mathcal{S}^c} p(s) \right) \right\}, \qquad (6)$$

*where $\mathcal{S}^c = \Omega \setminus \mathcal{S}$ is the complement of $\mathcal{S}$.*

To the best of our knowledge the result in Theorem 3 was not known before. Upper bounds on $P^\star(\mathrm{acc})$ are presented in Sun et al. (2024b), which are not necessarily tight. In contrast (6) provides an exact expression for the acceptance probability for the case of $K = 2$ drafts when $X_1$ and $X_2$ are independently sampled from $p(\cdot)$. The proof of Theorem 3, presented in Appendix E, applies the Fourier-Motzkin elimination to the linear program presented in Theorem 1 to characterize an analytical solution in the case of $K = 2$ drafts. The proof builds upon the proof of Theorem 2 but requires elimination of additional variables.

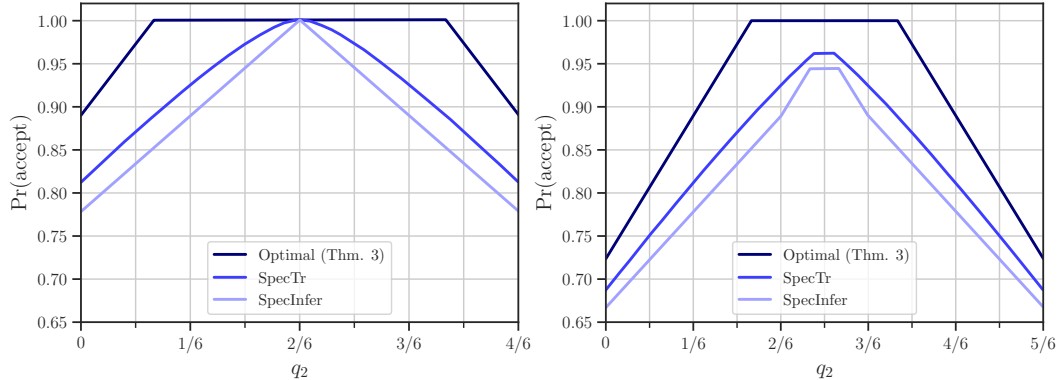

Figure 2: Numerical evaluation of $Pr(\text{accept})$ for the optimal scheme (Theorem 3) as well as two baseline schemes – SpecTr (Sun et al., 2024b) and SpecInfer (Miao et al., 2024). For sake of illustration we select alphabet $\Omega = \{1, 2, 3\}$ and $\mathbf{p} = [1/3, 1/3, 1/3]$. The left plot sets $\mathbf{q} = [1/3, q_2, 2/3 - q_2]$ while the right plot sets $\mathbf{q} = [1/6, q_2, 5/6 - q_2]$ where $q_2$ is varied on the x-axis.

**Remark 2** *Note that* (6) *can be expressed as:*

$$P^\star(\text{acc}) = \min_{\mathcal{S} \subseteq \Omega} \left\{ \sum_{s \in \mathcal{S}} q(s) - \left( \sum_{s \in \mathcal{S}} p(s) \right)^2 + 1 \right\}, \tag{7}$$

*which leads us to conjecture that the optimal acceptance probability in the general case of $K > 2$ drafts is attained by replacing the exponent of $2$ in the second term in* (7) *to $K$.*

We provide numerical evaluation of the optimal acceptance probability in Fig. 2. For sake of illustration we assume that $\Omega$ is of size three, and assume $\mathbf{p} = [1/3, 1/3, 1/3]$. We consider $\mathbf{q} = [1/3, q_2, 2/3 - q_2]$ in the left plot and $\mathbf{q} = [1/6, q_2, 5/6 - q_2]$ in the right plot. The value of $q_2$ is varied on the $x$-axis. We compare the optimal acceptance probability in Theorem 3 with two baseline schemes SpecTr (Sun et al., 2024b) and SpecInfer (Miao et al., 2024). We observe that the optimal acceptance probability can equal 1 for a wide range of $q_2$. This is consistent with Theorem 2. In contrast the baseline schemes seem to achieve an acceptance probability of 1 only in the special case when $q_2 = 1/3$ so that $\mathbf{q} = [1/3, 1/3, 1/3]$.

## 4 Faster Importance Weighted Speculative Sampling

We discuss fast approaches for computing $\beta_y(\cdot)$ in our canonical decomposition in Fig. 1. Ideally we wish to select $\beta_y(\cdot)$ that maximize the acceptance probability in (2), but this involves a computationally expensive linear program. Fortunately we can develop procedures for faster computation that still achieve high acceptance probability. In practice the distribution of the target and draft model is often concentrated over a small number of tokens. It has also been observed that sampling from a high probability set such as the top-$p$ set (the set of high probability tokens with aggregate probability exceeding a threshold) leads to more coherent outputs (Meister et al., 2023). After such sampling the effective alphabet size, i.e., the number of tokens with non-zero probability, is generally small. We report some measurements of the effective alphabet size for the OPT model in Appendix H. This motivates us to develop some approaches for speeding up our proposed solution by reducing the number of variables required in optimization. Throughout this section we consider the case when the drafts have identical distribution and discuss the case of non-identical draft distributions in Appendix I. We focus on the case of $K = 2$ drafts and then discuss how to tackle the general case.

**Linear Program for $K = 2$ drafts**: We first consider the case of $K = 2$ drafts and revisit the linear program that needs to be solved to compute the optimal acceptance probability in (2). We will then explain how to reduce the variables in the linear program for faster computation. Let $X_1$ and $X_2$ denote the input tokens and $Y$ denote the selected token. Note that when $X_1 = X_2 = i$, we have that $\beta_i(i, i) = 1$. Furthermore when the draft models have identical distributions, from symmetry,

we have $\beta_y(i,j) = \beta_y(j,i)$. It is more convenient to introduce a new set of variables $w_{i,j}$ which are defined as:

$$w_{i,j} = \Pr(Y = i \mid \{X_1, X_2\} = \{i, j\}), \tag{8}$$

i.e., $w_{i,j}$ denotes the probability that the output token is $i$ given that we see the pair $\{i, j\}$ at the input in any order. We next discuss how to formulate a linear program involving the variables $w_{i,j}$ to maximize the acceptance probability in (2). For convenience we let $\mathbf{p} = (p_1, \ldots, p_n)$ as the probability vector for the draft model and $\mathbf{q} = (q_1, \ldots, q_n)$ as the probability vector for the target model. The acceptance probability i.e., the objective in (2), is given by $\sum_{i=1}^n \min(p_I(i), q_i)$, where

$$p_I(k) = p_k^2 + \sum_{i=1, i \neq k}^n 2p_i p_k w_{k,i} \tag{9}$$

denotes probability that the selected token is $Y = k$. Furthermore for our definition of $w_{i,j}$ in (8), the associated constraints are: $w_{i,j} + w_{j,i} = 1$ and $0 \leq w_{i,j} \leq 1$. This optimization can be cast as a linear programming problem with $O(n^2)$ variables which may be slow in practice. We next discuss techniques to reduce the number of variables in optimization without a significant loss in the acceptance probability.

**Faster Approximations to Linear Program**: In order to propose a faster approximation, we note that when the draft and target distribution are concentrated over a few tokens, the linear programming solution will not be sensitive to most choices of $w_{i,j}$. As a result one can heuristically set most of the variables. We refer to this method as the *truncated LP* scheme and present the pseudocode in Algorithm 2 in Appendix G. Assume that the vocabulary $\Omega = \{1, 2, \ldots, n\}$ has the tokens sorted in decreasing order i.e., $q_1 - p_1^2 \geq q_2 - p_2^2 \ldots \geq q_n - p_n^2$. We partition $\Omega$ into two sets $\Omega_1 = \{1, 2, \ldots, s\}$ and $\Omega_2 = \{s+1, \ldots, n\}$, where $s$ is a design parameter to select. We fix a subset of weights as follows:

$$w_{i,j} = \begin{cases} 1, & i \in \Omega_1, j \in \Omega_2 \\ 1, & i \in \Omega_2, j \in \Omega_2, i < j \end{cases} \tag{10}$$

while we leave the weights $w_{i,j}$ for $i < j$ and $i, j \in \Omega_1$ as free parameters. The intuition behind the choice of weights in (10) is that in these cases we prefer token $i$ over token $j$ to increase $p_I(i)$ further, which is in turn can decrease the difference between $q_i$ and $p_I(i)$. Note that (9) reduces to:

$$p_I(k) = \begin{cases} p_k^2 + \sum_{i=1, i \neq k}^s 2p_i p_k w_{k,i} + \sum_{i=s+1}^n 2p_i p_k, & k \in \Omega_1 \\ p_k^2 + \sum_{i=k+1}^n 2p_i p_k, & k \in \Omega_2 \end{cases} \tag{11}$$

Our objective is to maximize reduces to $\sum_{k=1}^s \min(p_I(k), q_k)$ over the variables $w_{i,j}$. Thus the number of variables is reduced to $O(s^2)$. We further show in Appendix F that if $P^\star(\mathrm{acc})$ is the optimal acceptance probability associated by applying the linear program over all $O(n^2)$ weight variables and $\tilde{P}(\mathrm{acc})$ is the acceptance probability for the truncated program then:

$$\tilde{P}(\mathrm{acc}) \geq P^\star(\mathrm{acc}) - \sum_{x \in \Omega_2} \left( q(x) - p^2(x) \right)^+ \tag{12}$$

Thus if $\Omega_2$ is selected so that the penalty term is small then the decrease in the acceptance probability can be kept small.

In the experiments we observed that for well-trained target models the drop in accuracy is negligible even for small values of $s$. Thus by appropriately truncating the number of variables to optimize in the linear program we expect to have a faster implementation. We discuss the computational complexity of the proposed method in Appendix G.

Our second proposal is to truncate the alphabet when performing importance sampling. In particular let $\Omega_0 \subseteq \Omega$ be a high probability subset of $\Omega$ under the target distribution. We re-normalize the target distribution to $\tilde{q}$ so that it is supported entirely over $\Omega_0$ and use this distribution in our proposed scheme to generate an output $Y \sim \tilde{q}(\cdot)$. Since we want the output to follow the distribution $q(\cdot)$, we perform the following post-processing. We accept $Y$ with a probability $p_a = \sum_{x \in \Omega_0} q(x)$ and reject with probability $p_r = 1 - p_a$. If rejected, we output a token from $\Omega \setminus \Omega_0$ such that any $x \in \Omega \setminus \Omega_0$ is selected with probability proportional to $q(x)$. This ensures that the output token follows the original distribution $q(\cdot)$. We refer to this scheme as the *truncated alphabet* scheme.

**Remark 3** *To tackle the case of $K > 2$ drafts we propose to group the input tokens into groups of size $2$ and then apply the two-draft importance sampling scheme in a multi-stage manner. For example if $S = \{X_1, X_2, X_3\}$ and $K = 3$ we first apply the fast importance weighted sampling to the group $\{X_1, X_2\}$ to output an intermediate token $Y_1$ with distribution say $p_1(\cdot)$. Then we apply importance weighted sampling to the input $(Y_1, X_3)$, where the tokens now have non-identical distributions, and produce an output token $Y$ to which speculative sampling is applied.*

## 5 EXPERIMENTAL RESULTS

**Setup**. We conduct experiments using an instance of A100 GPU with 80GB memory. We use the OPT models (Zhang et al., 2022), where the draft model has 125 million parameters and the target model has 13B parameters. For evaluation purposes we consider the datasets associated with the XSum (Narayan et al., 2018), Databricks-Dolly-15k (Conover et al., 2023) and the WMT18 (Bojar et al., 2018) tasks. All experiments were done over 4 arbitrarily chosen seeds and the performance was averaged over these.

### 5.1 EXPERIMENTS WITH TOP-$p$ SAMPLING

In this section we report experiments on LLM tasks when top-$p$ sampling is used with $p = 0.95$. In our first set of experiments in Fig. 3, we consider the case of $K = 2$ draft models, which share the same weights but use different temperatures for token generation. We set the temperature of the target model to $1.0$, and one that of the draft models to $1.2$ while we vary the temperature of the other draft model between the range of $1.0$ to $2.4$. In all our experiments we generate 5 tokens per call of the draft model.

Our baseline scheme is the single-draft speculative sampling scheme (Leviathan et al., 2023; Chen et al., 2023) when we only use a single draft with sampling temperature of $1.2$. The other two schemes are the SpecInfer (Miao et al., 2024) and our proposed importance sampling (IS) scheme. In the IS scheme we employ both truncated LP (with $s = 5$ as the truncation parameter) and truncated alphabet (to a size of $40$ tokens) as discussed in section 4. In Fig. 3 we report the performance achieved by the different schemes across the three tasks. The top plots report the block efficiency, which is the average number of tokens that are accepted per use of the draft model (Leviathan et al., 2023). The block efficiency achieved by the single-draft scheme is shown by the horizontal dotted red line in each plot, while the block efficiency of the IS scheme and SpecInfer are shown using the dark blue and light blue bars. We observe that the IS scheme consistently outperforms SpecInfer

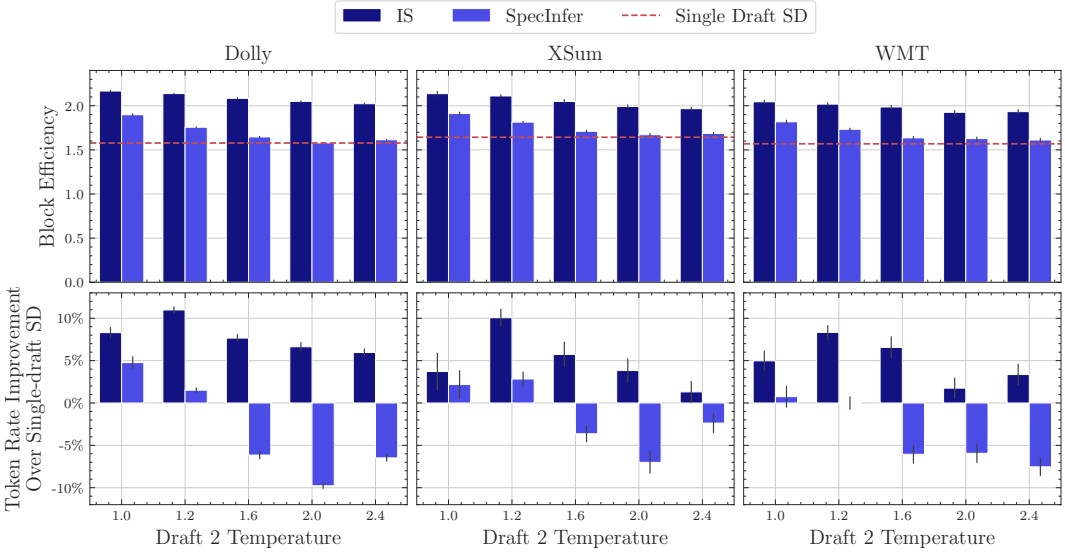

Figure 3: Performance comparison of different multi-draft schemes. The temperature of the first draft models is set to $1.2$, while we vary the temperature of the other draft.

across all three tasks. In fact when the temperature of the second draft is increased, the improvement in the block efficiency for SpecInfer is rather negligible compared to the single draft baseline. On the other hand our proposed IS scheme is able to achieve consistent improvements. The bottom plots show the percentage improvement in the token rate with respect to the baseline single draft scheme. We observe that when the temperature of the second draft increases, the gains are negative for the SpecInfer scheme as the computational time for the second draft does not translate into a sufficient improvement in the block efficiency. On the other hand our proposed importance sampling scheme shows a consistent improvement the token-rate over the baseline schemes due to improved block efficiency, despite the additional compute from the second draft. In the experiments involving the XSUM and WMT tasks we also measure the ROUGE-L (Lin, 2004) and BLEU (Papineni et al., 2002) scores respectively, which are reported in the Appendix J.3. In Appendix J.1 we provide a similar experiment when both drafts use identical temperatures. We are able to also include comparisons with the SpecTr scheme (Sun et al., 2024a) in that experiment. Likewise in Appendix J.2 we present a similar experiment involving $K = 3$ drafts.

In Table 1 we study the effect of choosing different parameters for vocabulary truncation and LP truncation discussed in Section 4. We use the same parameters as in the previous experiment but fix the temperature of the second draft to 2.0. As we increase the size of the vocabulary $\Omega_0$ from 10 to 40 we see improvements in the block efficiency as well as the token rates. Beyond 40, it appears that the gains in the block efficiency saturate and the token rate decreases. We also find that the block efficiency is not too sensitive to the choice of the parameter $s$ in the LP program and choosing $s = 5$ yields the best token rate.

Table 1: Effect of LP Truncation and Alphabet Truncation

|  |  | Block Efficiency | Token Rate % improvement to SD |
|---|---|---|---|
| Vocab. Truncation ($|\Omega_0|$) | 10 | $1.98 \pm 0.03$ | $-0.57 \pm 3.38\%$ |
|  | 20 | $2.00 \pm 0.04$ | $1.00 \pm 3.08\%$ |
|  | 40 | $2.05 \pm 0.04$ | $6.63 \pm 3.18\%$ |
|  | 50 | $2.03 \pm 0.05$ | $3.22 \pm 3.39\%$ |
| LP-Truncation Threshold ($s$) | 5 | $2.05 \pm 0.04$ | $6.63 \pm 3.18\%$ |
|  | 10 | $2.04 \pm 0.05$ | $1.52 \pm 3.47\%$ |
|  | 15 | $2.04 \pm 0.04$ | $1.74 \pm 2.36\%$ |

In Figure 4, we fix the temperature of the two draft models to 1.0, and vary the temperature the target model in the range of 0.2 to 1.0. The rest of the parameters remain the same. We again report the block efficiency and the improvement in the token rate over the single draft scheme with the same draft temperature over the Dolly task. As the two drafts have identical temperature we are also able to include comparisons with the SpecTr scheme. We again observe that our proposed method generally attains the best performance.

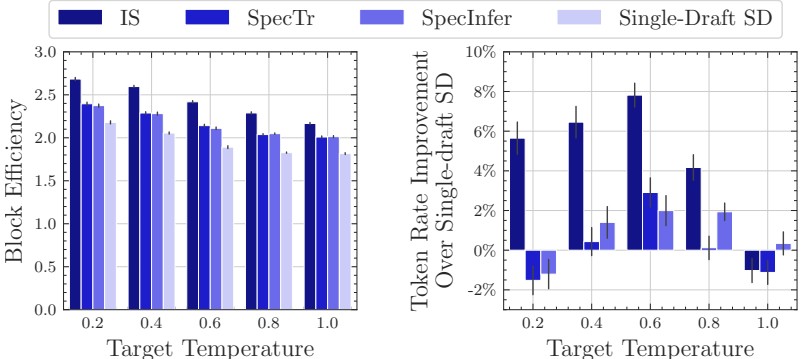

Figure 4: Performance comparison of different schemes on the Dolly task, while we vary the temperature of the target model and keeping the temperature of the two drafts to 1.0.

## 5.2 EXPERIMENTS WITH TOP-$k$ SAMPLING

We next consider experiments when top-$k$ sampling is applied to LLM tasks. We again use the same OPT models as in the previous subsection for both draft and target models. As before, we also sample a total of 5 tokens during each call to the draft model. In the Dolly task we use a sampling temperature of $1.0$ for both the target and draft models. In the XSUM task we use a temperature of $0.9$ for the target and a temperature of $0.3$ for the draft model.

In the first experiment, reported in Table 2, we compare the toke-level acceptance probability across different methods. We first generate a sequence of tokens via auto-regressive decoding using the target model. At each step we compute the logits generated by the draft and the target model, which we then use to compute the acceptance probability for different schemes. We average the acceptance probability over 100 instances. In the importance sampling (IS) scheme, we use a truncated linear program with a cut-off threshold $s = 5$. We also consider the theoretically optimal acceptance probability for the case of $K = 2$ drafts and also compute the natural extension of this expression for $K = 4, 8$ discussed in Remark 2 (reported using the gray font). We use top-$k$ sampling with $k = 5$ for both models in this experiment. We report the acceptance probabilities for the case of $K = 2, 4, 8$ drafts. In applying the IS scheme to handle $K > 2$ drafts, we use the successive selection scheme discussed in Remark 3. Note that the acceptance probability increases as we increase the number of drafts in each case.

Table 2: Comparison of average acceptance probability across different tasks.

| Scheme | XSUM | | | Dolly | | |
|---|---|---|---|---|---|---|
| | $K = 2$ | $K = 4$ | $K = 8$ | $K = 2$ | $K = 4$ | $K = 8$ |
| Optimal | 0.5009 | 0.5226 | 0.5419 | 0.6384 | 0.6731 | 0.6962 |
| IS | 0.4933 | 0.5145 | 0.5333 | 0.6348 | 0.6691 | 0.6919 |
| SpecTr | 0.4889 | 0.5083 | 0.5263 | 0.6246 | 0.6560 | 0.6800 |
| SpecInfer | 0.4875 | 0.5058 | 0.5227 | 0.6202 | 0.6489 | 0.6722 |

In Table 3 we compare the block efficiencies for different methods using $K = 2$ and $K = 3$ drafts. We apply top-$k$ sampling with $k = 10$ and $k = 5$ and use a temperature of $1.0$ for both models. In our IS scheme, we truncate the variables in the LP to $s = 7$. While our proposed scheme still achieves a higher block efficiency over prior works, the gains are relatively small. This can be explained by noting that in this setting the acceptance probability of the baseline schemes is already close to the theoretical limit, as noted in Table 2.

Table 3: Block Efficiency achieved in the Dolly Task with top-$k$ sampling

| Sampling | Scheme | $K = 2$ drafts | | $K = 3$ drafts | |
|---|---|---|---|---|---|
| | | Block Efficiency | Loss | Block Efficiency | Loss |
| | IS | $2.48 \pm 0.01$ | — | $2.59 \pm 0.02$ | — |
| top-$k$ ($k = 10$) | SpecTr | $2.43 \pm 0.01$ | 98% | $2.55 \pm 0.01$ | 98% |
| | SpecInfer | $2.38 \pm 0.02$ | 96% | $2.49 \pm 0.02$ | 96% |
| | IS | $2.52 \pm 0.02$ | — | $2.63 \pm 0.03$ | — |
| top-$k$ ($k = 5$) | SpecTr | $2.48 \pm 0.02$ | 98% | $2.56 \pm 0.03$ | 97% |
| | SpecInfer | $2.47 \pm 0.01$ | 98% | $2.55 \pm 0.04$ | 97% |

## 6 CONCLUSION

We revisit the problem of maximizing the acceptance probability for token-level multi-draft selection (Sun et al., 2024b). We first present a decomposition result that demonstrates connection to importance sampling. For the case of $K = 2$ (identical) drafts we establish a closed for solution for the acceptance probability and also present a necessary and sufficient condition for the acceptance probability to equal one. We propose a practical token selection scheme that mimics the theoretical decomposition principle and provides a way to tradeoff the computational speed with the acceptance probability. We present experimental results using OPT model and three tasks: Dolly, XSUM and WMT and demonstrate improvements in block efficiency and token rates over baselines schemes.

ACKNOWLEDGMENTS

We thank Mingu Lee, Wonseok Jeon, and Babak Ehteshami Bejnordi for their help with this work.

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

## A  Speculative Decoding – Background

We briefly review the proposal made in Leviathan et al. (2023); Chen et al. (2023). Let $\Omega$ denote the set of all permissible tokens associated with a language model. Given a context $x^t = (x(1), \ldots, x(t)) \in \Omega^t$ our target model $s$ produces conditional probabilities $\mathcal{M}_s(y|x^t)$ for each $y \in \Omega$. Following prior works and as explicitly stated in Sun et al. (2024b) we assume the following computational model:

1. **Standard Inference**: Given a context $x^t$, an auto-regressive model can output $\mathcal{M}_b(y|x^t)$ for each $y \in \Omega$ in $O(1)$ time.

2. **Parallelization along time-axis** Given a context $x^t$, an auto-regressive model can output $\mathcal{M}_b(y|x^i)$ for all $i \in \{1, \ldots, t\}$ and each $y \in \Omega$ again in $O(1)$ time.

Since the standard inference is a sequential process, the idea in Leviathan et al. (2023); Chen et al. (2023) is to exploit parallelization along time-axis to speed up inference. This is accomplished using a draft model $\mathcal{M}_s$ that is capable of generating multiple tokens with a much lower computation than $\mathcal{M}_b(\cdot)$. The main steps are summarized below. We assume that a context token $x^t$ is the input.

1. **Draft Construction**: The draft model can efficiently sample $L$ tokens in an auto-regressive manner, extending the context $x^t$ to $x(1), \ldots, x(t), \tilde{x}(t+1), \ldots, \tilde{x}(t+L)$. In addition, we keep the conditional probabilities $\mathcal{M}_s(y|x_1^t, \tilde{x}_{t+1}^{t+i})$ for each $0 \le i \le L$ and each $y \in \Omega$.

2. **Conditional Probability Computation**: Given the samples $\tilde{x}_{t+1}^{t+i}$ generated by $\mathcal{M}_s$ we compute the conditional probabilities $\mathcal{M}_b(y|x_1^t, \tilde{x}_{t+1}^{t+i})$ for each $0 \le i \le L$ and each $y \in \Omega$ using parallelization along time-axis.

3. **Draft Selection** We select first $L' \le L$ tokens and set $x(t+i) = \tilde{x}(t+i)$ for $i \le L'$. The selection is based on speculative decoding algorithm below.

---

**Algorithm 1** Speculative Sampling

1: **Input:** Distributions $p(\cdot)$ and $q(\cdot)$, Sample $X \sim p(\cdot)$
2: Compute residual distribution $p^{\text{res}}(x) = \frac{q(x) - \min(p(x), q(x))}{1 - \sum_{x'} \min(p(x'), q(x'))}$
3: Accept = False
4: With probability $\beta_{p,q}(X) = \min\left(1, \frac{q(X)}{p(X)}\right)$, set Accept = True.
5: **if** Accept = True **then**
6:     $Y = X$
7: **else**
8:     $Y \sim p^{res}(\cdot)$
9: **end if**
10: Return $Y$.

---

For convenience we denote $p(y) \equiv M_s(y|x^t)$ and $q(y) \equiv M_b(y|x^t)$. The selection procedure is as follows: Given a context $x^t$ and a candidate draft sequence $\tilde{x}_{t+1}^{t+L}$, the algorithm in the first step sets $p(\tilde{x}_{t+1}) = \mathcal{M}_s(\tilde{x}_{t+1}|x_1^t)$ and $q(\tilde{x}_{t+1}) = \mathcal{M}_b(\tilde{x}_{t+1}|x_1^t)$ respectively. It accepts $\tilde{x}_{t+1}$ with probability $\beta_{p,q}(\tilde{x}_{t+1}) = \min\left(1, \frac{q(\tilde{x}_{t+1})}{p(\tilde{x}_{t+1})}\right)$. If the token is accepted then $x(t+1) = \tilde{x}_{t+1}$ and we proceed with $\tilde{x}_{t+2}$. If the token is rejected then we sample $Y \sim p^{\text{res}}(\cdot)$ and set $x(t+1) = Y$. At this step we initiate a call to sample $L$ fresh tokens from the draft model with $x_1^{t+1}$ as input. This process continues until all $L$ tokens have been accepted or a token is rejected. The correctness of the proposed algorithm follows through direct computation of $\Pr(Y = y)$. We also compute the probability of accept:

$$\Pr(\text{accept}) = \sum_{x \in \mathcal{X}} \Pr(\text{acc}|X = x)p(x) \tag{13}$$

$$= \sum_{x \in \mathcal{X}} \beta_{p,q}(x)p(x) \tag{14}$$

$$= \sum_{x \in \mathcal{X}} \min(p(x), q(x)) = 1 - d_{TV}(p, q), \tag{15}$$

where $d_{TV}(\cdot)$ is the total variational distance. Note that $\Pr(\text{accept})$ is a key metric that determines the probability that a token from the draft model is accepted. The higher the value of $\Pr(\text{accept})$, the more efficient is the algorithm. In this paper we also study $\Pr(\text{accept})$ in the multi-draft setting.

## B    PROOF OF THEOREM 1

We will consider the case when there are $K$ drafts i.e., $X_1, \ldots, X_K$ are sampled i.i.d. from a draft model with distribution $p(\cdot)$, while the target model has a distribution of $q(\cdot)$. We assume the alphabet $\Omega = \{1, 2, \ldots, M\}$ for some arbitrary $M$.

**Analysis of Importance Weighted Sampling Scheme:**    We first consider the family of importance sampling schemes followed by speculative sampling and derive the acceptance probability. Assuming $Y$ denotes the selected sample in importance sampling, let:

$$\Pr(Y = y | X_1^K = x_1^K) = \begin{cases} \beta_y(x_1^K), & y \in \{x_1, \ldots, x_K\} \\ 0, & y \notin \{x_1, \ldots, x_K\} \end{cases} \tag{16}$$

where $\sum_y \beta_y(x_1, \ldots, x_K) = 1$ for each $x_1^K \in \Omega^K$ and $0 \leq \beta_y(x_1, \ldots, x_K) \leq 1$.

It follows that

$$\Pr(Y = y) = \sum_{x_1, \ldots, x_K \in \Omega} \beta_y(x_1^K) \cdot \prod_{i=1}^K p(x_i) \tag{17}$$

$$= \sum_{x_1, \ldots, x_K \in \Omega} \beta_y(x_1^K) \cdot \mathbb{I}_y(x_1, \ldots, x_K) \cdot \prod_{i=1}^K p(x_i) \tag{18}$$

where $\mathbb{I}_y(x_1, \ldots, x_K)$ denotes the indicator function that equals 1 if $y \in \{x_1, \ldots, x_K\}$ and equals 0 otherwise. Note that (18) follows since $\beta_y(x_1, \ldots, x_K) = 0$ if $y \notin \{x_1, \ldots, x_K\}$.

By applying speculative sampling to the selected sample $X_I$ the probability of acceptance is given by:

$$P^{\text{M-IS}}(\text{accept} = 1) = \sum_{y \in \Omega} \min(q(y), \Pr(X_I = y)) \tag{19}$$

$$= \sum_{y \in \Omega} \min\left( q(y), \sum_{x_1, \ldots, x_K \in \Omega} \beta_y(x_1^K) \cdot \mathbb{I}_y(x_1, \ldots, x_K) \cdot \prod_{i=1}^K p(x_i) \right) \tag{20}$$

Thus within the proposed class of importance sampling schemes, we can formulate our objective as:

$$\max_{\{\beta_y(x_1^K)\}_{y, x_1, \ldots, x_K}} \left\{ \sum_{y \in \Omega} \min\left( q(y), \sum_{x_1, \ldots, x_K \in \Omega} \beta_y(x_1^K) \cdot \mathbb{I}_y(x_1, \ldots, x_K) \cdot \prod_{i=1}^K p(x_i) \right) \right\} \tag{21}$$

such that $0 \leq \beta_y(x_1^K) \leq 1$ for each $y, x_1, \ldots, x_K \in \Omega$, and

$$\sum_{x_1^K \in \Omega^K} \beta_y(x_1^K) = 1, \quad \forall y \in \Omega, \tag{22}$$

and furthermore

$$\beta_y(x_1^K) = 0, \quad y \notin \{x_1, \ldots, x_K\}. \tag{23}$$

---

We use the notation $X_1^K$ as a short hand for $X_{1:K}$. Similarly we use $x_1^K$ as a shrot hand for $x_{1:k}$

**Analysis of Optimal Solution:** We now consider the problem of optimizing the acceptance probability for any given $p(\cdot)$ and $q(\cdot)$ in the general setting. Following the framework in Sun et al. (2024b), we seek to find $p_{Y|X_1,\ldots X_K}(y|x_1,\ldots,x_K)$ for each $y, x_1, \ldots, x_K \in \Omega$ such that we maximize

$$\Pr(\text{accept} = 1) = \Pr(Y \in \{X_1, \ldots, X_K\}) \tag{24}$$

subject to the marginal constraints on $P_Y(\cdot)$:

$$q(y) = \Pr(Y = y) = \sum_{x_1^K} \Pr(Y = y, X_1^K = x_1^K) = \sum_{x_1^k} p_{Y|X_1^K}(y|x_1^K) \prod_{i=1}^K p(x_i). \tag{25}$$

Next we consider:

$$q(y) = \sum_{x_1^k \in \Omega^k} p_{Y|X_1^K}(y|x_1^K) \prod_{i=1}^K p(x_i)$$

$$= \sum_{x_1^k \in \Omega^k} p_{Y|X_1^K}(y|x_1^K) \mathbb{I}_y(x_1, \ldots, x_K) \prod_{i=1}^K p(x_i)$$

$$+ \sum_{x_1^k \in \Omega^k} p_{Y|X_1^K}(y|x_1^K) \overline{\mathbb{I}}_y(x_1, \ldots, x_K) \prod_{i=1}^K p(x_i) \tag{26}$$

$$\geq \sum_{x_1^k \in \Omega^k} p_{Y|X_1^K}(y|x_1^K) \mathbb{I}_y(x_1, \ldots, x_K) \prod_{i=1}^K p(x_i) \tag{27}$$

where $\overline{\mathbb{I}}_y(x_1, \ldots, x_K) = 1 - \mathbb{I}_y(x_1, \ldots, x_K)$ denotes the complement of $\mathbb{I}$. Now note that:

$$\Pr(Y \in \{X_1, \ldots, X_K\})$$

$$= \sum_{x_1^K \in \Omega^K} \Pr(Y \in \{X_1, \ldots, X_K\} \mid X_1^K = x_1^K) p(X_1^K = x_1^K) \tag{28}$$

$$= \sum_{x_1^K \in \Omega^K} \sum_{y \in \Omega} p_{Y|X_1^K}(y|x_1^K) \mathbb{I}_y(x_1, \ldots, x_K) \left( \prod_{i=1}^K p(x_i) \right) \tag{29}$$

$$= \sum_{y \in \Omega} \sum_{x_1^K \in \Omega^K} p_{Y|X_1^K}(y|x_1^K) \mathbb{I}_y(x_1, \ldots, x_K) \left( \prod_{i=1}^K p(x_i) \right) \tag{30}$$

$$= \sum_{y \in \Omega} \min \left( q(y), \sum_{x_1^K \in \Omega_K} p_{Y|X_1^K}(y|x_1^K) \mathbb{I}_y(x_1, \ldots, x_K) \left( \prod_{i=1}^K p(x_i) \right) \right) \tag{31}$$

where we use (27) which implies that for any feasible $p_{Y|X_1^K}(y|x_1^K)$:

$$\sum_{x_1^k \in \Omega^k} p_{Y|X_1^K}(y|x_1^K) \mathbb{I}_y(x_1, \ldots, x_K) \prod_{i=1}^K p(x_i) \leq q(y) \tag{32}$$

is satisfied.

**Upper Bound on the optimal acceptance probability:** We now establish an upper bound on (31) and show that it coincides with the acceptance probability optimized in the importance weighted sampling scheme (21).

For each $x_1^K \in \Omega^K$, let us define

$$D(x_1^K) = \sum_{y \in \Omega} p_{Y|X_1^K}(y|x_1^K) \mathbb{I}_y(x_1, \ldots, x_K) \tag{33}$$

and furthermore with $N(x_1, \ldots, x_K)$ denoting the number of unique elements in $x_1^K$,

$$\tilde{p}_{Y|X_1^K}(y|x_1^K) = \begin{cases} \frac{p_{Y|X_1^K}(y|x_1^K)}{D(x_1^K)}, & y \in \{x_1, \ldots, x_K\}, \quad D(x_1^K) > 0 \\ \frac{1}{N(x_1 \ldots, x_K)} & y \in \{x_1, \ldots, x_K\}, \quad D(x_1^K) = 0, \\ 0 & y \notin \{x_1, \ldots, x_K\}. \end{cases} \tag{34}$$

Note by construction that for each $x_1^K \in \Omega^K$

$$\sum_{y \in \Omega} \tilde{p}_{Y|X_1^K}(y|x_1^K) = 1 \tag{35}$$

and

$$\tilde{p}_{Y|X_1^K}(y|x_1^K) = 0, \quad y \notin \{x_1, \ldots, x_K\} \tag{36}$$

and furthermore:

$$\tilde{p}_{Y|X_1^K}(y|x_1^K) \cdot \mathbb{I}_y(x_1, \ldots, x_K) \geq p_{Y|X_1^K}(y|x_1^K) \cdot \mathbb{I}_y(x_1, \ldots, x_K), \forall y, x_1, \ldots, x_K \in \Omega \tag{37}$$

Substituting (37) into (31) we have that for any feasible $p_{Y|X_1^K}(\cdot)$ there exists a $\tilde{p}_{Y|X_1^K}(\cdot)$ satisfying (35) and (36) such that:

$$Pr(Y \in \{X_1, \ldots, X_K\}) \leq \sum_{y \in \Omega} \min\left( q(y), \sum_{x_1^K \in \Omega_K} \tilde{p}_{Y|X_1^K}(y|x_1^K) \mathbb{I}_y(x_1, \ldots, x_K) \left( \prod_{i=1}^K p(x_i) \right) \right) \tag{38}$$

It thus follows that that optimal acceptance probability in the general case is upper bounded by optimizing the (38) over $\tilde{p}_{Y|X_1^K}(y|x_1^K)$ satisfying (35) and (36). But this problem precisely coincides with the optimization in the proposed class of IS schemes as stated in (21)-(23), thus establishing the optimality of the latter.

## B.1 EXTENSION BEYOND IID SETTING

The proof in Theorem 1 assumed that $x_1, \ldots, x_K$ are sampled form the same underlying distribution $p(\cdot)$. Here we provide a natural extension when the tokens are sampled are not sampled from a product distribution but instead from a joint distribution: $p(x_1, \ldots x_K)$.

**Theorem 4** *Let $P^\star(\mathrm{acc})$ be the acceptance probability for the optimal token level selection rule when $\mathcal{S} \sim p(x_1, \ldots x_K)$. Then we have*

$$P^\star(\mathrm{acc}) = \max_{\{\beta_y(x_{1:K})\}} \left\{ \sum_{y \in \Omega} \min\left( q(y), \sum_{x_1, \ldots, x_K \in \Omega} \beta_y(x_{1:K}) \cdot p(x_{1:K}) \right) \right\} \tag{39}$$

*where the maximum is over $\beta_y(x_{1:K})$ such that $0 \leq \beta_y(x_{1:K}) \leq 1$, and*

$$\sum_{x_{1:K} \in \Omega^K} \beta_y(x_{1:K}) = 1, \quad \forall y \in \Omega, \tag{40}$$

*and furthermore*

$$\beta_y(x_{1:K}) = 0, \quad y \notin \{x_1, \ldots, x_K\}. \tag{41}$$

*Furthermore if $\{\beta_y^\star(x_{1:K})\}$ denotes the parameters that achieve the maximum in (39), then $P^\star(\mathrm{acc})$ can be attained by a two step approach as follows: in the first step, given the list of input tokens $\{x_1, \ldots, x_K\}$, we apply Importance Weighted Speculative Sampling in Definition 1 with parameters $\beta_y^\star(x_1, \ldots, x_K)$ to output an intermediate token $y \in \{x_1, \ldots, x_K\}$; in the second step we apply a single-draft speculative sampling scheme (Chen et al., 2023; Leviathan et al., 2023) on the selected token $y$ to generate the final output token.*

The proof of Theorem 4 is identical to the proof of Theorem 1. We note that replacing the distribution of $\mathcal{S}$ from $\prod_{i=1}^K p(X_i)$ to the joint distribution $p(x_1, \ldots x_K)$ does not affect any of the steps as we did not use the property that the joint distribution factorizes in to the product.

## C    PROOF OF THEOREM 2

We first consider the special case of $\Omega = \{1, 2, 3\}$ to illustrate the key ideas. We then proceed with the proof.

**Example 2** *Consider the case when $\Omega = \{1, 2, 3\}$ and let $\mathbf{p} = (p_1, p_2, p_3)$ and $\mathbf{q} = (q_1, q_2, q_3)$ denote the draft and target model distribution for the current token of interest. We again assume $K = 2$ drafts. Let $X_1 = i$ and $X_2 = j$ denote the pair of input tokens and $Y$ denote the output of the importance weighted sampling scheme in step 1 in Fig. 1. Since $X_1 \sim p(\cdot)$ and $X_2 \sim p(\cdot)$ it is clear the the optimal TLSR does not depend on the order of $X_1$ and $X_2$ but only on the unordered set $\{X_1, X_2\}$ and let $\{i, j\}$ denote the realization. Let $\alpha_{i,j} = \Pr(Y = i, \{X_1, X_2\} = \{i, j\})$ denote the probability of the event that the (unordered) input tokens are $\{i, j\}$ and the output token is $Y = i$. Similarly let $\alpha_{j,i} = \Pr(Y = j, \{X_1, X_2\} = \{i, j\})$. Note that $\alpha_{i,i} = p_i^2$ must hold, as when $X_1 = X_2 = i$, clearly $Y = i$ in the Importance Weighted Sampling scheme. Note that $P^\star(acc) = 1$ requires that $\Pr(Y = i) = q_i$ for each $i \in \Omega$. This results in the following system of linear equations:*

$$q_1 = p_1^2 + \alpha_{1,2} + \alpha_{1,3}, \qquad q_2 = p_2^2 + \alpha_{2,1} + \alpha_{2,3}, \qquad q_3 = p_3^2 + \alpha_{3,1} + \alpha_{3,2} \qquad (42)$$

*subject to $\alpha_{i,j} + \alpha_{j,i} = 2p_i p_j$ and $0 \le \alpha_{i,j} \le 2p_i p_j$. We prove that (5) provides a necessary and sufficient condition that the above system of linear equations has a feasible solution.*

*Our initial attempt was to directly apply Fourer-Motzkin (FM) elimination technique Ziegler (2012); Dantzig and Curtis Eaves (1973) to (42). However a direct application of FM elimination does not appear to be tractable for arbitrary sized alphabets, as the elimination of each variable introduces a large number of inequalities. Our key observation is that (42) is equivalent to the following relaxed set of inequalities:*

$$q_1 \ge p_1^2 + \alpha_{1,2} + \alpha_{1,3}, \qquad q_2 \ge p_2^2 + \alpha_{2,1} + \alpha_{2,3}, \qquad q_3 \ge p_3^2 + \alpha_{3,1} + \alpha_{3,2} \qquad (43)$$

*with the same conditions on $\alpha_{i,j}$ as before. A solution to (42) exists if and only if a solution to the relaxation (43) exists. Indeed as a contradiction, suppose that a solution to (43) exists with strict inequality in one of conditions. Then summing over all the inequalities and using $\alpha_{i,j} + \alpha_{j,i} = 2p_i p_j$ gives $q_1 + q_2 + q_3 > (p_1 + p_2 + p_3)^2$. However since $\mathbf{p}$ and $\mathbf{q}$ are probability vectors both sides should sum to 1, leading to a contradiction. Our second key idea is to augment the system of inequalities in (43) with the following additional inequalities:*

$$q_1 + q_2 \ge (p_1 + p_2)^2 + \alpha_{1,3} + \alpha_{2,3},$$
$$q_1 + q_3 \ge (p_1 + p_3)^2 + \alpha_{1,2} + \alpha_{3,2}, \qquad q_2 + q_3 \ge (p_2 + p_3)^2 + \alpha_{2,1} + \alpha_{3,1} \qquad (44)$$

*Note that the inequalities in (44) are redundant and follow by simply adding each pair of inequalities in (43) and using $\alpha_{i,j} + \alpha_{j,i} = 2p_i p_j$. However applying FM eliminations simultaneously over the expanded system of inequalities involving (43) and (44) is surprisingly tractable. In fact we show that applying FM elimination for eliminating each $\alpha_{i,j}$ (and by extension $\alpha_{j,i}$) simply involves dropping that variable in the system of inequalities (43) and (44). For example eliminating $\alpha_{1,2}$ (and simultaneously $\alpha_{2,1}$) in the first step is equivalent to:*

$$q_1 \ge p_1^2 + \alpha_{1,3}, \ q_2 \ge p_2^2 + \alpha_{2,3}, \ q_3 \ge p_3^2 + \alpha_{3,1} + \alpha_{3,2} \qquad (45)$$
$$q_1 + q_2 \ge (p_1 + p_2)^2 + \alpha_{1,3} + \alpha_{2,3}, \ q_1 + q_3 \ge (p_1 + p_3)^2 + \alpha_{3,2}, \ q_2 + q_3 \ge (p_2 + p_3)^2 + \alpha_{3,1} \qquad (46)$$

*Eliminating all $\alpha_{i,j}$ in this fashion establishes that a feasible solution exists if and only if $q_i \ge p_i^2$ and $q_j + q_k \ge (p_j + p_k)^2$ for $i, j, k \in \Omega$ and $j \ne k$. This is precisely the condition in (5) for an alphabet of size $|\Omega| = 3$.*

We now proceed with the proof of the result.

**Setting of Linear System of Equations and its Relaxation:**    Following the simplified notation in the main text for the case of $K = 2$ drafts, we let $\mathbf{q} = (q_1, \ldots, q_n)$ be the target model distribution and $\mathbf{p} = (p_1, \ldots, p_n)$ be the draft model distribution. Also recall that we define $\alpha_{i,j} = \Pr(Y = i, \{X_1, X_2\} = \{i, j\})$ as discussed in the main text. In order to match the output distribution $\Pr(Y = i)$ to the target distribution, we need to satisfy the following system of linear equations:

$$q_1 - p_1^2 = \alpha_{1,2} + \ldots + \alpha_{1,n} \tag{47}$$

$$q_2 - p_2^2 = \alpha_{2,1} + \ldots + \alpha_{2,n} \tag{48}$$

$$\vdots \tag{49}$$

$$q_n - p_n^2 = \alpha_{n,1} + \ldots + \alpha_{n,n-1} \tag{50}$$

where $\alpha_{i,j} \geq 0$ and $\alpha_{i,j} + \alpha_{j,i} = 2p_i p_j = 2p_{i,j}$ for each $i \neq j \in \{1, \ldots, n\}$.

We instead consider a relaxed system of inequalities:

$$q_1 - p_1^2 \geq \alpha_{1,2} + \ldots + \alpha_{1,n} \tag{51}$$

$$q_2 - p_2^2 \geq \alpha_{2,1} + \ldots + \alpha_{2,n} \tag{52}$$

$$\vdots \tag{53}$$

$$q_n - p_n^2 \geq \alpha_{n,1} + \ldots + \alpha_{n,n-1} \tag{54}$$

where $\alpha_{i,j} \geq 0$ and $\alpha_{i,j} + \alpha_{j,i} = 2p_i p_j = 2p_{i,j}$ for each $i \neq j \in \{1, \ldots, n\}$. We note that the system of inequalities (47)-(50) has a solution if and only if the system of inequalities (51)-(54) has a solution. Indeed, for contradiction assume that one of the inequalities in (51)-(54) is a strict inequality. Then summing over the left and right hand sides and using $\alpha_{i,j} + \alpha_{j,i} = 2p_i p_j$ we get that

$$\sum_{i=1}^{n} q_i > \left( \sum_{i=1}^{n} p_i \right)^2, \tag{55}$$

which is a contradiction as both sides sum to 1. Thus it suffices to consider the system of linear inequalities.

**Augmented System of Inequalities:** Instead of the original system of inequalities (51)-(54), we consider an augmented system of inequalities defined as follows.

**Lemma 1** *Our original system* (51)-(54) *has a solution if an only if the following system has a solution:*

$$\sum_{s \in \mathcal{S}} q_s - \left( \sum_{s \in \mathcal{S}} p_s \right)^2 \geq \sum_{s \in \mathcal{S}} \sum_{t \in \mathcal{S}^c} \alpha_{s,t} \quad \forall \mathcal{S} \subseteq \{1, \ldots, n\} \tag{56}$$

*for $\alpha_{s,t} \geq 0$ and $\alpha_{s,t} + \alpha_{t,s} = 2p_{s,t}$ for $s, t \in \{1, 2, \ldots, n\}$ with $s \neq t$.*

To establish this, we use (51)-(54) and sum over $s \in \mathcal{S}$:

$$\sum_{s \in \mathcal{S}} \left( q_s - p_s^2 \right) \geq \sum_{s \in \mathcal{S}} \sum_{j=1, j \neq s}^{n} \alpha_{s,j} \tag{57}$$

$$= \sum_{s \in \mathcal{S}} \left( \sum_{t \in \mathcal{S}^c} \alpha_{s,t} + \sum_{t \in \mathcal{S} \setminus \{s\}} \alpha_{s,t} \right) \tag{58}$$

$$= \sum_{s \in \mathcal{S}} \sum_{t \in \mathcal{S}^c} \alpha_{s,t} + \sum_{s \in \mathcal{S}} \sum_{t \in \mathcal{S} \setminus \{s\}} \alpha_{s,t} \tag{59}$$

$$= \sum_{s \in \mathcal{S}} \sum_{t \in \mathcal{S}^c} \alpha_{s,t} + \sum_{(s,t) \in \mathcal{S} \times \mathcal{S}, t > s} (\alpha_{s,t} + \alpha_{t,s}) \tag{60}$$

$$= \sum_{s \in \mathcal{S}} \sum_{t \in \mathcal{S}^c} \alpha_{s,t} + \sum_{(s,t) \in \mathcal{S} \times \mathcal{S}, t > s} 2p_{s,t} \tag{61}$$

It follows that:

$$\sum_{s \in \mathcal{S}} q_s - \sum_{s \in \mathcal{S}} p_s^2 - \sum_{(s,t) \in \mathcal{S} \times \mathcal{S}, t > s} 2p_{s,t} \geq \sum_{s \in \mathcal{S}} \sum_{t \in \mathcal{S}^c} \alpha_{s,t} \tag{62}$$

$$\Rightarrow \sum_{s \in \mathcal{S}} q_s - \left(\sum_{s \in \mathcal{S}} p_s\right)^2 \geq \sum_{s \in \mathcal{S}} \sum_{t \in \mathcal{S}^c} \alpha_{s,t} \tag{63}$$

as required. The other inclusion follows by simply setting $\mathcal{S} = \{i\}$ for each $i$.

**Induction Argument**  We will prove the following by induction.

**Lemma 2** *Let*

$$\mathcal{V}_r = \{(i_1, j_1), (j_1, i_1), \dots, (i_r, j_r), (j_r, i_r)\} \tag{64}$$

*denote the indices (with $i_k < j_k$ for all $k = 1, \dots, r$) of the variables eliminated after $r$ rounds of FM elimination. Then the remaining constraints are given by:*

$$\sum_{s \in \mathcal{S}} q_s - \left(\sum_{s \in \mathcal{S}} p_s\right)^2 \geq \sum_{s \in \mathcal{S}} \sum_{t \in \mathcal{S}^c} \alpha_{s,t} \cdot \mathbb{I}((s,t) \notin \mathcal{V}_r), \quad \forall \mathcal{S} \subseteq \{1, \dots, n\} \tag{65}$$

**Remark 4** *When all the variables have been eliminated the right hand side in* (65) *will equal* 0 *for any choice of $\mathcal{S} \subseteq \{1, 2, \dots, n\}$ and we will recover the result Theorem 2.*

Note that the base case with $\mathcal{V}_r = \{\cdot\}$ immediately follows from (56). We will assume that the variables $\alpha_{i_q, j_q}$ and $\alpha_{j_q, i_q}$ are eliminated for $q \in \{1, \dots, r - 1\}$ and the associated Fourier-Motzkin (FM) conditions are given by:

$$\sum_{s \in \mathcal{S}} q_s - \left(\sum_{s \in \mathcal{S}} p_s\right)^2 \geq \sum_{s \in \mathcal{S}} \sum_{t \in \mathcal{S}^c} \alpha_{s,t} \cdot \mathbb{I}((s,t) \notin \mathcal{V}_{r-1}), \quad \forall \mathcal{S} \subseteq \{1, \dots, n\}. \tag{66}$$

At step $r$ we eliminate the variable $\alpha_{i_r, j_r}$ and $\alpha_{j_r, i_r}$ and we will show that (65) is satisfied. In applying the FM elimination, we only need to consider those inequalities in (66) where either $\alpha_{i_r, j_r}$ or $\alpha_{j_r, i_r}$ appears on the right hand side. The remaining equations will not be affected in this step of FM elimination and replacing $\mathcal{V}_{r-1}$ with $\mathcal{V}_r$ will not have any effect there. Any such inequality will be associated with a choice of $\mathcal{S}$ where either both $i_r$ and $j_r$ belong to $\mathcal{S}$ or neither $i_r$ and $j_r$ belong to $\mathcal{S}$. Thus we have:

$$\sum_{s \in \mathcal{S}} q_s - \left(\sum_{s \in \mathcal{S}} p_s\right)^2 \geq \sum_{s \in \mathcal{S}} \sum_{t \in \mathcal{S}^c} \alpha_{s,t} \cdot \mathbb{I}((s,t) \notin \mathcal{V}_r),$$
$$\forall \mathcal{S} \subseteq \{1, \dots, n\} : i_r \in \mathcal{S} \;\&\; j_r \in \mathcal{S} \text{ or } i_r \notin \mathcal{S} \;\&\; j_r \notin \mathcal{S}. \tag{67}$$

The FM elimination will only consider those inequalities in (66) where either $\alpha_{i_r, j_r}$ or $\alpha_{j_r, i_r}$ appears in the right hand side. The inequalities where $\alpha_{i_r, j_r}$ appears on the right hand side is associated with those subsets $\mathcal{S}_1$ of $\{1, \dots, n\}$ where $i_r \in \mathcal{S}_1$ and $j_r \notin \mathcal{S}_1$. Likewise the inequalities in (66) where $\alpha_{j_r, i_r}$ is appears on the right hand side are associated those subsets $\mathcal{S}_2 \in \{1, 2, \dots, n\}$ where $j_r \in \mathcal{S}_2$ and $i_r \notin \mathcal{S}_2$. Thus the FM elimination applied to variables $\alpha_{i_r, j_r}$ and $\alpha_{j_r, i_r}$ will consider the following system of equations:

$$\sum_{s \in \mathcal{S}_1} q_s - \left(\sum_{s \in \mathcal{S}_1} p_s\right)^2 \geq \sum_{s \in \mathcal{S}_1} \sum_{t \in \mathcal{S}_1^c} \alpha_{s,t} \cdot \mathbb{I}((s,t) \notin \mathcal{V}_{r-1}),$$
$$\forall \mathcal{S}_1 \subseteq \{1, \dots, n\}, i_r \in \mathcal{S}_1, j_r \notin \mathcal{S}_1, \tag{68}$$

$$\sum_{s \in \mathcal{S}_2} q_s - \left(\sum_{s \in \mathcal{S}_2} p_s\right)^2 \geq \sum_{s \in \mathcal{S}_2} \sum_{t \in \mathcal{S}_2^c} \alpha_{s,t} \cdot \mathbb{I}((s,t) \notin \mathcal{V}_{r-1}),$$
$$\forall \mathcal{S}_2 \subseteq \{1, \dots, n\}, j_r \in \mathcal{S}_2, i_r \notin \mathcal{S}_2, \tag{69}$$
$$\alpha_{i_r, j_r} + \alpha_{j_r, i_r} = 2p_{i_r, j_r} \tag{70}$$

$$\alpha_{i_r,j_r} \geq 0, \alpha_{j_r,i_r} \geq 0. \tag{71}$$

Accounting for (71) and using the fact that $\mathcal{V}_r = \mathcal{V}_{r-1} \cup \{(i_r, j_r), (j_r, i_r)\}$ we immediately have that:

$$\sum_{s \in \mathcal{S}_1} q_s - \left(\sum_{s \in \mathcal{S}_1} p_s\right)^2 \geq \sum_{s \in \mathcal{S}_1} \sum_{t \in \mathcal{S}_1^c} \alpha_{s,t} \cdot \mathbb{I}((s,t) \notin \mathcal{V}_r),$$
$$\forall \mathcal{S}_1 \subseteq \{1, \ldots, n\}, i_r \in \mathcal{S}_1, j_r \notin \mathcal{S}_1, \tag{72}$$

$$\sum_{s \in \mathcal{S}_2} q_s - \left(\sum_{s \in \mathcal{S}_2} p_s\right)^2 \geq \sum_{s \in \mathcal{S}_2} \sum_{t \in \mathcal{S}_2^c} \alpha_{s,t} \cdot \mathbb{I}((s,t) \notin \mathcal{V}_r),$$
$$\forall \mathcal{S}_2 \subseteq \{1, \ldots, n\}, j_r \in \mathcal{S}_2, i_r \notin \mathcal{S}_1. \tag{73}$$

In addition the FM elimination procedure is required to combine every possible inequality in (68) with every possible inequality in (69) and eliminate $\alpha_{i_r,j_r}$ and $\alpha_{j_r,i_r}$ by applying (70). For a specific choice of $\mathcal{S}_1$ and $\mathcal{S}_2$ the inequality we consider is of the form:

$$\sum_{s \in \mathcal{S}_1} q_S + \sum_{s \in \mathcal{S}_2} q_s - \left(\sum_{s \in \mathcal{S}_1} p_s\right)^2 - \left(\sum_{s \in \mathcal{S}_2} p_s\right)^2$$
$$\geq \sum_{s \in \mathcal{S}_1} \sum_{t \in \mathcal{S}_1^c} \alpha_{s,t} \cdot \mathbb{I}((s,t) \notin \mathcal{V}_{r-1}) + \sum_{s \in \mathcal{S}_2} \sum_{t \in \mathcal{S}_2^c} \alpha_{s,t} \cdot \mathbb{I}((s,t) \notin \mathcal{V}_{r-1}). \tag{74}$$

We will show that this inequality is redundant as it is dominated by the set of inequalities in (67). Let $\mathcal{R} = \mathcal{S}_1 \cap \mathcal{S}_2$ and $\mathcal{T} = \mathcal{S}_1 \cup \mathcal{S}_2$. Note that $i_r \notin \mathcal{R}$ and $j_r \notin \mathcal{R}$. Now consider the left hand side of (74).

$$\sum_{s \in \mathcal{S}_1} q_s + \sum_{s \in \mathcal{S}_2} q_s - \left(\sum_{s \in \mathcal{S}_1} p_s\right)^2 - \left(\sum_{s \in \mathcal{S}_2} p_s\right)^2$$

$$= \sum_{s \in \mathcal{S}_1 \backslash \mathcal{R}} q_s + \sum_{s \in \mathcal{S}_2 \backslash \mathcal{R}} q_s + 2\sum_{s \in \mathcal{R}} q_s - \left(\sum_{s \in \mathcal{S}_1 \backslash \mathcal{R}} p_s + \sum_{s \in \mathcal{R}} p_s\right)^2 - \left(\sum_{s \in \mathcal{S}_2 \backslash \mathcal{R}} p_s + \sum_{s \in \mathcal{R}} p_s\right)^2 \tag{75}$$

$$= \sum_{s \in \mathcal{T}} q_s + \sum_{s \in \mathcal{R}} q_s - \left(\sum_{s \in \mathcal{S}_1 \backslash \mathcal{R}} p_s\right)^2 - \left(\sum_{s \in \mathcal{R}} p_s\right)^2 - 2\left(\sum_{s \in \mathcal{S}_1 \backslash \mathcal{R}} p_s\right)\left(\sum_{s \in \mathcal{R}} q_s\right)$$
$$- \left(\sum_{s \in \mathcal{S}_2 \backslash \mathcal{R}} p_s\right)^2 - \left(\sum_{s \in \mathcal{R}} p_s\right)^2 - 2\left(\sum_{s \in \mathcal{S}_2 \backslash \mathcal{R}} p_s\right)\left(\sum_{s \in \mathcal{R}} p_s\right) \tag{76}$$

$$= \sum_{s \in \mathcal{T}} q_s + \sum_{s \in \mathcal{R}} q_s - \left(\sum_{s \in \mathcal{S}_1 \backslash \mathcal{R}} p_s\right)^2 - \left(\sum_{s \in \mathcal{R}} p_s\right)^2 - \left(\sum_{s \in \mathcal{S}_2 \backslash \mathcal{R}} p_s\right)^2 - 2\left(\sum_{s \in \mathcal{S}_1 \backslash \mathcal{R}} p_s\right)\left(\sum_{s \in \mathcal{R}} q_s\right)$$
$$- 2\left(\sum_{s \in \mathcal{S}_2 \backslash \mathcal{R}} p_s\right)\left(\sum_{s \in \mathcal{R}} p_s\right) - 2\left(\sum_{s \in \mathcal{S}_2 \backslash \mathcal{R}} p_s\right)\left(\sum_{s \in \mathcal{S}_1 \backslash \mathcal{R}} p_s\right) - 2\left(\sum_{s \in \mathcal{R}} p_s\right)^2$$
$$+ 2\left(\sum_{s \in \mathcal{S}_2 \backslash \mathcal{R}} p_s\right)\left(\sum_{s \in \mathcal{S}_1 \backslash \mathcal{R}} p_s\right) \tag{77}$$

$$= \left\{\sum_{s \in \mathcal{T}} q_s - \left(\sum_{s \in \mathcal{T}} p_s\right)^2\right\} + \left\{\sum_{s \in \mathcal{R}} q_s - \left(\sum_{s \in \mathcal{R}} p_s\right)^2\right\} + 2\left(\sum_{s \in \mathcal{S}_2 \backslash \mathcal{R}} p_s\right)\left(\sum_{s \in \mathcal{S}_1 \backslash \mathcal{R}} p_s\right) \tag{78}$$

We now consider the right hand side of (74). We recall that with $\mathcal{T} = \mathcal{S}_1 \cup \mathcal{S}_2$ and $\mathcal{R} = \mathcal{S}_1 \cap \mathcal{S}_2$ the following relations that can be easily established using Venn diagram of sets $\mathcal{S}_1$ and $\mathcal{S}_2$:

$$\mathcal{S}_1^c = \mathcal{T}^c \cup (\mathcal{S}_2 \setminus \mathcal{R}), \qquad \mathcal{T}^c \cap (\mathcal{S}_2 \setminus \mathcal{R}) = \{\cdot\} \tag{79}$$

$$\mathcal{S}_2^c = \mathcal{T}^c \cup (\mathcal{S}_1 \setminus \mathcal{R}), \qquad \mathcal{T}^c \cap (\mathcal{S}_1 \setminus \mathcal{R}) = \{\cdot\} \tag{80}$$

$$\mathcal{R}^c = \mathcal{T}^c \cup (\mathcal{S}_1 \setminus R) \cup (\mathcal{S}_2 \setminus R), \qquad (\mathcal{S}_1 \setminus R) \cap (\mathcal{S}_2 \setminus R) = \{\cdot\} \tag{81}$$

$$\mathcal{T} = \mathcal{R} \cup (\mathcal{S}_1 \setminus R) \cup (\mathcal{S}_2 \setminus R) \tag{82}$$

Now consider the following:

$$\sum_{s \in \mathcal{S}_1} \sum_{t \in \mathcal{S}_1^c} \alpha_{s,t} \cdot \mathbb{I}((s,t) \notin \mathcal{V}_{r-1})$$

$$= \sum_{s \in \mathcal{S}_1} \sum_{t \in \mathcal{T}^c} \alpha_{s,t} \cdot \mathbb{I}((s,t) \notin \mathcal{V}_{r-1}) + \sum_{s \in \mathcal{S}_1} \sum_{t \in \mathcal{S}_2 \setminus \mathcal{R}} \alpha_{s,t} \cdot \mathbb{I}((s,t) \notin \mathcal{V}_{r-1}) \tag{83}$$

$$= \sum_{s \in \mathcal{S}_1 \setminus \mathcal{R}} \sum_{t \in \mathcal{T}^c} \alpha_{s,t} \cdot \mathbb{I}((s,t) \notin \mathcal{V}_{r-1}) + \sum_{s \in \mathcal{R}} \sum_{t \in \mathcal{T}^c} \alpha_{s,t} \cdot \mathbb{I}((s,t) \notin \mathcal{V}_{r-1})$$

$$+ \sum_{s \in \mathcal{S}_1 \setminus \mathcal{R}} \sum_{t \in \mathcal{S}_2 \setminus \mathcal{R}} \alpha_{s,t} \cdot \mathbb{I}((s,t) \notin \mathcal{V}_{r-1}) + \sum_{s \in \mathcal{R}} \sum_{t \in \mathcal{S}_2 \setminus \mathcal{R}} \alpha_{s,t} \cdot \mathbb{I}((s,t) \notin \mathcal{V}_{r-1}) \tag{84}$$

where we use (79) in (83),

In a similar fashion we can express,

$$\sum_{s \in \mathcal{S}_2} \sum_{t \in \mathcal{S}_2^c} \alpha_{s,t} \cdot \mathbb{I}((s,t) \notin \mathcal{V}_{r-1})$$

$$= \sum_{s \in \mathcal{S}_2 \setminus \mathcal{R}} \sum_{t \in \mathcal{T}^c} \alpha_{s,t} \cdot \mathbb{I}((s,t) \notin \mathcal{V}_{r-1}) + \sum_{s \in \mathcal{R}} \sum_{t \in \mathcal{T}^c} \alpha_{s,t} \cdot \mathbb{I}((s,t) \notin \mathcal{V}_{r-1})$$

$$+ \sum_{s \in \mathcal{S}_2 \setminus \mathcal{R}} \sum_{t \in \mathcal{S}_1 \setminus \mathcal{R}} \alpha_{s,t} \cdot \mathbb{I}((s,t) \notin \mathcal{V}_{r-1}) + \sum_{s \in \mathcal{R}} \sum_{t \in \mathcal{S}_1 \setminus \mathcal{R}} \alpha_{s,t} \cdot \mathbb{I}((s,t) \notin \mathcal{V}_{r-1}) \tag{85}$$

Combing (84) and (85) and re-arranging terms, we get that:

$$\sum_{s \in \mathcal{S}_1} \sum_{t \in \mathcal{S}_1^c} \alpha_{s,t} \cdot \mathbb{I}((s,t) \notin \mathcal{V}_{r-1}) + \sum_{s \in \mathcal{S}_1} \sum_{t \in \mathcal{S}_1^c} \alpha_{s,t} \cdot \mathbb{I}((s,t) \notin \mathcal{V}_{r-1})$$

$$= \sum_{s \in \mathcal{S}_1 \setminus \mathcal{R}} \sum_{t \in \mathcal{T}^c} \alpha_{s,t} \cdot \mathbb{I}((s,t) \notin \mathcal{V}_{r-1}) + \sum_{s \in \mathcal{R}} \sum_{t \in \mathcal{T}^c} \alpha_{s,t} \cdot \mathbb{I}((s,t) \notin \mathcal{V}_{r-1}) + \sum_{s \in \mathcal{S}_2 \setminus \mathcal{R}} \sum_{t \in \mathcal{T}^c} \alpha_{s,t} \cdot \mathbb{I}((s,t) \notin \mathcal{V}_{r-1})$$

$$+ \sum_{s \in \mathcal{R}} \sum_{t \in \mathcal{S}_1 \setminus \mathcal{R}} \alpha_{s,t} + \sum_{s \in \mathcal{R}} \sum_{t \in \mathcal{S}_2 \setminus \mathcal{R}} \alpha_{s,t} \cdot \mathbb{I}((s,t) \notin \mathcal{V}_{r-1}) + \sum_{s \in \mathcal{R}} \sum_{t \in \mathcal{T}^c} \alpha_{s,t} \cdot \mathbb{I}((s,t) \notin \mathcal{V}_{r-1})$$

$$+ \sum_{s \in \mathcal{S}_1 \setminus \mathcal{R}} \sum_{t \in \mathcal{S}_2 \setminus \mathcal{R}} \alpha_{s,t} \cdot \mathbb{I}((s,t) \notin \mathcal{V}_{r-1}) + \sum_{s \in \mathcal{S}_2 \setminus \mathcal{R}} \sum_{t \in \mathcal{S}_1 \setminus \mathcal{R}} \alpha_{s,t} \cdot \mathbb{I}((s,t) \notin \mathcal{V}_{r-1}) \tag{86}$$

$$= \sum_{s \in \mathcal{T}} \sum_{t \in \mathcal{T}^c} \alpha_{s,t} \cdot \mathbb{I}((s,t) \notin \mathcal{V}_{r-1}) + \sum_{s \in \mathcal{R}} \sum_{t \in \mathcal{R}^c} \alpha_{s,t} \cdot \mathbb{I}((s,t) \notin \mathcal{V}_{r-1})$$

$$+ \sum_{s \in \mathcal{S}_1 \setminus \mathcal{R}} \sum_{t \in \mathcal{S}_2 \setminus \mathcal{R}} \alpha_{s,t} \cdot \mathbb{I}((s,t) \notin \mathcal{V}_{r-1}) + \sum_{s \in \mathcal{S}_1 \setminus \mathcal{R}} \sum_{t \in \mathcal{S}_2 \setminus \mathcal{R}} \alpha_{t,s} \cdot \mathbb{I}((s,t) \notin \mathcal{V}_{r-1}) \tag{87}$$

$$= \sum_{s \in \mathcal{T}} \sum_{t \in \mathcal{T}^c} \alpha_{s,t} \cdot \mathbb{I}((s,t) \notin \mathcal{V}_r) + \sum_{s \in \mathcal{R}} \sum_{t \in \mathcal{R}^c} \alpha_{s,t} \cdot \mathbb{I}((s,t) \notin \mathcal{V}_r)$$

$$+ \sum_{s \in \mathcal{S}_1 \setminus \mathcal{R}} \sum_{t \in \mathcal{S}_2 \setminus \mathcal{R}} (\alpha_{t,s} + \alpha_{s,t}) \cdot \mathbb{I}((s,t) \notin \mathcal{V}_{r-1}) \tag{88}$$

where we use (81) and (82) in (87) as well as the fact that $\mathbb{I}((s,t) \notin \mathcal{V}_{r-1}) = \mathbb{I}((t,s) \notin \mathcal{V}_{r-1})$ as the pair $(s,t)$ and $(t,s)$ is eliminated simultaneously. In (88) we use the fact that $\mathcal{T}$ contains both $i_r$ and

$j_r$ while $\mathcal{R}$ contains neither $i_{r_1}$ and $j_{r-1}$ and hence $\alpha_{i_r,j_r}$ or $\alpha_{j_r,i_r}$ do not appear in the first two terms in (88) so that $\mathcal{V}_{r-1}$ can be replaced by $\mathcal{V}_r$. Combining (78) and (88) it follows that the FM elimination for our choice of $\mathcal{S}_1$ and $\mathcal{S}_2$ leads to:

$$
\left\{ \sum_{s \in \mathcal{T}} q_s - \left( \sum_{s \in \mathcal{T}} p_s \right)^2 \right\} + \left\{ \sum_{s \in \mathcal{R}} q_s - \left( \sum_{s \in \mathcal{R}} p_s \right)^2 \right\} + 2 \left( \sum_{s \in \mathcal{S}_2 \backslash \mathcal{R}} p_s \right) \left( \sum_{s \in \mathcal{S}_1 \backslash \mathcal{R}} p_s \right)
$$
$$
\geq \sum_{s \in \mathcal{T}} \sum_{t \in \mathcal{T}^c} \alpha_{s,t} \cdot \mathbb{I}((s,t) \notin \mathcal{V}_r) + \sum_{s \in \mathcal{R}} \sum_{t \in \mathcal{R}^c} \alpha_{s,t} \cdot \mathbb{I}((s,t) \notin \mathcal{V}_r)
$$
$$
+ \sum_{s \in \mathcal{S}_1 \backslash \mathcal{R}} \sum_{t \in \mathcal{S}_2 \backslash \mathcal{R}} (\alpha_{t,s} + \alpha_{s,t}) \cdot \mathbb{I}((s,t) \notin \mathcal{V}_{r-1}) \tag{89}
$$

Note that this condition is equivalent to:

$$
\left\{ \sum_{s \in \mathcal{T}} q_s - \left( \sum_{s \in \mathcal{T}} p_s \right)^2 - \sum_{s \in \mathcal{T}} \sum_{t \in \mathcal{T}^c} \alpha_{s,t} \cdot \mathbb{I}((s,t) \notin \mathcal{V}_r) \right\}
$$
$$
+ \left\{ \sum_{s \in \mathcal{R}} q_s - \left( \sum_{s \in \mathcal{R}} p_s \right)^2 - \sum_{s \in \mathcal{R}} \sum_{t \in \mathcal{R}^c} \alpha_{s,t} \cdot \mathbb{I}((s,t) \notin \mathcal{V}_r) \right\}
$$
$$
+ \left\{ 2 \left( \sum_{s \in \mathcal{S}_2 \backslash \mathcal{R}} p_s \right) \left( \sum_{s \in \mathcal{S}_1 \backslash \mathcal{R}} p_s \right) - \sum_{s \in \mathcal{S}_1 \backslash \mathcal{R}} \sum_{t \in \mathcal{S}_2 \backslash \mathcal{R}} (\alpha_{t,s} + \alpha_{s,t}) \cdot \mathbb{I}((s,t) \notin \mathcal{V}_{r-1}) \right\} \geq 0. \tag{90}
$$

We now show that this condition is redundant as it is implied by other conditions. Since $\mathcal{T}$ and $\mathcal{R}$ satisfy the conditions in (67) we already have that:

$$
\sum_{s \in \mathcal{T}} q_s - \left( \sum_{s \in \mathcal{T}} p_s \right)^2 \geq \sum_{s \in \mathcal{T}} \sum_{t \in \mathcal{T}^c} \alpha_{s,t} \cdot \mathbb{I}((s,t) \notin \mathcal{V}_r) \tag{91}
$$

$$
\sum_{s \in \mathcal{R}} q_s - \left( \sum_{s \in \mathcal{R}} p_s \right)^2 \geq \sum_{s \in \mathcal{R}} \sum_{t \in \mathcal{R}^c} \alpha_{s,t} \cdot \mathbb{I}((s,t) \notin \mathcal{V}_r) \tag{92}
$$

Further since the sets $(\mathcal{S}_1 \backslash \mathcal{R})$ and $\mathcal{S}_2 \backslash \mathcal{R}$ are disjoint it follows that:

$$
2 \left( \sum_{t \in \mathcal{S}_2 \backslash \mathcal{R}} p_t \right) \left( \sum_{s \in \mathcal{S}_1 \backslash \mathcal{R}} p_s \right) - \sum_{s \in \mathcal{S}_1 \backslash \mathcal{R}} \sum_{t \in \mathcal{S}_2 \backslash \mathcal{R}} (\alpha_{t,s} + \alpha_{s,t}) \cdot \mathbb{I}((s,t) \notin \mathcal{V}_{r-1}) \tag{93}
$$
$$
= \sum_{t \in \mathcal{S}_2 \backslash \mathcal{R}} \sum_{s \in \mathcal{S}_1 \backslash \mathcal{R}} 2 p_s p_t - \sum_{s \in \mathcal{S}_1 \backslash \mathcal{R}} \sum_{t \in \mathcal{S}_2 \backslash \mathcal{R}} (\alpha_{t,s} + \alpha_{s,t}) \cdot \mathbb{I}((s,t) \notin \mathcal{V}_{r-1}) \tag{94}
$$
$$
= \sum_{t \in \mathcal{S}_2 \backslash \mathcal{R}} \sum_{s \in \mathcal{S}_1 \backslash \mathcal{R}} (2 p_s p_t - (\alpha_{t,s} + \alpha_{s,t}) \cdot \mathbb{I}((s,t) \notin \mathcal{V}_{r-1}) \geq 0 \tag{95}
$$

where we use the fact that by construction $\alpha_{s,t} + \alpha_{t,s} = 2 p_s p_t$. It thus follows that the condition (90) is implied by other conditions already presented in the FM elimination and is thus redundant. Since our choice $\mathcal{S}_1$ and $\mathcal{S}_2$ is arbitrary it follows that every combination of the form (74) is redundant and the only equations that remain upon elimination of $\alpha_{i_r,j_r}$ and $\alpha_{j_r,i_r}$ are given by (67), (72) and (73). This completes the induction step in Lemma 2 and the proof.

# D  CONNECTION BETWEEN THEOREM 2 AND POLYHEDRAL CONE REPRESENTATION

We consider the case of $\Omega = \{1,2,3\}$ for sake of concreteness. We discuss how the characterization of $P^\star(\mathrm{acc}) = 1$ is related to dual representation of a polyhedral cone. Let $\mathbf{p} = (p_1, p_2, p_3)$ denote

the draft probability and $\mathbf{q} = (q_1, q_2, q_3)$ denote the target probability vector. As before we define $\alpha_{i,j} = \Pr(Y = i, \{X_1, X_2\} = \{i, j\})$. We need to solve the following system of equations:

$$q_1 - p_1^2 = \alpha_{1,2} + \alpha_{1,3} \tag{96}$$

$$q_2 - p_2^2 = \alpha_{2,1} + \alpha_{2,3} \tag{97}$$

$$q_3 - p_3^2 = \alpha_{3,1} + \alpha_{3,2} \tag{98}$$

subject to the conditions that $\alpha_{i,j} + \alpha_{j,i} = 2p_i p_j$ and $0 \leq \alpha_{i,j} \leq 2p_i p_j$. Using the fact that $q_1 + q_2 + q_3 = 1$ and $p_1 + p_2 + p_3 = 1$, it suffices the consider the following system of equations:

$$\alpha_{1,2} + \alpha_{1,3} = q_1 - p_1^2 \tag{99}$$

$$\alpha_{2,1} + \alpha_{2,3} = q_2 - p_2^2 \tag{100}$$

$$\alpha_{1,2} + \alpha_{2,1} = 2p_{1,2} \tag{101}$$

$$\alpha_{1,3} + \alpha_{3,1} = 2p_{1,3} \tag{102}$$

$$\alpha_{2,3} + \alpha_{3,2} = 2p_{2,3} \tag{103}$$

with the additional requirement that $\alpha_{i,j} \geq 0$. We will represent this system of equations in matrix form. Our variables of interest are $\mathbf{x} = [\alpha_{1,2}, \alpha_{1,3}, \alpha_{2,1}, \alpha_{2,3}, \alpha_{3,1}, \alpha_{3,2}]^T \geq 0$. Our equality constraints can be expressed in the following form:

$$\mathbf{A} \cdot \mathbf{x} = \mathbf{b}, \qquad \mathbf{x} \geq 0 \tag{104}$$

where

$$\mathbf{A} = \begin{bmatrix} 1, 1, 0, 0, 0, 0 \\ 0, 0, 1, 1, 0, 0 \\ 1, 0, 1, 0, 0, 0 \\ 0, 1, 0, 0, 1, 0 \\ 0, 0, 0, 1, 0, 1 \end{bmatrix}, \qquad \mathbf{x} = \begin{bmatrix} \alpha_{1,2} \\ \alpha_{1,3} \\ \alpha_{2,1} \\ \alpha_{2,3} \\ \alpha_{3,1} \\ \alpha_{3,2} \end{bmatrix}, \qquad \mathbf{b} = \begin{bmatrix} q_1 - p_1^2 \\ q_2 - p_2^2 \\ 2p_{1,2} \\ 2p_{1,3} \\ 2p_{2,3} \end{bmatrix} \tag{105}$$

Upon application of Farakas' Lemma it follows that the system (104) has a solution if and only if every $\mathbf{y}$ that satisfies $\mathbf{y}^T \mathbf{A} \geq 0$ also satisfies $\mathbf{y}^T \mathbf{b} \geq 0$, where $\mathbf{b}$ depends on $\mathbf{p}$ and $\mathbf{q}$ as in (105). Let us define

$$\mathbf{B} = \mathbf{A}^T = \begin{bmatrix} 1, 0, 1, 0, 0 \\ 1, 0, 0, 1, 0 \\ 0, 1, 1, 0, 0 \\ 0, 1, 0, 0, 1 \\ 0, 0, 0, 1, 0 \\ 0, 0, 0, 0, 1 \end{bmatrix} \tag{106}$$

and note that the set

$$\mathcal{B} = \{\mathbf{y} : \mathbf{By} \geq 0\} \tag{107}$$

denotes a polyhedral cone in $\mathbb{R}^5$. We need to show that for each $\mathbf{y} \in \mathcal{B}$ we must have that $\mathbf{y}^T \mathbf{b} \geq 0$. The representation (107) is the so-called hyperplane representation of the code as each row of $\mathbf{B}$ defines a hyperplane. We would like to find an equivalent generator representation of the form:

$$\mathcal{R} = \{\mathbf{z} : \mathbf{z} = \mathbf{R}\lambda, \lambda \geq 0\} \tag{108}$$

The Minikowski-Weyl Theorem (Fukuda and Prodon, 1995) guarantees that for every $\mathbf{B}$ in (107) there exists a $\mathbf{R}$ in (108) of finite dimensions such that $\mathcal{B} = \mathcal{R}$. Furthermore the double-description method is an algorithmic way of computing $\mathbf{R}$ given $\mathbf{B}$ and vice versa. Using the package *skeleton* for double description (Zny, 2018) we could show that for the $\mathbf{B}$ matrix in (106) the associated $R$ matrix is given by:

$$\mathbf{R}^T = \begin{bmatrix} & & \mathbf{I}_5 & & \\ 1 & 1 & -1 & 0 & 0 \\ -1 & 0 & 1 & 1 & 0 \\ 0 & -1 & 1 & 0 & 1 \\ -1 & -1 & 1 & 1 & 1 \end{bmatrix} \tag{109}$$

where $\mathbf{I}_5$ is a $5 \times 5$ identity matrix. The generator representation in (108) is convenient as in order to show that (104) has a feasible solution, it suffices to show that $\mathbf{R}^T\mathbf{b} \geq 0$. Indeed substitution of (109) and (105) yields

$$\mathbf{R}^T\mathbf{b} = \begin{bmatrix} \mathbf{b} \\ q_1 + q_2 - (p_1 + p_2)^2 \\ -q_1 + p_1^2 + 2p_{1,2} + 2p_{1,3} \\ -q_2 + p_2^2 + 2p_{1,2} + 2p_{2,3} \\ -q_1 - q_2 + p_1^2 + p_2^2 + 2p_{1,2} + 2p_{1,3} + 2p_{2,3} \end{bmatrix} = \begin{bmatrix} \mathbf{b} \\ q_1 + q_2 - (p_1 + p_2)^2 \\ q_2 + q_3 - (p_2 + p_3)^2 \\ q_1 + q_3 + (p_1 + p_3)^2 \\ q_3 - p_3^2 \end{bmatrix} \quad (110)$$

In the last step we use the fact that $\sum q_i = \sum p_i = 1$. It thus follows that $\mathbf{R}^T\mathbf{b} \geq 0$ if and only if $q_i \geq p_i^2$ and $q_i + q_j \geq (p_i + p_j)^2$ holds as stated in Theorem 2. Thus this approach provides an alternative proof for Theorem 2 for the case of $|\Omega| = 3$. We did not however find a simple approach to analytically compute the generator representation $\mathcal{R}$ from the hyperplane representation $\mathcal{B}$ for arbitrary dimensions. On the other hand we used the numerical implementation of the double description method to compute $\mathbf{B}$ and $\mathbf{R}$ for the case of up-to $K = 6$ drafts and $|\Omega| \leq 14$ and demonstrate that the natural counterpart of our result in Theorem 2 appears to be valid in all these cases.

## D.1 INTUITION FOR THE CASE WHEN $K = 3$ AND $|\Omega| = 3$

We consider the case of $\Omega = \{1, 2, 3\}$ and $K = 3$ drafts. We let $\mathbf{q} = (q_1, q_2, q_3)$ and $\mathbf{p} = (p_1, p_2, p_3)$ denote the probabilities of the target and draft models. We build upon the ideas in Example 2 in Section C and provide some intuition that the necessary and sufficient condition for the acceptance probability to equal 1 is:

$$q_i > p_i^3, \quad q_i + q_j \geq (p_i + p_j)^3, \qquad \forall i, j \in \{1, 2, 3\}, i \neq j.$$

We let $\alpha_{1,1,2}(1) = \Pr(Y = 1, \{X_1, X_2, X_3\} = \{1, 1, 2\})$ i.e., the probability that the selected token equals 1 and the input tokens are $\{1, 1, 2\}$ in any order. Likewise we let $\alpha_{1,1,2}(2) = \Pr(Y = 2, \{X_1, X_2, X_3\} = \{1, 1, 2\})$. Note that $\alpha_{1,1,2}(1) + \alpha_{1,1,2}(2) = 3p_1^2p_2$ is the probability of observing the input tokens to equal $\{1, 1, 2\}$ in some order. We define $\alpha_{i,i,j}(i)$ and $\alpha_{i,i,j}(j)$ for other values of $i, j$ in a similar fashion. Finally we let $\alpha_{1,2,3}(i) = \Pr(Y = i, \{X_1, X_2, X_3\} = \{1, 2, 3\})$ for each $i \in \{1, 2, 3\}$ be the probability that token $i$ is selected and the input tokens are $\{1, 2, 3\}$ in some order. We observe that $\sum_i \alpha_{1,2,3}(i) = 6p_1p_2p_3$. We note that the probability that the selected token $Y = 1$ is given by:

$$p_I(1) = p_1^3 + \alpha_{1,1,2}(1) + \alpha_{1,2,2}(1) + \alpha_{1,1,3}(1) + \alpha_{1,3,3}(1) + \alpha_{1,2,3}(1) \quad (111)$$

Here the right hand side denotes all the events that a token 1 appears among the in put tokens and that token 1 is selected. Following the same relaxation as in Example 2 we can show that the acceptance probability equals 1 if

$$q_1 \geq p_I(1) = p_1^3 + \alpha_{1,1,2}(1) + \alpha_{1,2,2}(1) + \alpha_{1,1,3}(1) + \alpha_{1,3,3}(1) + \alpha_{1,2,3}(1) \quad (112)$$

$$q_2 \geq p_I(2) = p_2^3 + \alpha_{1,1,2}(2) + \alpha_{1,2,2}(2) + \alpha_{2,2,3}(2) + \alpha_{2,3,3}(2) + \alpha_{1,2,3}(2) \quad (113)$$

$$q_3 \geq p_I(3) = p_3^3 + \alpha_{1,1,3}(3) + \alpha_{1,3,3}(3) + \alpha_{2,2,3}(3) + \alpha_{2,3,3}(3) + \alpha_{1,2,3}(3) \quad (114)$$

Following Example 2 we also consider the following augmented system of inequalities:

$$q_1 + q_2 \geq (p_1 + p_2)^3 + \alpha_{1,1,3}(1) + \alpha_{1,3,3}(1) + \alpha_{1,2,3}(1) + \alpha_{2,2,3}(2) + \alpha_{2,3,3}(2) + \alpha_{1,2,3}(2)$$
$$(115)$$

$$q_1 + q_3 \geq (p_1 + p_3)^3 + \alpha_{1,1,3}(1) + \alpha_{1,3,3}(1) + \alpha_{1,2,3}(1) + \alpha_{2,2,3}(3) + \alpha_{2,3,3}(3) + \alpha_{1,2,3}(3)$$
$$(116)$$

$$q_2 + q_3 \geq (p_2 + p_3)^3 + \alpha_{1,1,2}(2) + \alpha_{1,2,2}(2) + \alpha_{1,2,3}(2) + \alpha_{1,1,3}(3) + \alpha_{1,3,3} + \alpha_{1,2,3}(3) \quad (117)$$

We note that (115) is a direct consequence of adding (112) and (113) and using the condition that $\alpha_{i,i,j}(i) + \alpha_{i,i,j}(j) = 3p_i^2p_j$. The inequalities (116) and (117) follow in a similar manner. Although these inequalities are redundant their presence aids in simplifying the Fourier-Motzkin elimination as in the case of $K = 2$ drafts. We do not considering summing all the inequalities in (112)-(114) as this implies $q_1 + q_2 + q_3 \geq (p_1 + p_2 + p_3)^3$, which holds trivially as both sides are equal to 1. In

order to find conditions when the probability equals 1, we need to apply Fourier-Motzkin elimination to the system of inequalities in (112)-(117), where the variables satisfy the conditions that

$$\alpha_{i,i,j}(i) + \alpha_{i,i,j}(j) = 3p_i^2 p_j \tag{118}$$

and

$$\alpha_{1,2,3}(1) + \alpha_{1,2,3}(2) + \alpha_{1,2,3}(3) = 6p_1 p_2 p_3. \tag{119}$$

and all the variables are non-negative. By following elimination procedure we can show that, similar to Example 2 and Lemma 2, the elimination of each variable in this augmented system simply amounts to dropping it from the right hand side in the system of inequalities in (112)-(117). Thus the intuition behind our conjecture is that when one considers an augmented system of inequalities, only terms of the form $\left( \sum_{s \in \mathcal{S}} p_s \right)^K$ appear and lower order terms do not appear. After application of the Fourier-Motzkin elimination the required condition follows.

## E    PROOF OF THEOREM 3

As in the proof of Theorem 2, we let $\mathbf{p} = [p_1, \ldots, p_n]$ be the distribution of the draft model and $\mathbf{q} = [q_1, \ldots, q_n]$ be the distribution of the target model. Our optimization problem can be expressed as follows:

$$\text{maximize} \sum_{i=1}^{n} t_i, \tag{120}$$

$$t_i \leq \min \left( q_i, p_i^2 + \sum_{j \neq i} \alpha_{i,j} \right), \tag{121}$$

$$\alpha_{i,j} + \alpha_{j,i} = 2p_i p_j, 0 \leq \alpha_{i,j} \leq 1. \tag{122}$$

In order to solve this linear program analytically we introduce an additional variable $z$ satisfying a single inequality $z \leq t_1 + \ldots t_n$. We provide the range of feasible feasible values of $z$ and pick the maximum. Following the techniques used in the proof of Theorem 2 we have the following Lemma:

**Lemma 3** *Upon applying Fourier-Motzkin elimination technique to eliminate variables $\alpha_{i,j}$ in (120)-(122), we have the following system of inequalities with $\Omega = \{1, \ldots, n\}$:*

$$t_i \leq q_i, i \in \Omega \tag{123}$$

$$\sum_{i \in \mathcal{S}} t_i \leq \left( \sum_{i \in \mathcal{S}} p_i \right)^2 + 2 \left( \sum_{i \in \mathcal{S}} p_i \right) \left( \sum_{i \in \mathcal{S}^c} p_i \right), \quad \forall \mathcal{S} \subseteq \Omega, \mathcal{S}^c = \Omega \setminus \mathcal{S} \tag{124}$$

$$z \leq \sum_{i=1}^{n} t_i. \tag{125}$$

We will defer the proof of this lemma after the main proof. We will use (123)-(125) to establish the following step by induction.

**Lemma 4** *Suppose that we apply Fourier Motzkin elimination to eliminate variables $t_1, \ldots, t_{j-1}$ in (123)-(125). Let $\Omega_1 = \{1, \ldots, j-1\}$ and $\Omega_2 = \{j, \ldots, n\}$ be partition of $\Omega$. Then we have*

$$z \leq \sum_{i \in \mathcal{S}} q_i + \sum_{i \in \mathcal{V}} t_i + \left( \sum_{i \in \mathcal{S}^c \cup \mathcal{V}^c} p_i \right)^2 + 2 \left( \sum_{i \in \mathcal{S} \cup \mathcal{V}} p_i \right) \left( \sum_{i \in \mathcal{S}^c \cup \mathcal{V}^c} p_i \right),$$
$$\forall \mathcal{S} \subseteq \Omega_1, \mathcal{V} \subseteq \Omega_2, \mathcal{S}^c = \Omega_1 \setminus \mathcal{S}, \mathcal{V}^c = \Omega_2 \setminus \mathcal{V} \tag{126}$$

$$\sum_{i \in \mathcal{S}} t_i \leq \left( \sum_{i \in \mathcal{S}} p_i \right)^2 + 2 \left( \sum_{i \in \mathcal{S}} p_i \right) \left( \sum_{i \in \mathcal{S}^c} p_i \right), \quad \forall \mathcal{S} \subseteq \Omega_2, \quad \mathcal{S}^c = \Omega_2 \setminus \mathcal{S} \tag{127}$$

$$t_i \leq q_i, \qquad \forall i \in \Omega_2 \tag{128}$$

Note that this results implies the main result as by setting $\Omega_1 = \Omega$ and $\Omega_2 = \{\cdot\}$ we have:

$$z \leq \sum_{i \in \mathcal{S}} q_i + \left( \sum_{i \in \mathcal{S}^c} p_i \right)^2 + 2 \left( \sum_{i \in \mathcal{S}} p_i \right) \left( \sum_{i \in \mathcal{S}^c} p_i \right) \tag{129}$$

We first consider the base case: $j = 1$. In this case $\Omega_1 = \{\cdot\}$ is the empty set and $\Omega_2 = \Omega$. Thus $\mathcal{S} = \mathcal{S}^c = \{\cdot\}$ and $\mathcal{V} \subseteq \Omega$ and $\mathcal{V}^c = \Omega \setminus \mathcal{V}$. In this case (126) reduces to:

$$z \leq \sum_{i \in \mathcal{V}} t_i + \left( \sum_{i \in \mathcal{V}^c} p_i \right)^2 + 2 \left( \sum_{i \in \mathcal{V}} p_i \right) \left( \sum_{i \in \mathcal{V}^c} p_i \right) \tag{130}$$

and (127) and (128) have $\Omega_2 = \Omega$. Note that (127) and (128) are equivalent to (123) and (124). It thus suffices to show the equivalence between (130) and (125). To show that the condition (130) implies (125) it suffices to set $\mathcal{V} = \Omega$ and $\mathcal{V}^c = \{\cdot\}$. To show that (125) implies (130), for any $\mathcal{V} \subseteq \Omega$, we can express:

$$z \leq \sum_{i \in \mathcal{V}} t_i + \sum_{i \in \mathcal{V}^c} t_i \tag{131}$$

$$\leq \sum_{i \in \mathcal{V}} t_i + \left( \sum_{i \in \mathcal{V}^c} p_i \right)^2 + 2 \left( \sum_{i \in \mathcal{V}} p_i \right) \left( \sum_{i \in \mathcal{V}^c} p_i \right) \tag{132}$$

where we use (124) in the second term. This completes the proof of the base case.

For induction we assume that for some $j > 0$ the application of Fourier-Motzkin elimination on eliminate $t_1, \ldots, t_{j-1}$ leads to (126)-(128) with $\Omega_1 = \{1, \ldots, j-1\}$ and $\Omega_2 = \{j, \ldots, n\}$. We want to show that upon applying Fourier-Motzkin elimination to eliminate $t_j$, we reduce the system of inequalities again to (126)-(128) with $\Omega_1' = \{1, \ldots, j\}$ and $\Omega_2' = \{j+1, \ldots, n\}$.

Let us consider those $\mathcal{V} \subseteq \Omega_2 = \{j, \ldots, n\}$ where $j \notin \mathcal{V}$ in (126). Each such $\mathcal{V} \subseteq \Omega_2' = \{j+1, \ldots, n\}$ as $j \notin \mathcal{V}$. Since the variable $t_j$ does not appear in the right hand side in (126), the Fourier-Motzkin elimination will not modify the inequality. We can reinterpret (126) as:

$$z \leq \sum_{i \in \mathcal{S}'} q_i + \sum_{i \in \mathcal{V}'} t_i + \left( \sum_{i \in \mathcal{S}'^c \cup \mathcal{V}'^c} p_i \right)^2 + 2 \left( \sum_{i \in \mathcal{S}' \cup \mathcal{V}'} p_i \right) \left( \sum_{i \in \mathcal{S}'^c \cup \mathcal{V}'^c} p_i \right),$$
$$\forall \mathcal{S}' \subseteq \Omega_1', j \notin \mathcal{S}', \mathcal{V}' \subseteq \Omega_2', \mathcal{S}'^c = \Omega_1' \setminus \mathcal{S}', \mathcal{V}'^c = \Omega_2' \setminus \mathcal{V}' \tag{133}$$

Next consider the case in (126) where $j \in \mathcal{V}$. In order to apply Fourier-Motzkin elimination, we express $\mathcal{V} = \{j\} \cup \mathcal{V}'$ where $\mathcal{V}' \subseteq \Omega_2' = \{j+1, \ldots, n\}$. We explicitly consider the variable $t_j$ in (126) below.

$$z \leq \sum_{i \in \mathcal{S}} q_i + \sum_{i \in \mathcal{V}'} t_i + t_j + \left( \sum_{i \in \mathcal{S}^c \cup \mathcal{V}^c} p_i \right)^2 + 2 \left( \sum_{i \in \mathcal{S} \cup \mathcal{V}} p_i \right) \left( \sum_{i \in \mathcal{S}^c \cup \mathcal{V}^c} p_i \right), \tag{134}$$

We first combine (134) with the inequality $t_j \leq q_j$ and introduce $\Omega_1' = \Omega_1 \cup \{j\}$, $\Omega_2' = \Omega_2 \setminus \{j\}$, $\mathcal{S}' = \mathcal{S} \cup \{j\}$ and $\mathcal{S}'^c = \Omega_1' \setminus \mathcal{S}'$, $\mathcal{V}' = \mathcal{V} \setminus \{j\} \subseteq \Omega_2'$ and $\mathcal{V}'^c = \Omega_2' \setminus \mathcal{V}'$ to have:

$$z \leq \sum_{i \in \mathcal{S}'} q_i + \sum_{i \in \mathcal{V}'} t_i + \left( \sum_{i \in \mathcal{S}'^c \cup \mathcal{V}'^c} p_i \right)^2 + 2 \left( \sum_{i \in \mathcal{S}' \cup \mathcal{V}'} p_i \right) \left( \sum_{i \in \mathcal{S}'^c \cup \mathcal{V}'^c} p_i \right),$$
$$\forall \mathcal{S}' \subseteq \Omega_1', j \in \mathcal{S}', \mathcal{V}' \subseteq \Omega_2', \mathcal{S}'^c = \Omega_1' \setminus \mathcal{S}', \mathcal{V}'^c = \Omega_2' \setminus \mathcal{V}' \tag{135}$$

Note that (133) and (135) recover all the upper bounds on $z$ in the induction step for (126). We further need to show that the Fourier-Motzkin elimination does not introduce any further inequalities during the elimination of $t_j$. In particular with $j \in \mathcal{V}$ consider combining:

$$z \leq \sum_{i \in \mathcal{S}} q_i + \sum_{i \in \mathcal{V}} t_i + \left( \sum_{i \in \mathcal{S}^c \cup \mathcal{V}^c} p_i \right)^2 + 2 \left( \sum_{i \in \mathcal{S} \cup \mathcal{V}} p_i \right) \left( \sum_{i \in \mathcal{S}^c \cup \mathcal{V}^c} p_i \right), \tag{136}$$

with the inequality:

$$\sum_{i \in \mathcal{W}} t_i \le \left( \sum_{i \in \mathcal{W}} p_i \right)^2 + 2 \left( \sum_{i \in \mathcal{W}} p_i \right) \left( \sum_{i \in \mathcal{W}^c} p_i \right) \tag{137}$$

where $\mathcal{W} \subseteq \Omega_2 = \{j, \dots, n\}$ and $j \in \mathcal{W}$. Defining $\mathcal{W}_1 = \mathcal{W} \setminus \mathcal{V}$ and $\mathcal{U} = \mathcal{W} \cap \mathcal{V}$ we have:

$$\sum_{i \in \mathcal{W}} t_i = \left( \sum_{i \in \mathcal{W}_1} p_i + \sum_{i \in \mathcal{U}} p_i \right)^2 + 2 \left( \sum_{i \in \mathcal{W}_1} p_i + \sum_{i \in \mathcal{U}} p_i \right) \left( \sum_{i \in \Omega \setminus \mathcal{W}} p_i \right) \tag{138}$$

$$= \left( \sum_{i \in \mathcal{W}_1} p_i \right)^2 + \left( \sum_{i \in \mathcal{U}} p_i \right)^2 + 2 \left( \sum_{i \in \mathcal{W}_1} p_i \right) \left( \sum_{i \in \mathcal{U}} p_i \right)$$

$$+ 2 \left( \sum_{i \in \mathcal{W}_1} p_i \right) \left( \sum_{i \in \Omega \setminus \mathcal{W}} p_i \right) + 2 \left( \sum_{i \in \mathcal{U}} p_i \right) \left( \sum_{i \in \Omega \setminus \mathcal{W}} p_i \right) \tag{139}$$

$$= \left( \sum_{i \in \mathcal{W}_1} p_i \right)^2 + \left( \sum_{i \in \mathcal{U}} p_i \right)^2 + 2 \left( \sum_{i \in \mathcal{W}_1} p_i \right) \left( \sum_{i \in \mathcal{U}} p_i + \sum_{i \in \Omega \setminus \mathcal{W}} p_i \right)$$

$$+ \left( \sum_{i \in \mathcal{U}} p_i \right) \left( \sum_{i \in \Omega \setminus \mathcal{W}} p_i \right) \tag{140}$$

$$= \left( \sum_{i \in \mathcal{W}_1} p_i \right)^2 + \left( \sum_{i \in \mathcal{U}} p_i \right)^2 + 2 \left( \sum_{i \in \mathcal{W}_1} p_i \right) \left( \sum_{i \in \Omega \setminus \mathcal{W}_1} p_i \right) + 2 \left( \sum_{i \in \Omega \setminus \mathcal{W}} p_i \right) \left( \sum_{i \in \mathcal{U}} p_i \right). \tag{141}$$

Next we consider (136):

$$z \le \sum_{i \in \mathcal{S}} q_i + \sum_{i \in \mathcal{V}_1} t_i + \sum_{i \in \mathcal{U}} t_1 + \left( \sum_{i \in \mathcal{S}^c \cup \mathcal{V}^c} p_i \right)^2$$

$$+ 2 \left( \sum_{i \in \mathcal{S} \cup \mathcal{V}_1} p_i \right) \left( \sum_{i \in \mathcal{S}^c \cup \mathcal{V}^c} p_i \right) + 2 \left( \sum_{i \in \mathcal{U}} p_i \right) \left( \sum_{i \in \mathcal{S}^c \cup \mathcal{V}^c} p_i \right), \tag{142}$$

where we use the fact that $\mathcal{V} = \mathcal{V}_1 \cup \mathcal{U}$. Adding (142) to (141) and eliminating $t_i$ where $i \in \mathcal{U}$, we get:

$$z + \sum_{i \in \mathcal{W}_1} t_i \le \sum_{i \in \mathcal{S}} q_i + \sum_{i \in \mathcal{V}_1} t_i + \left( \sum_{i \in \mathcal{S}^c \cup \mathcal{V}^c} p_i \right)^2$$

$$+ 2 \left( \sum_{i \in \mathcal{S} \cup \mathcal{V}_1} p_i \right) \left( \sum_{i \in \mathcal{S}^c \cup \mathcal{V}^c} p_i \right) + 2 \left( \sum_{i \in \mathcal{U}} p_i \right) \left( \sum_{i \in \mathcal{S}^c \cup \mathcal{V}^c} p_i \right)$$

$$+ \left( \sum_{i \in \mathcal{W}_1} p_i \right)^2 + \left( \sum_{i \in \mathcal{U}} p_i \right)^2 + 2 \left( \sum_{i \in \mathcal{W}_1} p_i \right) \left( \sum_{i \in \Omega \setminus \mathcal{W}_1} p_i \right) + 2 \left( \sum_{i \in \Omega \setminus \mathcal{W}} p_i \right) \left( \sum_{i \in \mathcal{U}} p_i \right) \tag{143}$$

$$= \sum_{i \in \mathcal{S}} q_i + \sum_{i \in \mathcal{V}_1} t_i + \left( \sum_{i \in \mathcal{S}^c \cup \mathcal{V}^c} p_i + \sum_{i \in \mathcal{U}} p_i \right)^2$$

$$+ 2 \left( \sum_{i \in \mathcal{S} \cup \mathcal{V}_1} p_i \right) \left( \sum_{i \in \mathcal{S}^c \cup \mathcal{V}^c} p_i \right) + 2 \left( \sum_{i \in \Omega \setminus \mathcal{W}} p_i \right) \left( \sum_{i \in \mathcal{U}} p_i \right)$$

$$+ \left( \sum_{i \in \mathcal{W}_1} p_i \right)^2 + 2 \left( \sum_{i \in \mathcal{W}_1} p_i \right) \left( \sum_{i \in \Omega \setminus \mathcal{W}_1} p_i \right) \tag{144}$$

Next note that $\mathcal{S} \cup \mathcal{V}_1 \subseteq \Omega \setminus \mathcal{W}$. This follows since $\Omega = \Omega_1 \cup \Omega_2$ and $\mathcal{S} \subseteq \Omega_2$, $\mathcal{V}_1, \mathcal{W} \subseteq \Omega_2$ and $\mathcal{V}_1 \cup \mathcal{W} = \cdot$ by definition as $\mathcal{V}_1 = \mathcal{V} \setminus \mathcal{W}$. Thus the application of Fourier-Motzkin elimination with $\mathcal{V}_1^c = \mathcal{V}^c \cup \mathcal{U}$ gives:

$$z + \sum_{i \in \mathcal{W}_1} t_i \leq \sum_{i \in \mathcal{S}} q_i + \sum_{i \in \mathcal{V}_1} t_i + \left( \sum_{i \in \mathcal{S}^c \cup \mathcal{V}_1^c} p_i \right) + 2 \left( \sum_{i \in \mathcal{S} \cup \mathcal{V}_1} p_i \right) \left( \sum_{i \in \mathcal{S}^c \cup \mathcal{V}_1^c} p_i \right)$$
$$+ \left( \sum_{i \in \mathcal{W}_1} p_i \right)^2 + 2 \left( \sum_{i \in \mathcal{W}_1} p_i \right) \left( \sum_{i \in \Omega \setminus \mathcal{W}_1} p_i \right) \tag{145}$$

However the above inequality is a consequence of the following:

$$z \leq \sum_{i \in \mathcal{S}} q_i + \sum_{i \in \mathcal{V}_1} t_i + \left( \sum_{i \in \mathcal{S}^c \cup \mathcal{V}_1^c} p_i \right) + 2 \left( \sum_{i \in \mathcal{S} \cup \mathcal{V}_1} p_i \right) \left( \sum_{i \in \mathcal{S}^c \cup \mathcal{V}_1^c} p_i \right)$$
$$\sum_{i \in \mathcal{W}_1} t_i \leq \left( \sum_{i \in \mathcal{W}_1} p_i \right)^2 + 2 \left( \sum_{i \in \mathcal{W}_1} p_i \right) \left( \sum_{i \in \Omega \setminus \mathcal{W}_1} p_i \right) \tag{146}$$

where $\mathcal{V}_1 \subseteq \Omega_2$, $\mathcal{V}_1^c = \Omega_2 \setminus \mathcal{V}_1$, $\mathcal{S} \subset \Omega_1$ and $\mathcal{S}^c \subseteq \Omega_1 \setminus \mathcal{S}$ and $\mathcal{W}_1 \subseteq \Omega_2$. which are already implied in the induction step. Thus we conclude that each combination of the form (136) and (137) is redundant and need not be included in the next step of the Fourier-Motzkin elimination. This concludes the analysis of the upper bound on $z$ in (126).

It remains to establish the induction for (127) and (128) i.e., upon elimination of $t_j$ results in

$$\sum_{i \in \mathcal{S}} t_i \leq \left( \sum_{i \in \mathcal{S}} p_i \right)^2 + 2 \left( \sum_{i \in \mathcal{S}} p_i \right) \left( \sum_{i \in \mathcal{S}^c} p_i \right), \quad \forall \mathcal{S} \subseteq \Omega_2', \quad \mathcal{S}^c = \Omega_2' \setminus \mathcal{S} \tag{147}$$
$$t_i \leq q_i, \qquad \forall i \in \Omega_2' \tag{148}$$

where $\Omega_2' = \{j + 1, \ldots, n\}$. Naturally every inequality (147) and (148) is already contained in (127) and (128) where $j \notin \mathcal{S}$. So we only need to show that the application of Fourier-Motzkin elimination to remove any other inequality does not result in any additional inequality. Note that the elimination of $t_j$ simply involves combining each inequality with $t_j > 0$. Thus any inequality in (127) where $\mathcal{S} \subseteq \Omega_2$ with $j \in \mathcal{S}$ reduces to:

$$\sum_{i \in \mathcal{S} \setminus \{j\}} t_i \leq \left( \sum_{i \in \mathcal{S}} p_i \right)^2 + 2 \left( \sum_{i \in \mathcal{S}} p_i \right) \left( \sum_{i \in \mathcal{S}^c} p_i \right) \tag{149}$$

We show that (149) is weaker than

$$\sum_{i \in \mathcal{S} \setminus \{j\}} t_i \leq \left( \sum_{i \in \mathcal{S} \setminus \{j\}} p_i \right)^2 + 2 \left( \sum_{i \in \mathcal{S} \setminus \{j\}} p_i \right) \left( \sum_{i \in \mathcal{S}^c \cup \{j\}} p_i \right) \tag{150}$$

which is already contained in (127) and hence redundant. In particular consider the right hand side of (149):

$$\left( \sum_{i \in \mathcal{S} \setminus \{j\}} p_i + p_j \right)^2 + 2 \left( \sum_{i \in \mathcal{S} \setminus \{j\}} p_i + p_j \right) \left( \sum_{i \in \mathcal{S}^c} p_i \right) \tag{151}$$

$$\geq p_j^2 + 2p_j \left( \sum_{i \in \mathcal{S} \setminus \{j\}} p_i \right) + \left( \sum_{i \in \mathcal{S} \setminus \{j\}} p_i \right)^2 + 2 \left( \sum_{i \in \mathcal{S} \setminus \{j\}} p_i \right) \left( \sum_{i \in \mathcal{S}^c} p_i \right) \tag{152}$$

$$\geq \left( \sum_{i \in \mathcal{S} \setminus \{j\}} p_i \right)^2 + 2 \left( \sum_{i \in \mathcal{S} \setminus \{j\}} p_i \right) \left( \sum_{i \in \mathcal{S}^c \cup \{j\}} p_i \right), \tag{153}$$

which implies that (149) is indeed weaker.

Thus we have completed the induction step. Continuing the induction to eliminate all variables $t_1, \ldots, t_n$ results in

$$z \leq \sum_{i \in \mathcal{S}} q_i + \left( \sum_{i \in \mathcal{S}^c} p_i \right)^2 + 2 \left( \sum_{i \in \mathcal{S}} p_i \right) \left( \sum_{i \in \mathcal{S}^c} p_i \right), \qquad \forall \mathcal{S} \in \Omega \tag{154}$$

as claimed. It now only remains to establish the proof of Lemma 3 which we will do. As we are considering the elimination of $\alpha_{i,j}$ it suffices to consider the following inequalities:

$$t_i \leq p_i^2 + \sum_{j=1, j \neq i}^{n} \alpha_{i,j}, \qquad i = 1, 2, \ldots, n \tag{155}$$

$$\alpha_{i,j} + \alpha_{j,i} = 2p_i p_j, \qquad 0 \leq \alpha_{i,j} \leq 1 \tag{156}$$

We will show the following by induction. Suppose that at step $r >= 1$ let

$$\mathcal{V}_r = \{(i_1, j_1), (j_1, i_1), \ldots, (i_{r-1}, j_{r-1}), (j_{r-1}, i_{r-1})\} \tag{157}$$

denotes the indices (with $i_k \leq j_k$) of variables that are eliminated using the Fourier-Motzkin elimination. Then the resulting system of inequalities is given by:

$$\sum_{s \in \mathcal{S}} t_s \leq \left( \sum_{s \in \mathcal{S}} p_s \right)^2 + \sum_{s \in \mathcal{S}} \sum_{t \in \mathcal{S}^c} \alpha_{s,t} \cdot \mathbb{I}((s,t) \notin \mathcal{V}_r) + \sum_{s \in \mathcal{S}} \sum_{t \in \mathcal{S}^c} 2p_s p_t \cdot \mathbb{I}((s,t) \in \mathcal{V}_r) \tag{158}$$

for all $\mathcal{S} \subseteq \Omega = \{1, 2, \ldots, n\}$. For the base case, consider the case when $r = 1$ i.e., $\mathcal{V}_r = \{\cdot\}$. The condition in (158) reduces to:

$$\sum_{s \in \mathcal{S}} t_s \leq \left( \sum_{s \in \mathcal{S}} p_s \right)^2 + \sum_{s \in \mathcal{S}} \sum_{t \in \mathcal{S}^c} \alpha_{s,t} \cdot \qquad \forall \mathcal{S} \subseteq \Omega \tag{159}$$

We show that (159) is equivalent to (155). Indeed setting $\mathcal{S} = \{i\}$ in (158) and using $\alpha_{s,t} = 2p_s p_t$ recovers (155) for each $i = 1, 2, \ldots, n$. We will show that the conditions (155) and (156) also imply (158). Note that for any $\mathcal{S} \subseteq \Omega$:

$$\sum_{s \in \mathcal{S}} t_s \leq \sum_{s \in \mathcal{S}} p_s^2 + \sum_{s \in \mathcal{S}} \sum_{i=1, i \neq s}^{n} \alpha_{s,t} \tag{160}$$

$$= \sum_{s \in \mathcal{S}} p_s^2 + \sum_{s \in \mathcal{S}} \left( \sum_{i \in \mathcal{S}^c} \alpha_{s,i} + \sum_{i \in \mathcal{S} \setminus \{s\}} \alpha_{s,i} \right) \tag{161}$$

$$= \sum_{s \in \mathcal{S}} p_s^2 + \sum_{s \in \mathcal{S}} \sum_{i \in \mathcal{S}^c} \alpha_{s,i} + \sum_{s \in \mathcal{S}} \sum_{i \in \mathcal{S} \setminus \{s\}} \alpha_{s,i} \tag{162}$$

$$= \sum_{s \in \mathcal{S}} p_s^2 + \sum_{s \in \mathcal{S}} \sum_{i \in \mathcal{S}^c} \alpha_{s,i} + \sum_{(s,i) \in \mathcal{S} \times \mathcal{S}, i > s} (\alpha_{s,i} + \alpha_{i,s}) \tag{163}$$

$$= \sum_{s \in \mathcal{S}} p_s^2 + \sum_{s \in \mathcal{S}} \sum_{i \in \mathcal{S}^c} \alpha_{s,i} + \sum_{(s,i) \in \mathcal{S} \times \mathcal{S}, i > s} 2p_{s,i} \tag{164}$$

$$= \left( \sum_{s \in \mathcal{S}} p_s \right)^2 + \sum_{s \in \mathcal{S}} \sum_{i \in \mathcal{S}^c} \alpha_{s,i} \tag{165}$$

We thus recover (159) from (155). This establishes the base case.

For the induction step, let us assume that we have eliminated all $\alpha_{i,j}$ where the indices $(i, j)$ are in the set $\mathcal{V}_r$ and that (155) is satisfied. We consider elimination of indices $(i_r, j_r)$ and $(j_r, i_r)$ associated with $\alpha_{i_r, j_r}$ and $\alpha_{j_r, i_r}$:

$$\sum_{s \in \mathcal{S}} t_s \leq \left( \sum_{s \in \mathcal{S}} p_s \right)^2 + \sum_{s \in \mathcal{S}} \sum_{i \in \mathcal{S}^c} \alpha_{s,i} \cdot \mathbb{I}((s,i) \notin \mathcal{V}_r) + \sum_{s \in \mathcal{S}} \sum_{i \in \mathcal{S}^c} 2p_s p_i \cdot \mathbb{I}((s,i) \in \mathcal{V}_r) \quad (166)$$

We need to show that upon elimination of $\alpha_{i_r, j_r}$ and $\alpha_{j_r, i_r}$ using Fourier-Motzkin elimination the resulting system of inequalities is given by:

$$\sum_{s \in \mathcal{S}} t_s \leq \left( \sum_{s \in \mathcal{S}} p_s \right)^2 + \sum_{s \in \mathcal{S}} \sum_{i \in \mathcal{S}^c} \alpha_{s,i} \cdot \mathbb{I}((s,i) \notin \mathcal{V}_{r+1}) \sum_{s \in \mathcal{S}} \sum_{i \in \mathcal{S}^c} 2p_s p_i \cdot \mathbb{I}((s,i) \in \mathcal{V}_{r+1}), \quad (167)$$

with

$$\mathcal{V}_{r+1} = \{(i_1, j_1), (j_1, i_1), \ldots, (i_r, j_r), (j_r, i_r)\}. \quad (168)$$

We note that in the Fourier-Motzkin elimination step we have to only consider those inequalities where either $\alpha_{i_r, j_r}$ or $\alpha_{j_r, i_r}$ appears on the right hand side of (166). This is equivalent to having $i_r \in \mathcal{S}$ and $j_r \in \mathcal{S}^c$ or $j_r \in \mathcal{S}$ and $i_r \in \mathcal{S}^c$. For those $\mathcal{S}$ that do not satisfy either condition, we immediately have (167). If the selected $\mathcal{S}$ follow either of these cases, combining (166) with $\alpha_{i_r, j_r} \leq 2p_{i_r} p_{j_r}$ and $\alpha_{j_r, i_r} \leq 2p_{i_r} p_{j_r}$, we reduce to (167). At this point all the equations in (167) have been recovered. Nevertheless Fourier-Motzkin elimination requires us to also consider all pairwise equations where $\mathcal{S}_1, \mathcal{S}_2 \subseteq \Omega$, $i_r \in \mathcal{S}_1$ and $j_r \notin \mathcal{S}_1$ and $i_r \notin \mathcal{S}_2$ and $j_r \in \mathcal{S}_2$:

$$\sum_{s \in \mathcal{S}_1} t_s - \left( \sum_{s \in \mathcal{S}_1} p_s \right)^2 \leq \sum_{s \in \mathcal{S}_1} \sum_{i \in \mathcal{S}_1^c} \alpha_{s,i} \mathbb{I}((s,i) \notin \mathcal{V}_r) + \sum_{s \in \mathcal{S}_1} \sum_{i \in \mathcal{S}_1^c} 2p_s p_i \cdot \mathbb{I}((s,i) \in \mathcal{V}_r) \quad (169)$$

$$\sum_{s \in \mathcal{S}_2} t_s - \left( \sum_{s \in \mathcal{S}_2} p_s \right)^2 \leq \sum_{s \in \mathcal{S}_2} \sum_{i \in \mathcal{S}_2^c} \alpha_{s,i} \mathbb{I}((s,i) \notin \mathcal{V}_r) + \sum_{s \in \mathcal{S}_2} \sum_{i \in \mathcal{S}_2^c} 2p_s p_i \cdot \mathbb{I}((s,i) \in \mathcal{V}_r) \quad (170)$$

The Fourier-Motzkin elimination step requires us to combine (169) and (170) and use $\alpha_{i_r, j_r} + \alpha_{j_r, i_r} = 2p_{i_r} p_{j_r}, \alpha_{i_r, j_r}, \alpha_{j_r, i_r} \geq 0$ to eliminate $\alpha_{i_r, j_r}$ and $\alpha_{j_r, i_r}$ in the induction step.

$$\sum_{s \in \mathcal{S}_1} t_s - \left( \sum_{s \in \mathcal{S}_1} p_s \right)^2 + \sum_{s \in \mathcal{S}_2} t_s - \left( \sum_{s \in \mathcal{S}_2} p_s \right)^2$$
$$\leq \sum_{s \in \mathcal{S}_1} \sum_{i \in \mathcal{S}_1^c} \alpha_{s,i} \mathbb{I}((s,i) \notin \mathcal{V}_r) + \sum_{s \in \mathcal{S}_1} \sum_{i \in \mathcal{S}_1^c} 2p_s p_i \cdot \mathbb{I}((s,i) \in \mathcal{V}_r)$$
$$+ \sum_{s \in \mathcal{S}_2} \sum_{i \in \mathcal{S}_2^c} \alpha_{s,i} \mathbb{I}((s,i) \notin \mathcal{V}_r) + \sum_{s \in \mathcal{S}_2} \sum_{i \in \mathcal{S}_2^c} 2p_s p_i \cdot \mathbb{I}((s,i) \in \mathcal{V}_r) \quad (171)$$

We will show that each such inequality is redundant and already implied by the set of equations already established in (167).

Let $\mathcal{R} = \mathcal{S}_1 \cap \mathcal{S}_2$ and $\mathcal{T} = \mathcal{S}_1 \cup \mathcal{S}_2$. Note that $i_r \notin \mathcal{R}$ and $j_r \notin \mathcal{R}$. First following the same steps leading to (78) we can show that:

$$\sum_{s \in \mathcal{S}_1} t_s - \left( \sum_{s \in \mathcal{S}_1} p_s \right)^2 + \sum_{s \in \mathcal{S}_2} t_s - \left( \sum_{s \in \mathcal{S}_2} p_s \right)^2$$
$$= \left\{ \sum_{s \in \mathcal{T}} t_s - \left( \sum_{s \in \mathcal{T}} p_s \right)^2 \right\} + \left\{ \sum_{s \in \mathcal{R}} t_s - \left( \sum_{s \in \mathcal{R}} p_s \right)^2 \right\} + 2 \left( \sum_{s \in \mathcal{S}_2 \setminus \mathcal{R}} p_s \right) \left( \sum_{s \in \mathcal{S}_1 \setminus \mathcal{R}} p_s \right).$$
$$(172)$$

Next, following the steps leading to (84) we have that:

$$
\sum_{s \in \mathcal{S}_1} \sum_{i \in \mathcal{S}_1^c} \alpha_{s,i} \cdot \mathbb{I}((s,i) \notin \mathcal{V}_r) + 2 p_s p_i \cdot \mathbb{I}((s,i) \in \mathcal{V}_r)
$$

$$
= \sum_{s \in \mathcal{S}_1} \sum_{i \in \mathcal{T}^c} \alpha_{s,i} \cdot \mathbb{I}((s,i) \notin \mathcal{V}_r) + 2 p_s p_i \cdot \mathbb{I}((s,i) \in \mathcal{V}_r)
$$

$$
+ \sum_{s \in \mathcal{S}_1} \sum_{i \in \mathcal{S}_2 \setminus \mathcal{R}} \alpha_{s,i} \cdot \mathbb{I}((s,i) \notin \mathcal{V}_r) + 2 p_s p_i \cdot \mathbb{I}((s,i) \in \mathcal{V}_r) \tag{173}
$$

$$
= \sum_{s \in \mathcal{S}_1 \setminus \mathcal{R}} \sum_{i \in \mathcal{T}^c} \alpha_{s,i} \cdot \mathbb{I}((s,i) \notin \mathcal{V}_r) + 2 p_s p_i \cdot \mathbb{I}((s,i) \in \mathcal{V}_r)
$$

$$
+ \sum_{s \in \mathcal{R}} \sum_{i \in \mathcal{T}^c} \alpha_{s,i} \cdot \mathbb{I}((s,i) \notin \mathcal{V}_r) + 2 p_s p_i \cdot \mathbb{I}((s,i) \in \mathcal{V}_r)
$$

$$
+ \sum_{s \in \mathcal{S}_1 \setminus \mathcal{R}} \sum_{i \in \mathcal{S}_2 \setminus \mathcal{R}} \alpha_{s,i} \cdot \mathbb{I}((s,i) \notin \mathcal{V}_r) + 2 p_s p_i \cdot \mathbb{I}((s,i) \in \mathcal{V}_r)
$$

$$
+ \sum_{s \in \mathcal{R}} \sum_{i \in \mathcal{S}_2 \setminus \mathcal{R}} \alpha_{s,i} \cdot \mathbb{I}((s,i) \notin \mathcal{V}_r) + 2 p_s p_i \cdot \mathbb{I}((s,i) \in \mathcal{V}_r) \tag{174}
$$

and, likewise,

$$
\sum_{s \in \mathcal{S}_2} \sum_{i \in \mathcal{S}_2^c} \alpha_{s,i} \cdot \mathbb{I}((s,i) \notin \mathcal{V}_r) + 2 p_s p_i \cdot \mathbb{I}((s,i) \in \mathcal{V}_r)
$$

$$
= \sum_{s \in \mathcal{S}_2 \setminus \mathcal{R}} \sum_{i \in \mathcal{T}^c} \alpha_{s,i} \cdot \mathbb{I}((s,i) \notin \mathcal{V}_r) + 2 p_s p_i \cdot \mathbb{I}((s,i) \in \mathcal{V}_r)
$$

$$
+ \sum_{s \in \mathcal{R}} \sum_{i \in \mathcal{T}^c} \alpha_{s,i} \cdot \mathbb{I}((s,i) \notin \mathcal{V}_r) + 2 p_s p_i \cdot \mathbb{I}((s,i) \in \mathcal{V}_r)
$$

$$
+ \sum_{s \in \mathcal{S}_2 \setminus \mathcal{R}} \sum_{i \in \mathcal{S}_1 \setminus \mathcal{R}} \alpha_{s,i} \cdot \mathbb{I}((s,i) \notin \mathcal{V}_r)) + 2 p_s p_i \cdot \mathbb{I}((s,i) \in \mathcal{V}_r)
$$

$$
+ \sum_{s \in \mathcal{R}} \sum_{i \in \mathcal{S}_1 \setminus \mathcal{R}} \alpha_{s,i} \cdot \mathbb{I}((s,i) \notin \mathcal{V}_r) + 2 p_s p_i \cdot \mathbb{I}((s,i) \in \mathcal{V}_r) \tag{175}
$$

Next we combine the terms to get:

$$
\sum_{s \in \mathcal{S}_1} \sum_{i \in \mathcal{S}_1^c} \alpha_{s,i} \cdot \mathbb{I}((s,i) \notin \mathcal{V}_r) + 2 p_s p_i \cdot \mathbb{I}((s,i) \in \mathcal{V}_r)
$$

$$
+ \sum_{s \in \mathcal{S}_2} \sum_{i \in \mathcal{S}_2^c} \alpha_{s,i} \cdot \mathbb{I}((s,i) \notin \mathcal{V}_r) + 2 p_s p_i \cdot \mathbb{I}((s,i) \in \mathcal{V}_r)
$$

$$
= \Bigg\{ \sum_{s \in \mathcal{S}_1 \setminus \mathcal{R}} \sum_{i \in \mathcal{T}^c} \alpha_{s,i} \cdot \mathbb{I}((s,i) \notin \mathcal{V}_r) + 2 p_s p_i \cdot \mathbb{I}((s,i) \in \mathcal{V}_r)
$$

$$
+ \sum_{s \in \mathcal{R}} \sum_{i \in \mathcal{T}^c} \alpha_{s,i} \cdot \mathbb{I}((s,i) \notin \mathcal{V}_r) + 2 p_s p_i \cdot \mathbb{I}((s,i) \in \mathcal{V}_r)
$$

$$
+ \sum_{s \in \mathcal{S}_2 \setminus \mathcal{R}} \sum_{i \in \mathcal{T}^c} \alpha_{s,i} \cdot \mathbb{I}((s,i) \notin \mathcal{V}_r) + 2 p_s p_i \cdot \mathbb{I}((s,i) \in \mathcal{V}_r) \Bigg\}
$$

$$
+ \Bigg\{ \sum_{s \in \mathcal{R}} \sum_{i \in \mathcal{S}_2 \setminus \mathcal{R}} \alpha_{s,i} \cdot \mathbb{I}((s,i) \notin \mathcal{V}_r) + 2 p_s p_i \cdot \mathbb{I}((s,i) \in \mathcal{V}_r)
$$

$$
+ \sum_{s \in \mathcal{R}} \sum_{i \in \mathcal{T}^c} \alpha_{s,i} \cdot \mathbb{I}((s,i) \notin \mathcal{V}_r) + 2 p_s p_i \cdot \mathbb{I}((s,i) \in \mathcal{V}_r)
$$

$$
+ \sum_{s \in \mathcal{R}} \sum_{i \in \mathcal{S}_1 \setminus \mathcal{R}} \alpha_{s,i} \cdot \mathbb{I}((s,i) \notin \mathcal{V}_r) + 2 p_s p_i \cdot \mathbb{I}((s,i) \in \mathcal{V}_r) \Bigg\} \tag{176}
$$

$$+ \sum_{s \in \mathcal{S}_1 \setminus \mathcal{R}} \sum_{i \in \mathcal{S}_2 \setminus \mathcal{R}} \alpha_{s,i} \cdot \mathbb{I}((s,i) \notin \mathcal{V}_r) + 2 p_s p_i \cdot \mathbb{I}((s,i) \in \mathcal{V}_r)$$

$$+ \sum_{s \in \mathcal{S}_2 \setminus \mathcal{R}} \sum_{i \in \mathcal{S}_1 \setminus \mathcal{R}} \alpha_{s,i} \cdot \mathbb{I}((s,i) \notin \mathcal{V}_r)) + 2 p_s p_i \cdot \mathbb{I}((s,i) \in \mathcal{V}_r) \tag{177}$$

$$= \sum_{s \in \mathcal{T}} \sum_{i \in \mathcal{T}^c} \alpha_{s,i} \cdot \mathbb{I}((s,i) \notin \mathcal{V}_r) + 2 p_s p_i \cdot \mathbb{I}((s,i) \in \mathcal{V}_r)$$

$$+ \sum_{s \in \mathcal{R}} \sum_{i \in \mathcal{R}^c} \alpha_{s,i} \cdot \mathbb{I}((s,i) \notin \mathcal{V}_r) + 2 p_s p_i \cdot \mathbb{I}((s,i) \in \mathcal{V}_r)$$

$$+ \sum_{s \in \mathcal{S}_1 \setminus \mathcal{R}} \sum_{i \in \mathcal{S}_2 \setminus \mathcal{R}} (\alpha_{s,i} + \alpha_{i,s}) \cdot \mathbb{I}((s,i) \notin \mathcal{V}_r) + 4 p_s p_i \cdot \mathbb{I}((s,i) \in \mathcal{V}_r) \tag{178}$$

$$= \sum_{s \in \mathcal{T}} \sum_{i \in \mathcal{T}^c} \alpha_{s,i} \cdot \mathbb{I}((s,i) \notin \mathcal{V}_r) + 2 p_s p_i \cdot \mathbb{I}((s,i) \in \mathcal{V}_r)$$

$$+ \sum_{s \in \mathcal{R}} \sum_{i \in \mathcal{R}^c} \alpha_{s,i} \cdot \mathbb{I}((s,i) \notin \mathcal{V}_r) + 2 p_s p_i \cdot \mathbb{I}((s,i) \in \mathcal{V}_r)$$

$$+ \sum_{s \in \mathcal{S}_1 \setminus \mathcal{R}} \sum_{i \in \mathcal{S}_2 \setminus \mathcal{R}} 2 p_s p_i + 2 p_s p_i \cdot \mathbb{I}((s,i) \in \mathcal{V}_r) \tag{179}$$

$$\tag{180}$$

Thus the resulting inequality from Fourier-Motzkin elimination is given by:

$$\left\{ \sum_{s \in \mathcal{T}} t_s - \left( \sum_{s \in \mathcal{T}} p_s \right)^2 \right\} + \left\{ \sum_{s \in \mathcal{R}} t_s - \left( \sum_{s \in \mathcal{R}} p_s \right)^2 \right\} + 2 \left( \sum_{s \in \mathcal{S}_2 \setminus \mathcal{R}} p_s \right) \left( \sum_{s \in \mathcal{S}_1 \setminus \mathcal{R}} p_s \right)$$

$$\leq \sum_{s \in \mathcal{T}} \sum_{i \in \mathcal{T}^c} \alpha_{s,i} \cdot \mathbb{I}((s,i) \notin \mathcal{V}_r) + 2 p_s p_i \cdot \mathbb{I}((s,i) \in \mathcal{V}_r)$$

$$+ \sum_{s \in \mathcal{R}} \sum_{i \in \mathcal{R}^c} \alpha_{s,i} \cdot \mathbb{I}((s,i) \notin \mathcal{V}_r) + 2 p_s p_i \cdot \mathbb{I}((s,i) \in \mathcal{V}_r)$$

$$+ \sum_{s \in \mathcal{S}_1 \setminus \mathcal{R}} \sum_{i \in \mathcal{S}_2 \setminus \mathcal{R}} 2 p_s p_i + 2 p_s p_i \cdot \mathbb{I}((s,i) \in \mathcal{V}_r) \tag{181}$$

Next note that since $(i_r, j_r) \in \mathcal{T}$ and $(i_r, j_r) \notin \mathcal{R}$, the inequalities:

$$\sum_{s \in \mathcal{T}} t_s \leq \left( \sum_{s \in \mathcal{T}} p_s \right)^2 + \sum_{s \in \mathcal{R}} \sum_{i \in \mathcal{R}^c} \alpha_{s,i} \cdot \mathbb{I}((s,i) \notin \mathcal{V}_r) + 2 p_s p_i \cdot \mathbb{I}((s,i) \in \mathcal{V}_r)$$

$$\sum_{s \in \mathcal{R}} t_s \leq \left( \sum_{s \in \mathcal{R}} p_s \right)^2 + \sum_{s \in \mathcal{R}} \sum_{i \in \mathcal{R}^c} \alpha_{s,i} \cdot \mathbb{I}((s,i) \notin \mathcal{V}_r) + 2 p_s p_i \cdot \mathbb{I}((s,i) \in \mathcal{V}_r) \tag{182}$$

are already constructed in the induction step. Also clearly (181) is implied by these since:

$$2 \left( \sum_{s \in \mathcal{S}_2 \setminus \mathcal{R}} p_s \right) \left( \sum_{s \in \mathcal{S}_1 \setminus \mathcal{R}} p_s \right) = \sum_{s \in \mathcal{S}_1 \setminus \mathcal{R}} \sum_{i \in \mathcal{S}_2 \setminus \mathcal{R}} 2 p_s p_i \tag{183}$$

is an identity since $\mathcal{S}_1 \setminus cR$ and $\mathcal{S}_2 \setminus \mathcal{R}$ are disjoint. Thus each such inequality form the Fourier-Motzkin elimination is redundant and we have completed the induction step and in turn established Lemma 3.

## F  PROOF OF EQUATION (12)

First we consider the non-truncated program and let $w_{i,j}$ be the variables for $i, j \in \Omega$ with $i < j$ that maximize the objective:

$$\sum_{i=1}^{n} \min(q_i, p_I(i)) \tag{184}$$

where

$$p_I(i) = p_i^2 + \sum_{j=1,j\neq i}^{n} 2p_i p_j w_{i,j} \tag{185}$$

Note that we have

$$P^\star(\text{acc}) = \sum_{i=1}^{n} \min(q_i, p_I(i)) \leq \sum_{i=1}^{s} \min(q_i, p_I(i)) + \sum_{i=s+1}^{n} q_i \tag{186}$$

For the truncated linear program we have for each $i \in \{1, 2, \ldots, s\}$:

$$\tilde{p}_I(i) = p_i^2 + \sum_{j=1,j\neq i}^{s} 2p_i p_j \tilde{w}_{i,j} \sum_{j=s+1}^{n} 2p_i p_j \tag{187}$$

and for $i > s$:

$$\tilde{p}_I(i) = p_i^2 + \sum_{j=i+1}^{n} 2p_i p_j \tag{188}$$

We consider a potentially sub-optimal choice of weights $\tilde{w}_{i,j} = w_{i,j}$ for $i,j \in \{1, \ldots, s\}$ for the truncated linear program. Note that

$$\tilde{p}_I(i) \geq p_I(i), \qquad \forall i \leq s \tag{189}$$

and

$$\tilde{p}_I(i) \geq p_i^2, \qquad \forall i > s. \tag{190}$$

As a result, using (186) we have:

$$\tilde{P}(\text{acc}) \geq \sum_{i=1}^{n} \min(q_i, \tilde{p}_I(i)) \tag{191}$$

$$\geq \sum_{i=1}^{s} \min(q_i, p_I(i)) + \sum_{i=s+1}^{n} \min(q_i, p_i^2) \tag{192}$$

$$\geq P^\star(\text{acc}) - \sum_{i=s+1}^{n} q_i + \sum_{i=s+1}^{n} \min(q_i, p_i^2) \tag{193}$$

$$= P^\star(acc) - \sum_{i=s+1}^{n} (q_i - p_i^2)^+ \tag{194}$$

## G  COMPUTATIONAL COMPLEXITY OF TRUNCATED LP

We study the computational complexity of the truncated linear program (see Algorithm 2) and propose a variation that could improve it further. Let $\Omega = \{1, 2, \ldots, n\}$ denote the vocabulary size under consideration. Note that the truncated linear program has two steps:

- Sort the candidate tokens based on $q_i - p_i^2$. This requires $O(n \log n)$ computations. We select that $s$ largest tokens and identify this set as $\Omega_1$.
- Apply linear program for the tokens in $\Omega_1$ which is the size of $s$. There are $O(s^2)$ variables and standard implementation of the linear program involves a complexity of $O(s^6)$.

Thus the overall complexity of our proposed scheme is $O(s^6 + n \log n)$ where $n$ is the alphabet size. In practice we keep the size of $\Omega_1$ relatively small e.g., we set $s = 5$ in many of our experiments. This avoids slowdown arising from the $s^6$ term in the linear program. Secondly, in practice, $\Omega$ is a subset of tokens from the original vocabulary after top-p sampling. So even when the original vocabulary is large, the value of $n$ in practice is relatively modest. In Appendix H we present the histogram of the alphabet size after top-p sampling to illustrate this point. Thus the $O(n \log n)$ is an acceptable cost for our implementation.

---

**Algorithm 2** Truncated LP

---

1: **Input:** Threshold $s$, Input tokens $X_1, X_2$ sampled independently from $p(\cdot)$
2: **Output:** Selected token $Y_I$, output distribution $p_I(\cdot)$.
3: Order vocabulary $\Omega = \{1, 2, \ldots, n\}$, sorted in decreasing order with $(q_i - p_i^2)$.
4: Set $\Omega_1 = \{1, \ldots, s\}$ and $\Omega_2 = \{s+1, \ldots, n\}$.
5: For $i, j \in \Omega_2$ or $i \in \Omega_1$ and $j \in \Omega_2$ set $w_{i,j}$ in (10)
6: For $i, j \in \Omega_1$, compute $w_{i,j}$ as a solution to a linear program:
         • Maximize: $\sum_{i=1}^{s} \min(q_i, p_I(i))$, where $p_I(i)$ is defined in (11)
         • Constraints: $w_{i,j} \geq 0$, $w_{i,j} + w_{j,i} = 1$
7: Compute $p_I(\cdot)$ according to (11).
8: **if** $X_1 = X_2$ **then**
9:      Set $Y_I = X_1$.
10: **else**
11:      Let $\{X_1, X_2\} = \{i, j\}$ and $i < j$.
12:      Set $Y_I = i$ with probability $w_{i,j}$, and $Y_I = j$ otherwise.
13: **end if**

---

### G.1 VARIANT OF TRUNCATED LP

We now present a variation of the LP that avoids the $O(n \log n)$ cost from sorting $\Omega$. The proposed method is as follows:

- Select $s$ tokens with largest values of $q_i - p_i^2$. This requires $O(n \cdot s)$ computations. We identify this set as $\Omega_1$ and the put the remaining tokens in $\Omega_2$.

- We propose the following heuristic choice of weights:

$$w_{i,j} = \begin{cases} 1, & i \in \Omega_1, j \in \Omega_2 \\ 1/2, & i \in \Omega_2, j \in \Omega_2, i \neq j \end{cases} \tag{195}$$

Note that this results in the following distribution of the selected token:

$$p_I(k) = \begin{cases} p_k^2 + \sum_{i=1, i \neq k}^{s} 2p_i p_k w_{k,i} + \sum_{i=s+1}^{n} 2p_i p_k, & k \in \Omega_1 \\ p_k^2 + \sum_{i \neq k}^{n} p_i p_k, & k \in \Omega_2 \end{cases} \tag{196}$$

We leave the weights $w_{i,j}$ for $i < j$ and $i, j \in \Omega_1$ as free parameters.

- Apply linear program for the tokens in $\Omega_1$ which is the size of $s$. There are $O(s^2)$ variables and standard implementation of the linear program involves a complexity of $O(s^6)$.

It can be seen that the overall complexity of the proposed method is $O(n \cdot s + s^6)$. By following the same steps as in Section F can also be verified that the resulting probabilities in (196) are also satisfy (12) yielding the same theoretical guarantee.

## H HISTOGRAM OF ALPHABET SIZE FOR OPT MODEL

We present additional evidence that after top-p sampling the effective alphabet size can be significantly reduced. We consider the OPT model and report the histogram of the alphabet size under the draft and target models for the XSum task in Fig. 5, the Dolly task in Fig. 6, and the WMT task in Fig. 7. The histogram is truncated to 40 tokens to make the figures clearer.

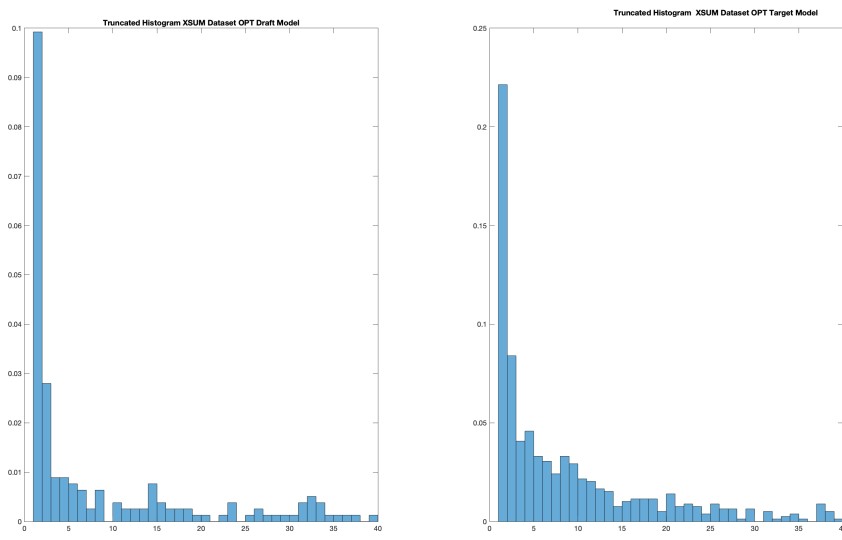

Figure 5: Truncated Histogram for OPT Draft and Target Models for the effective alphabet size after top-p sampling with p=0.95 on XSum dataset

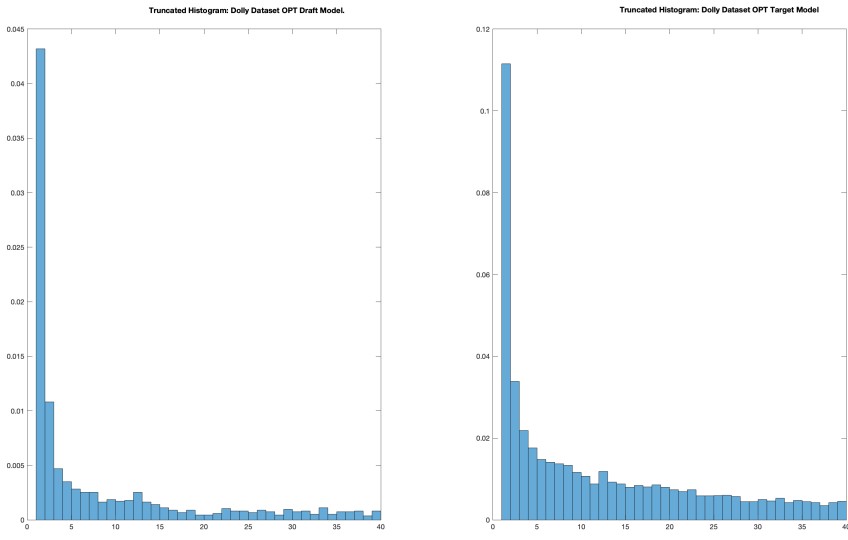

Figure 6: Truncated Histogram for OPT Draft and Target Models for the effective alphabet size after top-p sampling with p=0.95 on Dolly dataset

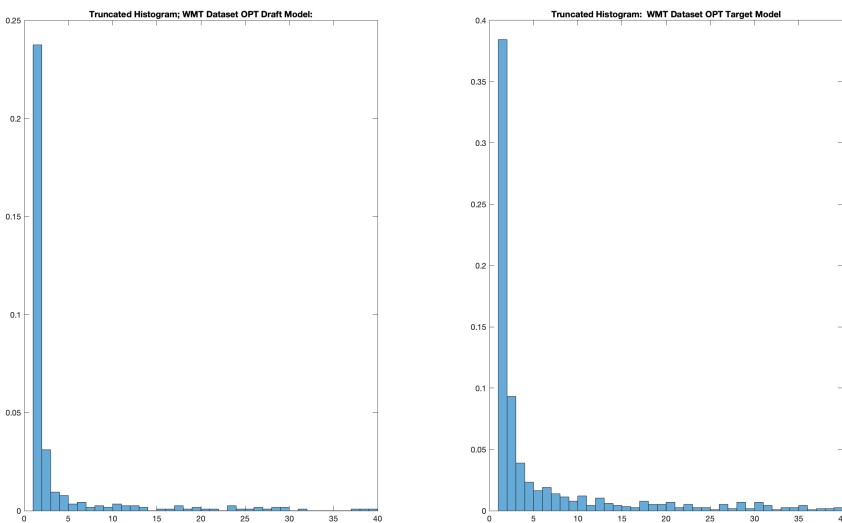

Figure 7: Truncated Histogram for OPT Draft and Target Models for the effective alphabet size after top-p sampling with p=0.95 on WMT dataset

# I  LP AND FAST LP VERSION FOR NON-IDENTICAL DRAFT DISTRIBUTIONS, $K = 2$

For the case of $K = 2$ drafts we explain how the importance weighted sampling scheme and its faster variants can be extended when the two tokens are sampled independently but from different distribution i.e. $X_1 \sim p_1(\cdot)$ and $X_2 \sim p_2(\cdot)$. We let $\mathbf{p}_1 = (p_{1,1}, \ldots, p_{1,n})$ and $\mathbf{p}_2 = (p_{2,1}, \ldots, p_{2,n})$ denote the distributions of the draft models to sample $X_1$ and $X_2$. We let $\mathbf{q} = (q_1, \ldots, q_n)$ denote the target distribution.

The order of the tokens matters and accordingly for $i < j$, we define:

$$w_{i,j} = \Pr(Y = i | X_1 = i, X_2 = j), \bar{w}_{i,j} = 1 - w_{i,j} = \Pr(Y = j | X_1 = i, X_2 = j) \quad (197)$$

$$w_{j,i} = \Pr(Y = i | X_1 = j, X_2 = i), \bar{w}_{j,i} = 1 - w_{j,i} = \Pr(Y = j | X_1 = j, X_2 = i) \quad (198)$$

If $Y$ denotes the selected token, then considering all cases where token $i$ appears as one of the input tokens, we have:

$$p_I(i) = p_{1,i}p_{2,i} + \sum_{j=1}^{i-1} p_{1,i}p_{2,j}\bar{w}_{i,j} + \sum_{j=i+1}^{n} p_{1,i}p_{2,j}w_{i,j} + \sum_{j=1}^{i-1} p_{1,j}p_{2,i}\bar{w}_{j,i} + \sum_{j=i+1}^{n} p_{1,j}p_{2,i}w_{j,i}$$
$$(199)$$

We need to find $w_{i,j}$ and $w_{j,i}$ that maximizes $\sum_{i=1}^{n} \min(q_i, p_I(i))$. This is a linear program in variables $w_{i,j}$ satisfying $0 \leq w_{i,j} \leq 1$. The truncated version of LP is obtained by sorting the tokens in $\Omega$ based on $q_i - p_{1,i}p_{2,i}$ again considering sets $\Omega_1 = \{1, 2, \ldots, s\}$ and $\Omega_2 = \{s + 1, \ldots, n\}$. We treat $w_{i,j}$ as variables that need to be optimized if $i, j \in \Omega_1$. If $i \in \Omega_1$ and $j \in \Omega_2$ we set $w_{i,j} = 1$. If both $i, j \in \Omega_2$ we set $w_{i,j} = 1$ if $i < j$ and $0$ if $i > j$. The resulting distribution is given as follows. For $i \in \{1, \ldots, s\}$:

$$\tilde{p}_I(i) = p_{1,i}p_{2,i} + \sum_{j=1}^{i-1} p_{1,i}p_{2,j}\bar{w}_{i,j} + \sum_{j=i+1}^{n} p_{1,i}p_{2,j}w_{i,j} + \sum_{j=1}^{i-1} p_{1,j}p_{2,i}\bar{w}_{j,i}$$
$$+ \sum_{j=i+1}^{n} p_{1,j}p_{2,i}w_{j,i} + \sum_{j=s+1}^{n} (p_{1,j}p_{2,i} + p_{1,i}p_{2,j}) \quad (200)$$

and for $i = s + 1, \ldots, n$, we have:

$$\tilde{p}_I(i) = p_{1,i}p_{2,i} + \sum_{j=i+1}^{n} (p_{1,j}p_{2,i} + p_{1,i}p_{2,j}) \quad (201)$$

Upon following the sequence of steps leading to (194) we can show that

$$\tilde{P}(\text{acc}) \geq P^\star(\text{acc}) - \sum_{i \in \Omega_2} (q_i - p_{1,i}p_{2,i})^+. \quad (202)$$

The truncated alphabet scheme can be applied in a similar fashion by considering a high probability subset $\Omega_0 \subseteq \Omega$ and only keeping those input tokens that belong to $\Omega_0$. We generate truncated distributions $\tilde{p}_1(\cdot)$ and $\tilde{p}_2(\cdot)$ and apply the linear program on these followed by speculative sampling using the target distribution $q(\cdot)$.

## J ADDITIONAL EXPERIMENTAL RESULTS

### J.1 TWO DRAFT MODELS WITH IDENTICAL TEMPERATURES

Here, we consider the case where identical draft models are used to generate candidate tokens sequences. We compare the performance of our method with SpecTr (Sun et al., 2024b) and SpecInfer (Miao et al., 2024), as well as single-draft speculative sampling (Leviathan et al., 2023; Chen et al., 2023), where we use the same temperature for the draft model.

In Figure 8, we set the temperature of the target model to 1.0, while we vary the temperature of the $K = 2$ draft models between the range of 1.2 to 2.4, and we report the performance achieved by the different schemes across the three tasks discussed earlier. The top plots report the block efficiency, which is the average number of tokens that are accepted per use of the draft model (Leviathan et al., 2023). The bottom plots show the percentage improvement in the token rate with respect to the baseline single draft scheme.

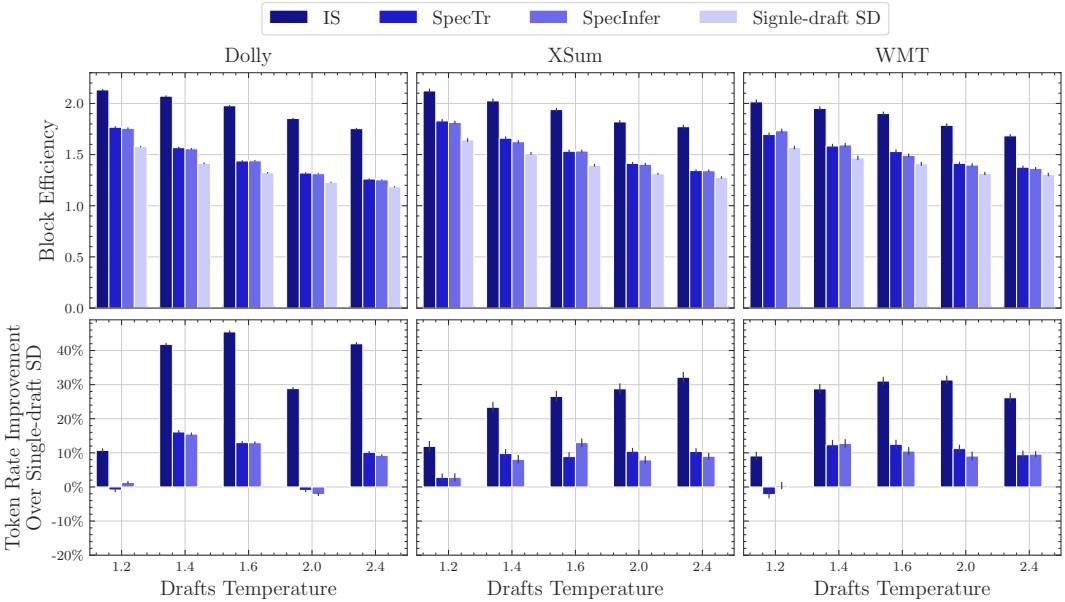

Figure 8: Performance comparison of different multi-draft schemes, while we vary the temperature of the two draft models.

We observe that the IS scheme consistently outperforms the both SpecTr and SpecInfer across all three tasks. In fact when the temperature of the second draft is increased, the improvement in the block efficiency of both SpecTr and SpecInfer is rather negligible compared to the single draft baseline, while the token rate are in fact lower than the single draft baseline. On the other hand our proposed IS scheme is able to achieve consistent improvements in all the three tasks. In the experiments involving the XSum and WMT tasks we also measure the ROUGE-L (Lin, 2004) and BLEU (Papineni et al., 2002) scores respectively, which are reported in the Appendix J.3.

Table 4 compares the block efficiencies for different multi-draft speculative sampling methods using $K = 2$ to $K = 6$ drafts when all the drafts are identical and use a sampling temperature of 1.2. We again see a consistent improvement by the proposed importance sampling scheme.

Table 4: Block efficiency achieved in the Dolly task for different number of draft models.

| Scheme | $K = 2$ | $K = 3$ | $K = 4$ | $K = 5$ | $K = 6$ |
|---|---|---|---|---|---|
| IS | $2.13 \pm 0.05$ | $2.22 \pm 0.05$ | $2.26 \pm 0.05$ | $2.27 \pm 0.05$ | $2.28 \pm 0.06$ |
| SpecInfer | $1.76 \pm 0.04$ | $1.86 \pm 0.05$ | $1.95 \pm 0.05$ | $2.00 \pm 0.04$ | $2.04 \pm 0.05$ |
| SpecTr | $1.77 \pm 0.04$ | $1.89 \pm 0.05$ | $1.96 \pm 0.05$ | $2.03 \pm 0.06$ | $2.08 \pm 0.04$ |

### J.2 THREE DRAFT MODELS

In this section, we consider the case of $K = 3$ draft models that use different temperatures for token generation. As stated before, in case of non-identical draft models, the only plausible multi-draft sampling scheme for comparison is the SpecInfer scheme (Miao et al., 2024). We set the temperature of the target model to 1.0, and the temperature of two of draft models to 1.2 and 1.4, while we vary the temperature of the first draft model between the range of 1.0 to 1.6. As observed in Figure 9, the proposed IS scheme outperforms SpecInfer and Single-draft speculative sampling method in terms of block efficiency and achieved token rate.

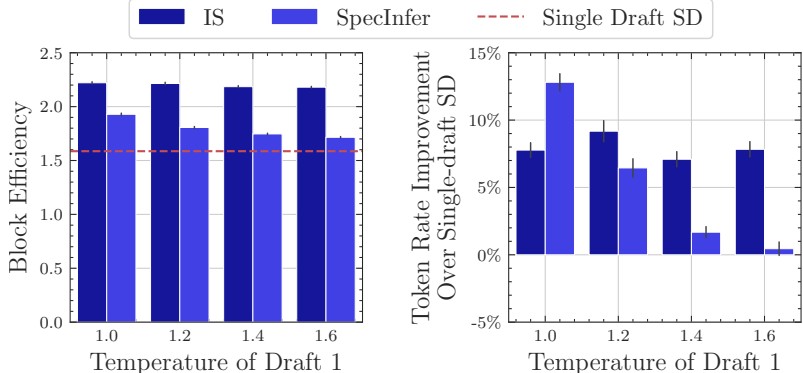

Figure 9: Performance comparison of different schemes on the Dolly dataset with $K = 3$ drafts. We vary the temperature of the first draft model and keeping the temperature of the other two drafts to 1.2 and 1.4, with the target model having a temperature of 1.0. The single-draft baseline uses a draft temperature of 1.2.

### J.3 ROUGE-L AND BLEU SCORES

In the experiments involving the XSum and WMT tasks we measure the ROUGE-L (Lin, 2004) and BLEU (Papineni et al., 2002) scores respectively. Table 5 and Table 6 show the ROUGE-L and BLEU scores for the case of identical draft models, corresponding to the experiment in Figure 8 in Section J.1.

Table 5: ROUGE-L scores on the XSum task across various decoders and sampling temperatures.

| Draft Temp. | 1.2 | 1.4 | 1.6 | 2.0 | 2.4 |
|---|---|---|---|---|---|
| **Decoder** | | | | | |
| IS | $0.186 \pm 0.004$ | $0.188 \pm 0.002$ | $0.191 \pm 0.003$ | $0.186 \pm 0.004$ | $0.187 \pm 0.003$ |
| Signle-draft SD | $0.190 \pm 0.006$ | $0.185 \pm 0.005$ | $0.190 \pm 0.004$ | $0.186 \pm 0.003$ | $0.186 \pm 0.004$ |
| SpecInfer | $0.184 \pm 0.004$ | $0.190 \pm 0.002$ | $0.187 \pm 0.001$ | $0.186 \pm 0.003$ | $0.186 \pm 0.004$ |
| SpecTr | $0.188 \pm 0.002$ | $0.182 \pm 0.006$ | $0.188 \pm 0.001$ | $0.185 \pm 0.006$ | $0.188 \pm 0.001$ |

Table 6: BLEU scores on the WMT dataset across various decoders and sampling temperatures.

| Draft Temp. | 1.2 | 1.4 | 1.6 | 2.0 | 2.4 |
|---|---|---|---|---|---|
| **Decoder** | | | | | |
| IS | $0.037 \pm 0.002$ | $0.038 \pm 0.004$ | $0.034 \pm 0.002$ | $0.039 \pm 0.003$ | $0.039 \pm 0.002$ |
| Signle-draft SD | $0.036 \pm 0.000$ | $0.037 \pm 0.003$ | $0.038 \pm 0.004$ | $0.037 \pm 0.003$ | $0.038 \pm 0.002$ |
| SpecInfer | $0.035 \pm 0.003$ | $0.039 \pm 0.004$ | $0.035 \pm 0.003$ | $0.034 \pm 0.009$ | $0.036 \pm 0.003$ |
| SpecTr | $0.039 \pm 0.001$ | $0.037 \pm 0.001$ | $0.039 \pm 0.001$ | $0.036 \pm 0.002$ | $0.035 \pm 0.001$ |

Similarly, Table 7 and Table 8 show the ROUGE-L and BLEU scores for the case of non-identical draft models, corresponding to the experiment in Figure 3 in Section 5.1.

Table 7: ROUGE-L scores on the XSum task across various decoders and sampling temperatures.

| | | | Temperature | | |
|---|---|---|---|---|---|
| Draft 1 | | | 1.2 | | |
| Draft 2 | 1.2 | 1.6 | 2.0 | 2.4 | N/A |
| **Decoder** | | | | | |
| IS | $0.187 \pm 0.004$ | $0.189 \pm 0.007$ | $0.189 \pm 0.001$ | $0.191 \pm 0.002$ | – |
| SpecInfer | $0.184 \pm 0.004$ | $0.190 \pm 0.003$ | $0.185 \pm 0.006$ | $0.189 \pm 0.006$ | – |
| Single-draft SD | – | – | – | – | $0.190 \pm 0.006$ |

Table 8: BLEU scores on the WMT dataset across various decoders and sampling temperatures.

| | | | Temperature | | |
|---|---|---|---|---|---|
| Draft 1 | | | 1.2 | | |
| Draft 2 | 1.2 | 1.6 | 2.0 | 2.4 | N/A |
| **Decoder** | | | | | |
| IS | $0.036 \pm 0.003$ | $0.035 \pm 0.002$ | $0.036 \pm 0.002$ | $0.035 \pm 0.002$ | – |
| SpecInfer | $0.035 \pm 0.003$ | $0.038 \pm 0.005$ | $0.041 \pm 0.002$ | $0.040 \pm 0.002$ | – |
| Single-draft SD | – | – | – | – | $0.036 \pm 0.000$ |

## K  NOTATIONS

Table 9: Table of notations summarizing symbols and their descriptions.

| Symbol | Description |
|---|---|
| $x_{1:t}$ or $x_1^t$ | Sequence of tokens $x_1, x_2, \cdots, x_t$ |
| $K$ | Number of draft models |
| $\Omega$ | Vocabulary of tokens |
| $n$ | Size of the vocabulary, $|\Omega|$ |
| $\mathcal{S}$ | The set of valid draft tokens under consideration at every step |
| $p$ | Distribution of the draft model at any step, with $p_i$ denoting the probability of token $i$ |
| $q$ | Distribution of the target model at any step, with $q_i$ denoting the probability of token $i$ |
| $\mathcal{M}_s$ | Draft model |
| $\mathcal{M}_b$ | Target model |
| $u^t$ | Context sequence as a sequence of $t$ tokens in $\Omega^t$ |
| $P^*(\text{acc})$ | Optimal acceptance probability |
| $\beta_y(x_1, \ldots, x_K)$ | Conditional probability that token $y$ is selected given the input tokens $x_1, \ldots, x_K$ |
| $\beta_y^*(x_1, \ldots, x_K)$ | Optimal Conditional probability that token $y$ is selected given the input tokens $x_1, \ldots, x_K$ |
| $Y_I$ | The selected token from the importance weighted sampling step |
| $Z$ | The final output token after the combined importance weighted sampling and speculative sampling step |
| $p_I(\cdot)$ | Distribution of the output token in the importance weighted sampling step |
| $I$ | Index of the selected token n the importance weighted sampling step |
| $w_{i,j}$ | Conditional probability that token $i$ is selected given the input tokens are $i$ and $j$ |
| $\alpha_{i,j}$ | Joint probability that token $i$ is selected and the input tokens are $i$ and $j$ |
| $\alpha_{i,j,k}$ | Joint probability that token $i$ is selected and the input tokens are $i$, $j$, and $k$ |
| $s$ | LP-Truncation Threshold parameter |
| $\Omega_0$ | High probability subset of tokens of the vocabulary $\Omega$ |

