# OpenReview forum: "Multi-Draft Speculative Sampling: Canonical Decomposition and Theoretical Limits"
_ICLR.cc/2025/Conference — ICLR 2025 Spotlight_

### Official Review · Reviewer_yfZp · 2024-10-22

**Soundness:** 3
**Presentation:** 3
**Contribution:** 2
**Rating:** 6
**Confidence:** 2

**Summary:**

This paper studies the problem of multi-draft speculative sampling, where the optimal scheme can be cast as a solution to a linear program. The key idea behind this work is that this optimal scheme can be decomposed into a two-stage solution. Based on this idea, the authors provide some theoretical findings towards the two-stage solution. Motivated by the analysis, the authors propose a new class of token-level selection scheme based on weighted importance sampling, which shows consistent improvement in various scenarios.

**Strengths:**

1. The introduction of the two-step scheme are simple and intuitive, making it easy to understand.
2. The authors provide a solid theoretical analysis for token-level optimal draft selection.
3. The truncated LP algorithm is novel to the field of speculative decoding and theoretically better than SpecTr.
4. Extensive experiments, including different datasets, different decoding temperature and different draft models demonstrate the effective of proposed method.

**Weaknesses:**

1. I think the contributions of this work is a little weak. Given that speculative decoding shows poor performance with batch size larger than 1, the multi-draft sampling is usually not an appropriate option for speculative decoding. I think the authors should emphasize the contributions to real-world speculative decoding applications.
2. While I appreciate that the authors add some examples for better explanation, I think the authors should add a clear notation section with distinct description.
3. For the truncated LP draft selection algorithm, there exists an operation of sorting, which takes a complexity of $O(n\log n)$ and may influence the final speedup. The authors should report its real-world time consuming and compare to vanilla multi-draft selection, especially the empirical results on models with large vocab size, e.g., Llama 3.1 with vocab size 128k. Besides, for multi-draft scenarios with $K > 2$, the authors propose a multi-stage manner, which may affect the final speedup as well.
4.  I think the authors should conduct experiments on more benchmarks (e.g. HumanEval [1], GSM8K [2] and MT-bench [3]) for a fair comparison. Besides, the authors should report the real-world speedup to demonstrate the effectiveness of proposed method.

I know that conducting a wide range of evaluation experiments is costly and time consuming, and I will increase my score if the authors clearly address my concerns.

[1] Chen, Mark, et al. "Evaluating large language models trained on code." *arXiv preprint arXiv:2107.03374* (2021).

[2] Cobbe, Karl, et al. "Training verifiers to solve math word problems, 2021." *URL https://arxiv. org/abs/2110.14168* (2021).

[3] Zheng, Lianmin, et al. "Judging llm-as-a-judge with mt-bench and chatbot arena." *Advances in Neural Information Processing Systems* 36 (2023): 46595-46623.

**Questions:**

1. Can proposed truncated LP combined with SOTA speculative decoding (e.g. Medusa [4] and Eagle [5]) methods for better performance?
2. Why the authors do not discuss and compare a multi-draft selection method RRS [6]?
3. Could you please discuss the limitations of this paper?

[4] Cai, Tianle, et al. "Medusa: Simple llm inference acceleration framework with multiple decoding heads." *arXiv preprint arXiv:2401.10774* (2024).

[5] Li, Yuhui, et al. "Eagle: Speculative sampling requires rethinking feature uncertainty." *arXiv preprint arXiv:2401.15077* (2024).

[6] Jeon, Wonseok, et al. "Recursive speculative decoding: Accelerating llm inference via sampling without replacement." arXiv preprint arXiv:2402.14160 (2024).

---

> ### Author Response · Authors · 2024-11-22
>
> **Weakness**
>
> > I think the contributions of this work is a little weak. Given that speculative decoding shows poor performance with batch size larger than 1, the multi-draft sampling is usually not an appropriate option for speculative decoding. I think the authors should emphasize the contributions to real-world speculative decoding applications.
>
> We observed that whenever there is a moderate gap between the draft and target model,  the improvement in the acceptance rate in the multi-draft setting can be worth the additional compute over the single-draft scheme. This is already demonstrated in our experiments in Fig 3 and 4 where we show improved token rate over baseline schemes.
>
> The multi-draft setting also naturally lends itself to parallelism as each draft model can be allocated a different processor in a multi-node GPU system. This could alleviate the compute overhead while preserving the improved acceptance rate, leading to further speedups.
>
> Finally, the use of multiple drafts opens the possibility of training them using a variety of ensemble learning methods. This can lead to a diverse set of draft models that better approximate the target model and lead to higher acceptance rates.
>
>
> > While I appreciate that the authors add some examples for better explanation, I think the authors should add a clear notation section with distinct description.
>
> We have included a new section in the Appendix (Section K) summarizing the notation in the paper.
>
> > For the truncated LP draft selection algorithm, there exists an operation of sorting, which takes a complexity of $O(n\log n)$ and may influence the final speedup. The authors should report its real-world time consuming and compare to vanilla multi-draft selection, especially the empirical results on models with large vocab size, e.g., Llama 3.1 with vocab size 128k.
>
> The sorting operation mentioned by the reviewer is on the set of tokens selected after top-p sampling.
> As the distributions tend to be concentrated on a small number of tokens, we did not find the sorting operation to be a bottleneck. In any case it should be noted that top-p sampling, a commonly used technique in many real world LLM tasks also has the $O(n \log n)$ complexity We believe that the fact that we get speedups in the actual token rate in the experiments already demonstrates that the sorting operation is not resulting in slowdowns.
>
> Motivated by your suggestion, we have added a section on the computational complexity in Section G in the Appendix to illustrate the cost from the sorting operation and the LP program. In fact in our experiments we observed that in many cases the computational burden can be dominated by the LP program. Finally if needed, the sorting operation can also be entirely avoided in the truncated LP. We have proposed a variant in Appendix (Section G.1) that replaces the $O(n\log n)$ cost with $O(n \cdot s)$ where $s$ is our truncation threshold. This method also has the same theoretical guarantees as our original proposal.
>
>
> > Besides, for multi-draft scenarios with $K>2$ , the authors propose a multi-stage manner, which may affect the final speedup as well.
>
> We present a new experimental result in Section J.2 involving three drafts. We observe  gains in block efficiency and token rates similar to our other experiments.
>
> > I think the authors should conduct experiments on more benchmarks (e.g. HumanEval [1], GSM8K [2] and MT-bench [3]) for a fair comparison. Besides, the authors should report the real-world speedup to demonstrate the effectiveness of proposed method.
>
> We appreciate the  suggestion to conduct experiments on the additional tasks.  While we already have results on three diverse tasks: WMT, XSUM and  Dolly in the paper, we will to report experiments on some of these  should the paper be accepted.

---

> > ### Author Response · Authors · 2024-11-22
> >
> > **Questions**
> >
> > > Can proposed truncated LP combined with SOTA speculative decoding (e.g. Medusa [4] and Eagle [5]) methods for better performance?
> >
> > - In the token selection process EAGLE employs a multi-round speculative sampling algorithm as in SpecInfer. As noted in Algorithm 1 in that work, their token selection algorithm takes the target distribution $p$ and the draft distributions $\hat{p}_i$ and the generated tokens $t_i$ as input and outputs a sampled token $t$. The truncated LP appears applicable to this setting. We note however that the novelty in EAGLE is in improving a token drafter and not in the token selection which is the focus of our work. Instead, we have already provided comparisons to SpecInfer in the present paper.
> >
> > - The method in Medusa does not appear to guarantee that the selected token follows the distribution of the target model. Thus our results are not directly comparable with this work.
> >
> >
> > > Why the authors do not discuss and compare a multi-draft selection method RRS [6]?
> >
> > The main contribution of our paper is on efficient token selection schemes. Accordingly our experiments have focused on the case when draft tokens are sampled in an i.i.d.\ manner and different approaches for token selection are employed. In particular we compare our approach for token selection with two recently published works:  SpecInfer [1] and SpecTr [2].  RRS considers sampling without replacement which was not considered in our experiments. We believe that the token selection scheme in  RRS paper is the same as SpecInfer when one considers sampling with replacement.
> >
> > Furthermore we would like to note that our theoretical development in not restricted to sampling with replacement. Our main decomposition result (Theorem $1$) also applies in the setting of sampling without replacement. We have clarified this in the revised version of the paper in Remark 1 and Section B.1 in the Appendix.
> >
> > [1] Ziteng Sun et al., Spectr: Fast speculative decoding via optimal transport. Advances in Neural Information Processing Systems, 36, 2024.
> >
> > [2] Xupeng Miao et al., Specinfer: Accelerating large language model serving with tree-based speculative inference and verification. In Proceedings of the 29th ACM International Conference on Architectural Support for Programming Languages and Operating Systems, Volume 3, pages 932–949, 2024.
> >
> > > Could you please discuss the limitations of this paper?
> >
> > One limitation of this work is that it focuses on the class of token-level selection schemes. A scheme that is optimal at token level may not be optimal at the block level, which is a natural metric of interest in speculative decoding. In fact very few works till date have addressed block-level verification schemes for speculative sampling. It will be interesting to extend the work to develop new approaches for block level verification in multi-drafts settings.

---

> > > ### Comment · Reviewer_yfZp · 2024-11-25
> > >
> > > Thanks for the authors' detailed explanation. The additional experiments have addressed most of my concerns, and I am updating my score.

---

### Official Review · Reviewer_V7r6 · 2024-11-04

**Soundness:** 3
**Presentation:** 2
**Contribution:** 4
**Rating:** 8
**Confidence:** 2

**Summary:**

This paper studies the optimal token-level draft selection for a speculative sampling. In particular, speculative sampling is a method for accelerating autoregressive sampling by using smaller "draft" models to generate candidate tokens and a larger model to determine which tokens to accept. The key problem studied in this paper is how to propose tokens to optimize the probability they will be accepted. Importantly, increasing the acceptance probability leads to less calls to the larger model which in theory could increase the tokens per seconds. The key theorem of this paper is to characterize the necessary and sufficient conditions when two tokens are generated by the draft model. Interestingly, unlike the one token case, this paper shows that one can reach probability 1 acceptance even if the draft and large models have different token distributions. Finally, the paper ends with (i) heuristics to practically leverage the proposed sampling scheme, e.g., truncating the token distributions (ii) a heuristic scheme for leveraging the proposed importance sampling to larger sequences and (iii) empirical evidence of the effectiveness of the approach for K=2 and larger.

**Strengths:**

This paper provides a strong theoretical contribution to a very important topic. Speculative decoding serves as a refreshing and novel direction for accelerating LLM inference. This work puts the multi-draft model selection problem on better theoretical footing.

The results seem to be sound from my reading and no obvious errors were found.

**Weaknesses:**

By and large the biggest complaint I have with this draft is the background exposition. Prior to reading this I was not familiar with speculative decoding. While I understand space is tight, many key concepts such as the role and form of accepting probability in the speculative decoder were not clearly explained. This is a *key* aspect of the work, and the draft would strongly benefit for giving it more treatment.

To be honest, I couldn't understand anything the first time I read the paper and had to go back to the original speculative sampling paper [1] before I could piece things together.

Nevertheless, after reading, I now understand that this paper is targeting the acceptance criteria as it determines the number of candidate tokens used via a Bernoulli like-process.

This plus the strong theoretical analysis are enough for me to lean towards acceptance.

[1]Accelerating Large Language Model Decoding with Speculative Sampling,  https://arxiv.org/abs/2302.01318

**Questions:**

- Could you give some intuition about the conjecture in remark 2. It says that the exponent over the sum of the p-weights should match K (the number of tokens to generate per round). It's not at all clear to me why these should be so tightly coupled and why other lower order terms wouldn't appear.

---

> ### Author Response · Authors · 2024-11-22
>
> > By and large the biggest complaint I have with this draft is the background exposition. Prior to reading this I was not familiar with speculative decoding. While I understand space is tight, many key concepts such as the role and form of accepting probability in the speculative decoder were not clearly explained. This is a key aspect of the work, and the draft would strongly benefit for giving it more treatment. To be honest, I couldn't understand anything the first time I read the paper and had to go back to the original speculative sampling paper [1] before I could piece things together. Nevertheless, after reading, I now understand that this paper is targeting the acceptance criteria as it determines the number of candidate tokens used via a Bernoulli like-process. This plus the strong theoretical analysis are enough for me to lean towards acceptance.
>
> We have included a review of speculative sampling in Appendix A in the revision and in particular highlighted the role of the acceptance probability.
>
> > Could you give some intuition about the conjecture in remark 2. It says that the exponent over the sum of the p-weights should match K (the number of tokens to generate per round). It's not at all clear to me why these should be so tightly coupled and why other lower order terms wouldn't appear.
>
> Our first discovery of such a relation was in the context of Theorem $2$ where we provide a necessary and sufficient condition for the acceptance probability to equal $1$. As explained in Appendix D, we noticed numerically that for the case of $K$ drafts, a natural counterpart of Theorem $2$ exists with the power of $2$ simply replaced by $K$. To provide further intuition, we have included a new subsection (Appendix D.1) where we consider the case of $K=3$ drafts and an alphabet of size $3$ in detail. Our key observation is that when we establish the augmented system of inequalities, only terms of the form $(\sum_{s\in {\mathcal S}}p_i)^3$  survive for the draft probabilities and hence lower order terms do not appear. This observation then motivated the expression in Remark 2, which was of course verified numerically in several cases.

---

> > ### Comment · Reviewer_V7r6 · 2024-11-27
> >
> > Thank you for the clarifications. After reading the other reviews and discussion I think this is still a strong paper an lean towards acceptance.

---

### Official Review · Reviewer_6EsD · 2024-11-05

**Soundness:** 4
**Presentation:** 4
**Contribution:** 4
**Rating:** 8
**Confidence:** 4

**Summary:**

The paper studies the problem of multi-draft speculative sampling and makes new observations and provides a new algorithm. Of the new observations, Theorems 2 and 3 standout which characterize the optimal acceptance rate for two drafts and further show when the acceptance rate is one. The new algorithm exploits certain structure in the optimal transport problem and provides a clever approximate solution.

**Strengths:**

The paper is well written and easy to follow. Theorems 2 and 3 are novel and the experimental results are convincing.

**Weaknesses:**

See questions section below.

**Questions:**

- In Figure 3, experiments are conducted for draft temperatures > 1.0. However typically, target model temperature is often set to 1.0 or less than 1.0. Given this, I was expecting draft temperatures also to be <= 1.0. I was wondering how these draft temperatures were chosen?

- Would it be possible to say how the gap between the proposed approach and previous approaches vary as the target model temperature varies from 0.0 to 1.0?

- Theorems 2  and 3 provide a nice characterization of acceptance probability. However, given two distributions p, q, evaluating the condition seems to require 2^{vocabulary size} number of computations. Is there a faster way to evaluate these quantities?

- There are some approximate ways to solve the optimal transport by Sinkhorn methods e.g., https://amsword.medium.com/a-simple-introduction-on-sinkhorn-distances-d01a4ef4f085. I was wondering if these ideas are applicable here?

- Architectures in ML, typically refer to neural architectures. Given several works which propose different drafting methods for speculative decoding, I was thinking "canonical architectures" refers to some new drafters. It might be good to clarify this distinction early on in the paper.

- Can you please expand on what token rate means? Is it the number of tokens / second?

---

> ### Author Response · Authors · 2024-11-22
>
> > In Figure 3, experiments are conducted for draft temperatures $> 1.0$. However typically, target model temperature is often set to 1.0 or less than 1.0. Given this, I was expecting draft temperatures also to be $<= 1.0$. I was wondering how these draft temperatures were chosen? Would it be possible to say how the gap between the proposed approach and previous approaches vary as the target model temperature varies from 0.0 to 1.0?
>
> Our proposed IS scheme outperforms the baselines in terms of block efficiency for most temperature parameters. However it only provides improvements in the token rates when the block efficiency improvement is large enough. Indeed, when there is a "big enough" gap between the target and the draft model we can better demonstrate how the various methods are able to close it. This motivated the choice of the temperature parameters in our experiments. We have also added Fig. 4 in Section 5 in the main paper based on your suggestion. In this new experiment we vary the target model temperature from 0.2 to 1.0 and the draft model temperatures are set to 1.0. We did experiment setting temperatures of the draft model below 1, but the gains in acceptance rate were not big enough (c.f. Table 2), so there wasn't enough "room" to demonstrate gains with  better schemes.
>
>
> > Theorems 2 and 3 provide a nice characterization of acceptance probability. However, given two distributions $p$, $q$, evaluating the condition seems to require $2^\text{vocabulary size}$ number of computations. Is there a faster way to evaluate these quantities?
>
> We did not focus on actual computation of the acceptance probability, as our proposed truncated LP scheme does not require this computation for implementation. We do compute the  optimal acceptance probability in Table 2. Here we first apply top-k sampling with $k=5$, so the resulting vocabulary is small and an exhaustive search can be applied.
>
> >There are some approximate ways to solve the optimal transport by Sinkhorn methods e.g., https://amsword.medium.com/a-simple-introduction-on-sinkhorn-distances-d01a4ef4f085. I was wondering if these ideas are applicable here?
>
> We believe that a direct application of the Sinkhorn method to the optimal transport formulation in the K-draft setting of Sun et al [1] may still be slow in practice. In particular, the cost matrix associated with the optimal transport formulation  will have $\Omega(n^K)$  non-zero entries, where $n$ denotes the alphabet size. As a result each iteration of the Sinkhorn algorithm could have a computational cost $\Omega(n^K)$, which may  not result in improvements to the token rate.
>
> [1] Ziteng Sun et al., Spectr: Fast speculative decoding via optimal transport. Advances in Neural Information
> Processing Systems, 36, 2024.
>
> >Architectures in ML, typically refer to neural architectures. Given several works which propose different drafting methods for speculative decoding, I was thinking ``canonical architectures" refers to some new drafters. It might be good to clarify this distinction early on in the paper.
>
> We have replaced the term "canonical architecture"  with "canonical decomposition" in most of the paper.
>
> >Can you please expand on what token rate means? Is it the number of tokens / second?
>
> Yes we define it as follows:
> $$
> \mathrm{Token~Rate} = \frac{\text{Number of Decoded Tokens}}{\text{Total Time Elapsed (seconds)}}.
> $$
>
> Instead of reporting the absolute value of token rate, in our experiments we report the percentage improvement in the token-rate  with respect to the single draft baseline, to make it easier to note the improvement from having multiple drafts.

---

### Official Review · Reviewer_aVvF · 2024-11-06

**Soundness:** 4
**Presentation:** 3
**Contribution:** 4
**Rating:** 8
**Confidence:** 3

**Summary:**

Previous papers showed the optimality for a token selection rule when one token was chosen from the draft model. The authors do the same for the case when two tokens are selected. The authors decompose the problem of token selection into two steps --- first select one of K tokens using an importance weighted sampling technique, and then using the optimal single token selection proposed in previous papers. Finding the optimal token selection rule then reduces to finding the best importance weighted sampling technique.

The authors then show a necessary and sufficient condition for the existence of an optimal token selection rule where two tokens are selected from the draft. They then present an algorithm to solve the resulting linear program faster than the O(n^2) standard solution.

The authors compare the Importance sampling algorithm with SpecInfer on various tasks, and show that IS comes closer to optimal than SpecInfer.

**Strengths:**

The results in the paper are quite original and significant, to my knowledge, which is a bit limited. The experiments show that the proposed algorithm works well in practice.

**Weaknesses:**

I would have liked the presentation in Section 4 to be a little clearer - request you to write out the full linear program, and then explain the truncation a bit more - what is the truncated program computing?

In the experiment section, please include results for Temperature < 1, which is much more common in practice.

**Questions:**

I would like to see a comparison to the results in https://openreview.net/forum?id=9KxnxWOBA5

---

> ### Author Response · Authors · 2024-11-22
>
> > I would have liked the presentation in Section 4 to be a little clearer - request you to write out the full linear program, and then explain the truncation a bit more - what is the truncated program computing?
>
> We have rewritten Section 4 to make the presentation clearer. We first present the full linear program and introduce the variables $w_{i,j}$ in the optimization program. Thereafter we explain the truncation procedure and how we reduce the number of variables involved in the optimization. We hope that the revised exposition makes the presentation clearer.
>
>
> > In the experiment section, please include results for Temperature $< 1$, which is much more common in practice.
>
> We have added Figure 4 in Section 5. In this experiment, we vary the target model temperature from 0.2 to 1.0 and the draft model temperatures are set to 1.0.
>
> While our proposed IS scheme outperforms the baselines in terms of block efficiency for most temperature parameters, it can provide improvements in the token rates when the block efficiency improvement is "large enough". When the draft temperature is set to less than $1$ we observed that all methods appear to have acceptance probability close to each other (c.f. Table 2). Thus there is not enough "room" for improvement with better schemes.
>
>
> > I would like to see a comparison to the results in https://openreview.net/forum?id=9KxnxWOBA5
>
> Thank you for bringing this paper to our attention. We explain the differences with our paper below.
>
> - Theorem 1 in our paper, which presents the canonical decomposition result (see Fig. 1) is not present in the other paper. Our result shows how importance weighted sampling and speculative decoding can be elegantly combined for optimal token selection in the multi-draft setting. Building upon this theoretical result, we propose a practical token selection scheme that mimics the proposed decomposition  and provides a a way to trade-off  the computational speed with the acceptance probability. We present experimental results that demonstrate that our proposed scheme can achieve improvements over baselines in standard metrics such as token-rate and block-efficiency.
>
> - The other paper focuses on computing the optimal acceptance probability and provides a generalization of Theorem 3 in our work, which only considers the case of two (identical) drafts. However in Appendix D and Section 3 in our paper, we also provide a procedure for numerically  verifying the generalization (of Theorem $2$) beyond two drafts. This numerical procedure is based on computing the dual representation of polyhedral cones using a technique known as the double description method. We believe that such an approach could also be of significant interest to the research community, in addition to the analytical results.
>
> - The experiments reported in the other paper appear to be limited to a  specific draft sampling scheme and primarily use the average token acceptance probability as a performance metric. In contrast, our paper presents a new token level selection scheme and evaluates its performance using standard metrics (token rate and block efficiency) and demonstrates  improvements over baselines in a number of different tasks.
>
> - We noticed that in the "author response" the authors of the other paper have raised a question about how we computed the acceptance probability in Table 2 in our paper and noted that a brute-force search may not be practical. We would like to clarify that we had explicitly mentioned (in the paragraph below table 2) that for this table top-k sampling  with $k=5$ was used in this experiment. This made it feasible to compute the optimal acceptance probability using Theorem 3. We also note that in many of our other experiments in the paper, we use the more commonly deployed top-$p$ sampling with a default value of $p=0.95$. To clarify our paper, we have put experimental results involving top-k and top-p sampling in separate sub-sections in Section 5.

---

> > ### Comment · Reviewer_aVvF · 2024-11-24
> > **Thank you!**
> >
> > Thank you for your comments. Please do include the detailed comparison in your final version. I will keep my recommendation.

---

### Author Response · Authors · 2024-11-22

We thank the reviewers for their comments. We have uploaded a revised version of the paper, and we summarize the main changes here:

- **New Experimental Results**: We have added a new experimental result (Figure 4 in Section 5 of the main paper) where the temperature of the target model is varied between 0.2 and 1.0, while the temperature of the draft models is set to 1.0. In Appendix J1, we present an additional experimental result where both drafts have the same temperatures, and in Appendix J2, we present an experimental result involving three draft models. These results complement the experimental results in Figure 3 in the main paper.

- **Main Result**: Our result in Theorem 1 is not restricted to a product distribution over the tokens; it applies to an arbitrary distribution. We have updated Remark 1 in the main paper and Appendix B.1 to indicate this. In particular, the decomposition result also applies if the draft tokens are sampled without replacement.

- **Truncated Linear Program**: We have revised Section 4 to better explain the truncated linear program. We also present the computational complexity of the truncated LP in Appendix G, as well as another variant that avoids the sorting operation in the truncated LP.

- **Review of Speculative Sampling**: We provide a brief review of speculative sampling in Appendix A.

- **Notations**: We have added a section in the Appendix (Section K) summarizing the mathematical notations used in the paper.

---

### Meta-Review · Area_Chair_B7Au · 2024-12-23

**Metareview:**

This paper considers the problem of multi-draft speculative decoding and derives a decomposition of the optimal solution as an importance sampling step followed by a token-level verification step. This decomposition helps the authors focus on optimizing a valid importance sampling distribution in the context of speculative decoding. For the case of two drafts, Theorems 2 and 3 fully characterize the optimal acceptance rate and further show when the acceptance rate is one. In particular, it is shown that the acceptance rate could be one even if the two distributions are not the same, which is unlike the guarantees in prior work. The decomposition also leads to a new algorithm that exploits the structure of the optimal transport problem and provides a clever approximate solution. Overall, the paper makes significant theoretical and experimental progress in multi-draft speculative decoding that deserves to be highlighted in the conference. Congratulations to the authors!

P.S. It would be nice to discuss and relate to the concurrent work on deriving the optimal acceptance rate of multi-draft speculative decoding (https://openreview.net/forum?id=9KxnxWOBA5) in the camera-ready version of the paper.

**Additional Comments On Reviewer Discussion:**

All reviewers agreed that this paper makes significant theoretical and empirical contributions to multi-draft speculative decoding that deserve to be highlighted in the conference.

---

### Decision · Program_Chairs · 2025-01-22

Accept (Spotlight)